# Spatial genomics maps the structure, nature and evolution of cancer clones

Artem Lomakin[1,2,3,15], Jessica Svedlund[4,15], Carina Strell[4,5], Milana Gataric[1,2], Artem Shmatko[3], Gleb Rukhovich[2,3], Jun Sung Park[1,2,3], Young Seok Ju[6], Stefan Dentro[1,2,3], Vitalii Kleshchevnikov[2], Vasyl Vaskivskyi[2], Tong Li[2], Omer Ali Bayraktar[2], Sarah Pinder[7,8], Andrea L. Richardson[9], Sandro Santagata[10,11,12], Peter J. Campbell[2], Hege Russnes[13,14], Moritz Gerstung[1,3 ✉], Mats Nilsson[4 ✉] & Lucy R. Yates[2 ✉]

Genome sequencing of cancers often reveals mosaics of different subclones present in the same tumour[1–3]. Although these are believed to arise according to the principles of somatic evolution, the exact spatial growth patterns and underlying mechanisms remain elusive[4,5]. Here, to address this need, we developed a workflow that generates detailed quantitative maps of genetic subclone composition across whole-tumour sections. These provide the basis for studying clonal growth patterns, and the histological characteristics, microanatomy and microenvironmental composition of each clone. The approach rests on whole-genome sequencing, followed by highly multiplexed base-specific in situ sequencing, single-cell resolved transcriptomics and dedicated algorithms to link these layers. Applying the base-specific in situ sequencing workflow to eight tissue sections from two multifocal primary breast cancers revealed intricate subclonal growth patterns that were validated by microdissection. In a case of ductal carcinoma in situ, polyclonal neoplastic expansions occurred at the macroscopic scale but segregated within microanatomical structures. Across the stages of ductal carcinoma in situ, invasive cancer and lymph node metastasis, subclone territories are shown to exhibit distinct transcriptional and histological features and cellular microenvironments. These results provide examples of the benefits afforded by spatial genomics for deciphering the mechanisms underlying cancer evolution and microenvironmental ecology.

Cancers are complex and dynamic entities that are constantly reshaped by the interactions between neoplastic cells and their microenvironments[4–6]. Whole-genome sequencing (WGS) analysis of the average cancer detects thousands of somatic mutations and multiple genetically related yet distinct groups of cells termed 'subclones'[2,7,8]. However, as genomic technologies typically assay DNA from dissociated tissues, the phenotypic consequences and the ecosystem pressures that are critical to fully understanding cancer evolution are lost[9,10]. Consequently, relatively little is currently known about the nature or causes of spatial patterns of cancer growth, phenotypic characteristics of distinct subclonal lineages or their interactions with tissue ecosystems[11]. Still, this information appears key because adverse cancer outcomes—growth, progression and recurrence—are properties of genetically distinct subclones[3,12–16].

Lineage tracing using somatic mutations is a powerful tool for inferring the ancestral relationships between cancer subclones, but methods to perform this in preserved human tissue context are lacking[3,14,17–20]. Histology-driven sampling, such as laser capture microdissection (LCM)[21], combined with low-input nucleic acid library sequencing or even single-cell sequencing goes some way towards resolving subclone spatial structure[19]. However, even the most exhaustive sampling strategy will struggle to provide an unbiased representation of the cancer clone territories, particularly across whole-tumour sections. Recently described spatial genomics approaches permit the de novo spatial detection of cancer clones with distinct copy number profiles, but this does not permit the detection of point mutations or quantitative read outs of intermixed clones[22,23]. It has previously been demonstrated that individual mutations can be detected in situ using in situ hybridization[24] or mutation-specific padlock probes[25–28]. However, these approaches are limited by the number of available fluorophores. Given that every cancer and subclone therein is genetically unique, to reconstruct ancestral relationships in both space and time, we need to be able to trace multiple, cancer-specific somatic mutations simultaneously[8].

To address this need, we developed a genetic clone mapping workflow that is centred around base-specific in situ sequencing (BaSISS)

[1]European Molecular Biology Laboratory, European Bioinformatics Institute (EMBL-EBI), Hinxton, UK. [2]Wellcome Sanger Institute, Hinxton, UK. [3]Division of AI in Oncology, German Cancer Research Centre DKFZ, Heidelberg, Germany. [4]Science for Life Laboratory, Department of Biochemistry and Biophysics, Stockholm University, Solna, Sweden. [5]Department of Immunology, Genetics and Pathology, Uppsala University, Uppsala, Sweden. [6]Laboratory of Cancer Genomics, GSMSE, KAIST, Daejeon, Korea. [7]Guys and St Thomas' NHS Trust, London, UK. [8]School of Cancer & Pharmaceutical Sciences, King's College London, London, UK. [9]Department of Pathology, John Hopkins Medicine, Baltimore, MD, USA. [10]Department of Pathology, Brigham and Women's Hospital, Harvard Medical School, Boston, MA, USA. [11]Laboratory of Systems Pharmacology, Harvard Program in Therapeutic Science, Boston, MA, USA. [12]Ludwig Center at Harvard, Harvard Medical School, Boston, MA, USA. [13]Department of Pathology, Institute for Cancer Research, Oslo University Hospital, Oslo, Norway. [14]Institute of Clinical Medicine, University of Oslo, Oslo, Norway. [15]These authors contributed equally: Artem Lomakin, Jessica Svedlund. ✉e-mail: moritz.gerstung@dkfz.de; mats.nilsson@scilifelab.se; ly2@sanger.ac.uk

technology. We derived quantitative maps of multiple genetic clones in eight tissues from two multifocal breast cancers that span the main histological stages of early cancer progression: ductal carcinoma in situ (DCIS), invasive cancer and lymph node metastasis[29]. In a case of DCIS, clones exhibited co-existence and segregation patterns in different parts of the breast ductal anatomy. By integrating genetic clone maps with multimodal spatial data layers, we found that genetically similar regions can be scattered across wide areas yet maintain similar transcriptional and histological features and foster recurrent ecosystems. Finally, we found that genetic progression, which encapsulates the historical order of events, does not necessarily translate directly to transitions in histological state that are commonly assumed to reflect the stages of cancer progression, thus warranting a combined genetic and histological assessment of cancer evolution.

## The BaSISS workflow

The BaSISS workflow is centred around fresh frozen tissue blocks that undergo serial cryosectioning to generate tissue for bulk WGS and z-stacked sections for in-tissue spatial clone mapping and spatial phenotyping (Fig. 1a). Following subclone detection from bulk WGS data, there are three core BaSISS steps. First, to facilitate detection of multiple clones of interest, BaSISS padlock probes with sequence-specific oligonucleotide target recognition arms are designed towards both mutant and wild-type alleles of clone-defining somatic variants. A unique 4–5 nucleotide reader barcode on each probe enables multiplexing[27]. BaSISS targets can take the form of any expressed somatic mutation, including point mutations and rearrangement breakpoints, and can be supplemented with copy number alterations (Supplementary Table 1). Second, BaSISS and transcript detection are performed as previously described for gene expression ISS using cyclical microscopy[27,30] (Fig. 1a and Supplementary Methods).

Third, continuous spatial subclone maps are generated using a statistical algorithm that exploits BaSISS signals as well as local cell counts (derived from the DAPI channel during the fluorescence microscopy of BaSISS) using two-dimensional Gaussian processes (Extended Data Fig. 1 and Supplementary Methods). The variational Bayesian model also accounts for unspecific or wrongly decoded signals and variable probe efficiency and is augmented by variant allele fractions in the bulk genomic sequencing data. In an optional, fourth characterization step, BaSISS clone maps can be aligned and integrated with additional layers of spatial phenotype data. In this study, we performed spatially resolved single-cell transcriptomics using targeted ISS (using a previously published 91 gene oncology, a novel 62 gene immune panel and drawing on published single-cell RNA sequencing data)[30,31] and immunohistochemistry (IHC) staining (Extended Data Fig. 2a–c and Supplementary Methods). Additional sections were obtained to perform validation of our workflow using LCM and low-input WGS as previously described[32] (Extended Data Fig. 3a).

## Two cases of multifocal breast cancer

The cohort includes eight tissue blocks from two patients (P1 and P2) who underwent a surgical mastectomy for a multifocal breast cancer. These patients were selected to permit a comparison between genetic and histological progression models in early breast cancer development[29] (Fig. 1b,c). P1 had two separate oestrogen receptor (ER)-positive, human epidermal growth factor receptor 2 (HER2)-negative primary invasive breast cancers (PBCs) within a 5-cm bed of DCIS; we used tissue blocks from both PBCs (samples P1-ER1 and P1-ER2) and three regions from DCIS (samples P1-D1, P1-D2 and P1-D3). P2 had two separate PBCs of the 'triple-negative' subtype (lacking the ER, progesterone receptor and HER2). We sampled both PBCs (samples P2-TN1 and P2-TN2) and an axillary lymph node that contained metastatic cancer deposits (sample P2-LN1) (Fig. 1b).

## Accurate and reproducible maps of clones

To demonstrate that spatial BaSISS signal counts can provide a meaningful read out of the underlying somatic genotype, we first focused on three samples from P1 (P1-ER1, P1-ER2 and P1-D1) (Fig. 1b). Previous multiregional WGS experiments identified mutation clusters that equated to six phylogenetic tree branches, and these were present at different levels across the three samples[3] (Fig. 2a,b, Extended Data Fig. 3b and Supplementary Table 3). To enable spatial detection of subclones, BaSISS padlock probes were designed towards 51 alleles that report on each branch of the phylogenetic tree: 25 single-base substitutions and the equivalent wild-type base, as well as an amplified oncogene (*FGFR1*) (black numbers; Fig. 2b and Supplementary Table 1). Subclones are referred to by a patient identifier and the colour of the corresponding node of the phylogenetic tree: P1-purple, P1-red, P1-grey, P1-orange, P1-green and P1-blue (Fig. 2b). A subclone genotype comprises the branch mutations accumulated as one moves from the tree root to the subclone node, therefore P1-green contains grey, blue and green branch mutations (Extended Data Fig. 3b). The bulk WGS-derived tree was corroborated by spatial co-occurrence of BaSISS signals and LCM–WGS validation data (Fig. 2c and Extended Data Figs. 3c,d and 4a).

On average, 97% of detected BaSISS spot signals were converted into feasible barcodes[33]. The median target-specific coverage across 300 mm² of breast tissue was 13,000-fold (Supplementary Table 4). BaSISS-derived variant allele fractions exhibited strong correlation across replicate experiments on serial tissue sections ($R = 0.76–0.93$, Pearson's; Extended Data Fig. 4b), demonstrating quantitative reproducibility.

BaSISS signals coloured according to their subclonal mutation branch revealed a first, albeit noisy, visual glimpse of subclonal growth structure (Fig. 2d). Broad patterns were preserved in technical replicate experiments using adjacent tissue sections (Extended Data Fig. 4c). Although the number of signals detected per nucleus ($n = 0.82$) does not provide single-cell resolution of the somatic genotype, it is possible to aggregate information (1) spatially over areas of approximately $100 \times 100$ µm², and (2) across alleles co-occurring in a particular subclone to infer the local clonal composition of different tumour clones and normal cells (Supplementary Table 4). This process generated detailed maps covering several squared centimetres of tumour tissues (Fig. 2e). Of note, the clone mapping algorithm also implicitly adjusts the observed allele frequencies for a range of systematic biases (Extended Data Fig. 4d), stemming from the use of RNA-derived signals, differential BaSISS probe sensitivity and allele confusion to produce highly consistent maps across replicates (Extended Data Fig. 4e). Although the raw BaSISS variant allele fractions of many probes were noisy owing to the aforementioned biases, the modelled allele frequencies were in highly accurate agreement with LCM–WGS validation data (Fig. 2c,g and Supplementary Table 5). This further corroborates the quantitative nature of BaSISS-derived clone maps that can be explored using an interactive web browser (https://www.cancerclonemaps.org/).

## Charting histogenomic relationships

Histology-driven sampling of well-defined stages of cancer progression can uncover mechanisms and markers of disease progression[10,19,29,34]. Up to two-thirds of PBCs contain both invasive cancer and intermixed DCIS, a non-obligate precursor lesion. How these distinct 'stages' of cancer development might relate to genetic diversification within the same tissue is generally unknown[35] (Fig. 1c). To demonstrate that BaSISS can chart these relationships across entire tissue sections, we examined three PBC samples with intermixed invasive and DCIS histology: P1-ER1, P1-ER2 and P2-TN1 (Fig. 3 and Extended Data Figs. 5 and 6a–c).

BaSISS detected 2–4 subclones per PBC in accordance with bulk WGS data. Clone maps (Fig. 3a,e) and the quantitative clonal composition of 73 individually annotated microregions (Fig. 3b,f and Extended Data

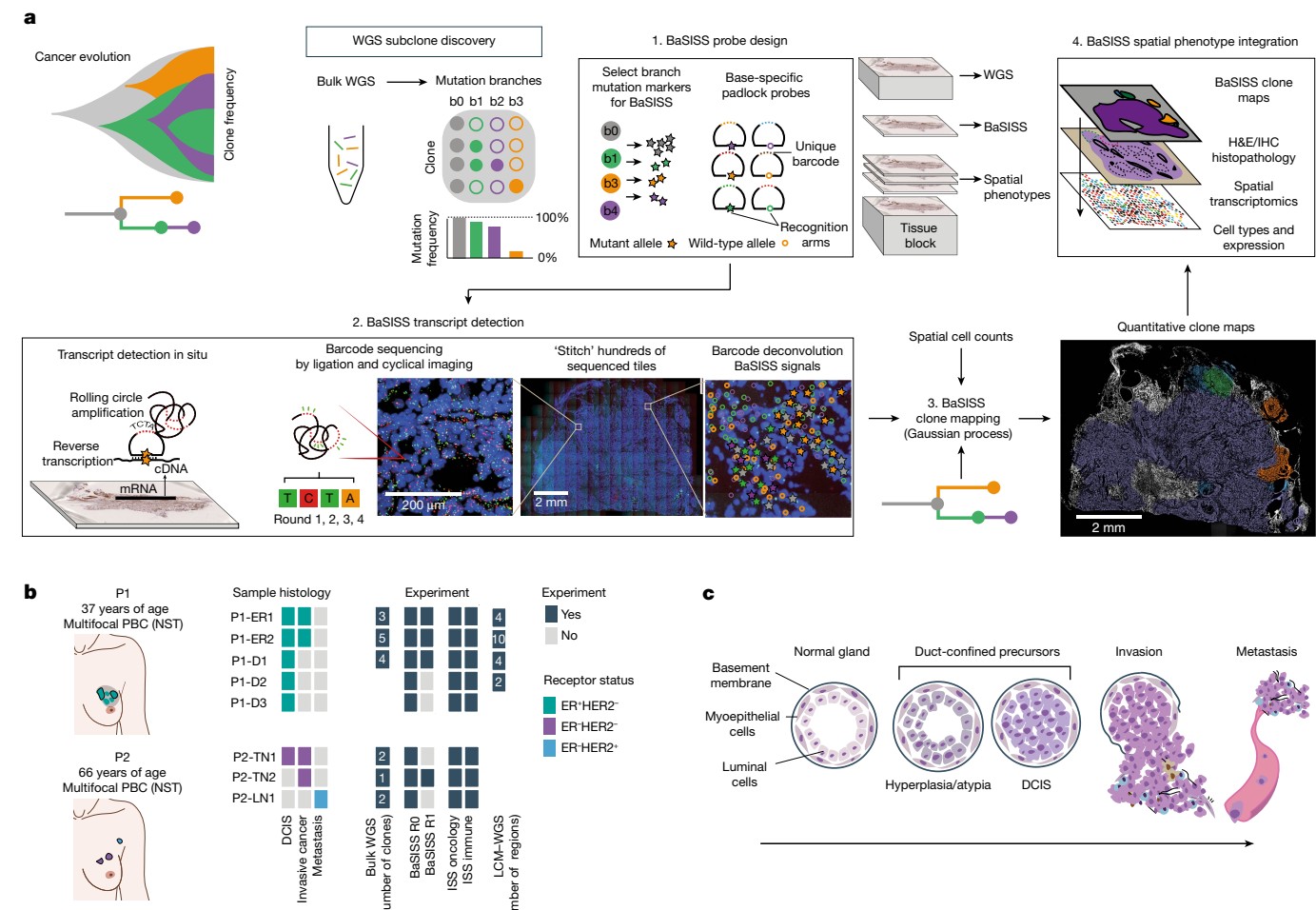

**Fig. 1 | The BaSISS workflow to generate cancer clone maps. a**, Following de novo mutation detection and subclone discovery in WGS data, the BaSISS workflow is performed as follows: (1) bespoke mutation-specific padlock probes are designed. (2) BaSISS transcripts are detected. To achieve this, BaSISS padlock probes hybridize to complementary DNA (cDNA) in situ. By virtue of a highly specific DNA ligase, only completely target-complementary padlock probes are ligated and form closed circles. Ligated probes are amplified through rolling circle amplification and their reader barcodes are detected in tissue space through sequencing by ligation with fluorophore-labelled interrogation probes and cyclical microscopy. (3) Mathematical modelling of BaSISS signals and the genotype of clones is then performed to derive clone maps. (4) Subsequent phenotype and microenvironment characterization of clones is then possible, by integrating clone fields with spatial datasets acquired from serial tissue sections. The BaSISS model and cell typing are described further in Extended Data Figs. 1 and 2. **b**, The two cases of multifocal primary breast cancer (PBC) used to develop the BaSISS approach. Coloured tiles report the histological features within each sample and the experiments performed. The number of clones identified by WGS and targeted by BaSISS are reported as white numerals. **c**, The traditional histological model of breast cancer progression. DCIS, ductal carcinoma in situ; H&E, haematoxylin and eosin; LN, lymph node; NST, invasive carcinoma of no special type; TME, tumour microenvironment.

Figs. 5a,b and 6a,b) revealed that individual subclones form spatial patterns that were, by varying degrees, related to the histological progression states. Normal tissue elements, including immune aggregates and histologically normal ducts, appear unstained consistent with a wild-type status for the targeted clones (green and yellow contours, respectively; Fig. 3a,e). In P1-ER2, an area of hyperplasia was predicted and confirmed by LCM–WGS to be genetically unrelated to the cancer (blue contour; Figs. 2c and 3a).

In each PBC, the genetic and histological progression models were broadly consistent, in which the invasive disease was mainly composed of cells from the most recently diverged subclone: P1-red, P1-purple and P2-purple in samples P1-ER1, P1-ER2 and P2-TN1, respectively (Fig. 3b,f). By contrast, earlier diverging clones colocalized entirely or in part to the histological pre-invasive lesion: DCIS. For example, in P1-ER2, BaSISS predicted that green branch mutations were completely absent from the invasive compartment, a conclusion that is supported by three separate microdissections (LCM–WGS) from distant regions of invasive cancer in P1-ER2 (Fig. 2c and Extended Data Fig. 5c).

However, in each PBC, there was a subclone that spanned both DCIS and invasive histology, revealing that disconnects between histological and genetic progression states can exist. This was the case for clone P1-red in P1-ER1 and clone P1-purple in P1-ER2. These DCIS-invasive spanning clones could be distinguished from each other by hundreds of private mutations, including different inactivating driver mutations in *PTEN*, indicating parallel evolution along these divergent lineages that resulted in two distinct instances of cancer invasion (total mutation numbers label the phylogenetic tree branches; Fig. 3b). The spatial predictions of the BaSISS model of intraductal acquisition of *PTEN* mutations and PTEN protein loss was confirmed by LCM–WGS and IHC, respectively (Fig. 2c and Extended Data Fig. 5d). In sample P2-TN1, the only predicted driver point mutation was a deleterious mutation in the tumour suppressor gene *TP53*, and this was detected in both DCIS and invasive compartments and was also present in all cancer regions of the second PBC, P2-TN2, consistent with an early onset in the development of this cancer (phylogenetic tree; Fig. 3e,f). These data therefore suggest

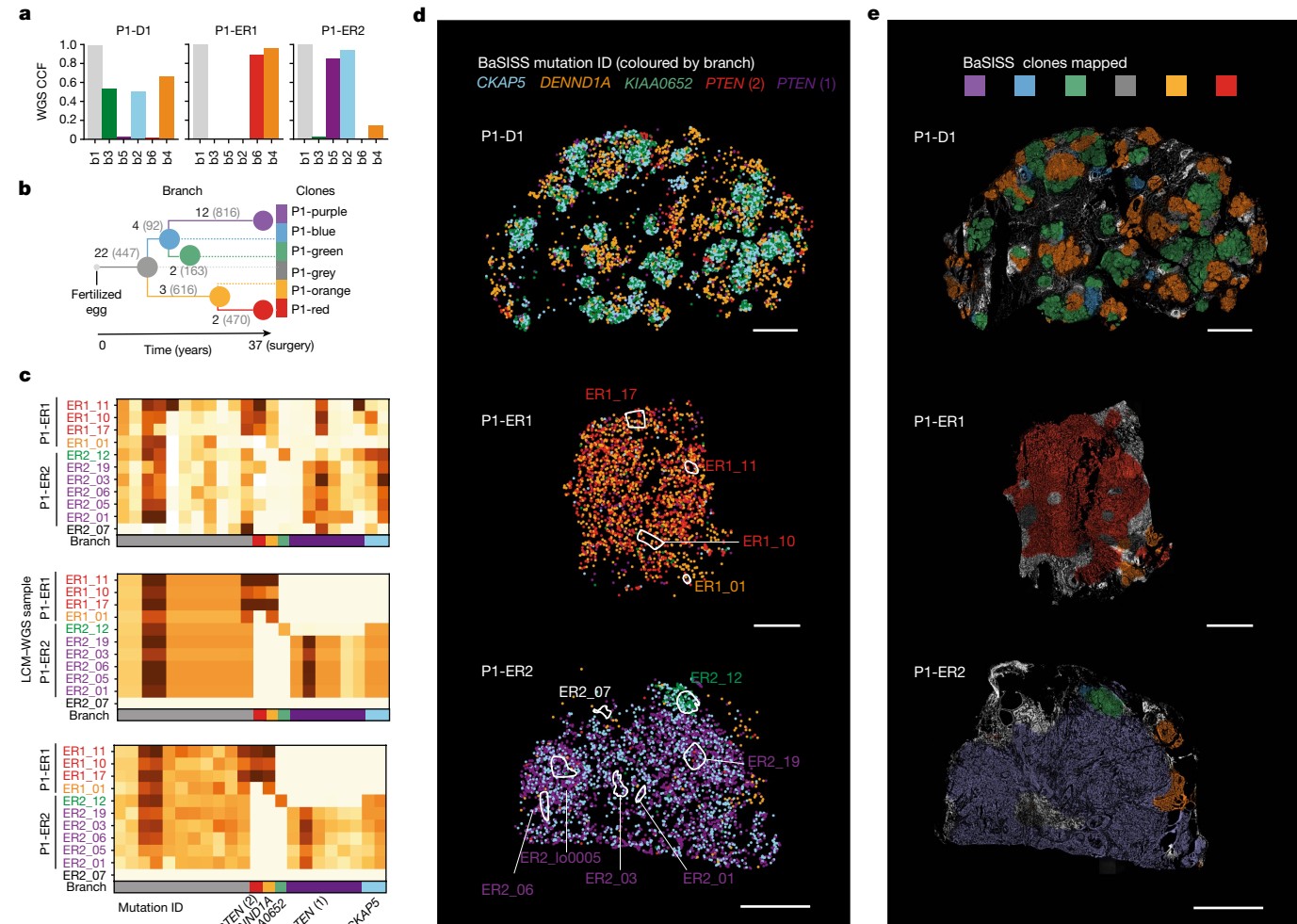

**Fig. 2 | Converting BaSISS spatial signals into maps of clones. a**, Bar plots of cancer cell fractions (CCFs) derived from bulk WGS of the P1 samples. **b**, Phylogenetic tree reconstructed from multiregional bulk WGS data from P1 (see Supplementary Methods for details). Each branch is labelled with the total number of WGS mutations defining the branch (grey text) and the number of BaSISS probes designed to target that branch (black text). **c**, Three heatmaps of variant allele fractions (VAFs) calculated using data derived from $n = 11$ regions of P1-ER1 and P1-ER2 (marked in **d**). Raw BaSISS VAFs (for each target mutation the number of mutant signals divided by total number of mutant plus wild-type signals) (top) and model-imputed BaSISS VAFs (middle) are derived from raw BaSISS signal data within these regions. In serial tissue cryosections, corresponding z-stack regions were identified and subjected to LCM–WGS. Resulting LCM–WGS VAFs are presented (bottom). Mean per-gene correlations are approximately 0.41 and 0.90 for BaSISS to LCM–WGS and model-imputed VAFs to LCM–WGS comparisons, respectively. Sample names are coloured according to the dominant BaSISS subclone in the sampled region. Each row represents a targeted mutation. The mutations plotted in **d** are labelled by their gene name; for *PTEN* there are two separate mutations. **d**, Spatial BaSISS detections of barcodes reporting on five selected mutations, coloured according to their targeted branch. White contours indicate LCM regions (relates to **c**). **e**, BaSISS clone maps in physical space projected on the DAPI image (nuclei are white), derived using BaSISS mathematical modelling of signals from 45 informative targets. Each clone has a different colour, and dominant clones are reported (shown if the CCF is more than 25% and the inferred local cell density is more than 300 cells per mm²). Scale bars, 2.5 mm (**d**,**e**).

that many, if not all, of the genetic events necessary to initiate the invasive transition in these three cancers were acquired within the ducts, and subsequently both intraductal expansion and stromal invasion ensued.

## Phenotypic changes accompany progression

Next, by integrating additional layers of spatial data, we sought to establish how phenotypic changes relate to genetic-state and histological-state transitions. In P1-ER1 and P1-ER2, consistent with a more proliferative phenotype, *PTEN*-mutant clone regions exhibited denser Ki-67 IHC nuclear staining, than *PTEN* wild-type ancestral clone regions (false-discovery rate (FDR) = 0.004 P1-red versus P1-orange; and FDR = 0.03 P1-purple versus P1-green) (Fig. 3c,d and Extended Data Fig. 5e). However, for a given genetic clone, the Ki-67 score was similar irrespective of whether it occupied a DCIS or invasive state, indicating that upregulation of Ki-67 is temporally related to acquisition of a *PTEN* mutation and precedes invasion.

By contrast, cellular resolution spatial transcriptomics analysis of P1-ER2 revealed that epithelial cell expression of several genes—*CLDN4* (encoding claudin 4), *ACTB* (encoding β-actin), *KRT5* (encoding keratin 5) and *CTSL2* (encoding lysosomal cysteine protease cathepsin V)—differed between DCIS and invasive compartments occupied by the same, P1-purple, clone (Extended Data Fig. 5f). These transcriptional changes might therefore be considered more closely linked to the histological transition rather than genetic changes traced by this approach. Expression of *CLDN4* was consistently lower in the invasive compartment than to each DCIS clone. However, for some genes such as *ACTB*, expression patterns changed in opposing directions in the invasive cancer relative to the sampled DCIS clone (expression is higher than P1-green DCIS

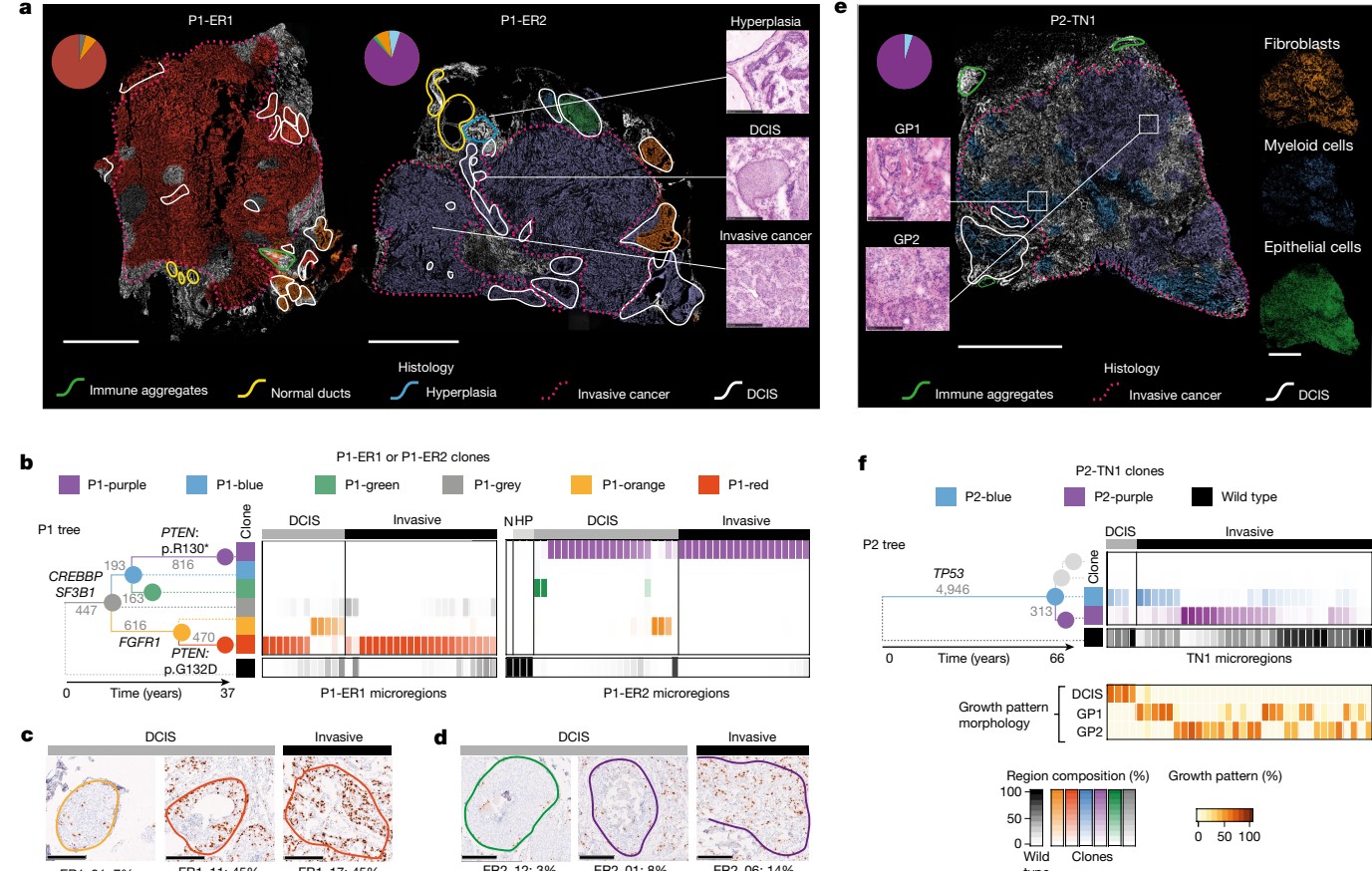

**Fig. 3 | Genetic clones mapped in histological context from three PBCs.** **a**, BaSISS maps of two PBCs from P1 with intermixed DCIS and invasive cancer. The most prevalent genetic clone is projected as a coloured field (corresponds to **b**) on DAPI images (reported if the CCF is more than 25% and the inferred local cell density is more than 300 cells per mm²). Scale bar, 2.5 mm. Pie charts report the WGS-estimated clone composition of P1-ER1 and P1-ER2. Inset images (right) are regions of P1-ER2 (H&E-stained serial tissue sections) that represent three histological progression states. Scale bar, 250 µm. **b**, The phylogenetic tree was inferred from P1 multiregion WGS: branches are scaled according to and annotated with the number of WGS mutations and driver mutation-containing genes. Branches and nodes are coloured to reflect the clones mapped in **a**. Heatmaps report clone composition in 34 and 44 histologically annotated epithelial-containing microregions of P1-ER1 and P1-ER2, respectively. Microregions include individual ducts or randomly selected similarly sized regions of invasive cancer (see Extended Data Figs. 4b and 5b and the web browser https://www.cancerclonemaps.org/ for microregion details). HP, hyperplasia; N, normal ducts. **c,d**, IHC in P1-ER1 (**c**) and P1-ER2 (**d**) for the proliferative marker Ki-67 in six clone territories (indicated by contour colour); the percentage of nuclei staining positive (brown) is reported. Scale bars, 250 µm. **e**, As in **a**, but a clone map of P2-TN1. Scale bar, 2.5 mm. Mini-images report ISS-derived cell types (right) and H&E tissue section snapshots of the two cancer growth patterns (GP1 and GP2) reported in P2-TN1 (left). Scale bar, 250 µm. **f**, Phylogenetic tree for P2 and heatmap of 36 P2-TN1 microregions, as in **b**. Branches relating to clones not detected in this sample (that is, only found in P2-LN1) are shaded grey. The bottom heatmap is the estimate by the histopathologist and reports the contribution of different growth patterns to the microregion, defined by distinct nuclear and architectural features (Supplementary Methods).

(FDR = 0.02) and lower than P1-purple DCIS (FDR = 0.013)) or were highly specific to a genetically more distant DCIS clone (Extended Data Fig. 5f).

Attempts to isolate the changes associated with invasive transition might also be confounded by heterogeneity within the invasive compartment. In P2-TN1, we therefore sought to examine whether the two genetically distinct invasive subclones (P2-blue and P2-purple) were phenotypically distinct. The two cancer clones exhibited distinct morphological (nuclear and architectural) features (*P* = 0.04, Fisher's exact test) (H&E image insets; Fig. 3e,f) and occupied neighbourhoods with different stroma (FDR = 0.02) and immune cells such as myeloid cell densities (FDR = 0.08) (mini-image insets; Fig. 3e and Extended Data Fig. 6a–c). Transcriptional programs were also distinct, with statistically significant differences in gene expression for 12 of 91 genes between clones (Extended Data Fig. 6d). Together, these data indicate that the particular clones sampled can have a profound effect on attempts to identify the phenotypic changes implicated in driving or arising during histological progression.

## Growth patterns of pre-invasive clones

To demonstrate that BaSISS can be used to chart growth patterns in relation to complex tissue structures, we turned our attention to three DCIS samples from P1 that spanned a tissue surface area of 224 mm² (P1-D1, P1-D2 and P1-D3) (Fig. 4a and Extended Data Fig. 7a). The adult female breast comprises multiple, branching ductal systems, termed lobes, that extend from the nipple surface to the acini of the lobules, as illustrated in Fig. 4c[36,37]. DCIS arises from the duct epithelium and is considered a lobar disease as it typically involves the ducts and lobules of a single lobe[38]. Although DCIS is known to be genetically heterogeneous[19], how DCIS clones are organized and grow through the wider duct system remains elusive[39].

The clone maps generated for the three samples formed striking mosaics of mainly green and orange, and occasional blue and grey that localized to areas of histologically confirmed DCIS (Fig. 4a and Extended Data Fig. 7a). Immune clusters and occasional normal or hyperplastic ducts appeared white (unstained), consistent with a different genetic

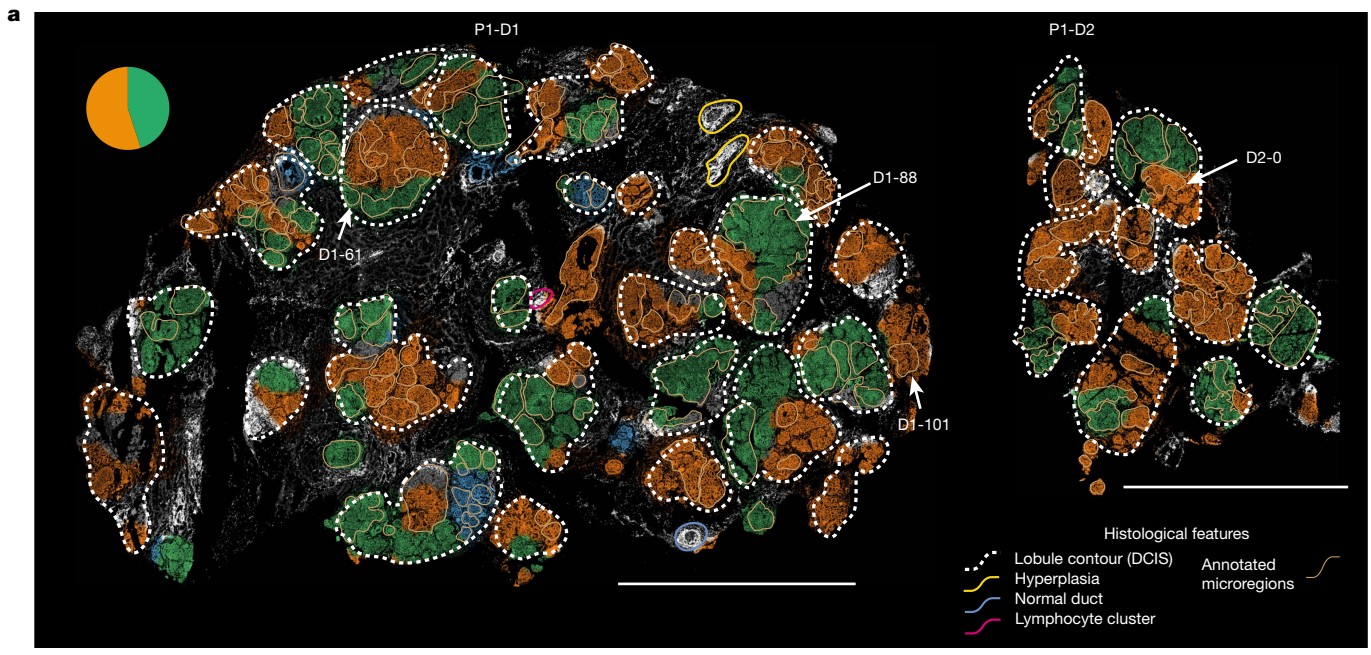

**a**, BaSISS maps of pure DCIS samples: P1-D1 and P1-D2.

*Histological features*
- Lobule contour (DCIS)
- Hyperplasia
- Normal duct
- Lymphocyte cluster
- Annotated microregions

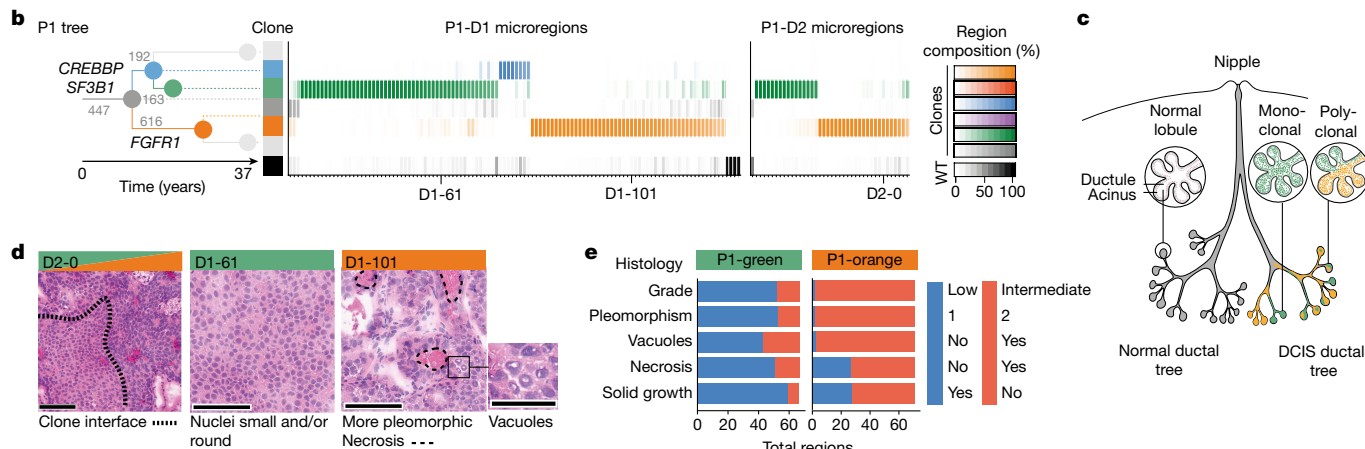

**Fig. 4 | Growth patterns and histological associations of DCIS clones.**
**a**, BaSISS maps of pure DCIS samples: P1-D1 and P1-D2. The most prevalent genetic clone is projected as a coloured field (which corresponds to **b**) on DAPI images (reported if the CCF is more than 25% and the inferred local cell density is more than 300 cells per mm²). Scale bar, 5 mm. The quantitative, continuous nature of these data can be examined via an interactive web browser (https://www.cancerclonemaps.org/). The pie chart reports the WGS-estimated clone composition of P1-D1. The white dashed contours delineate morphologically defined lobules. The beige contours mark 114 and 40 manually selected microregions in P1-D1 and P1-D2, respectively, the clonal composition of which is reported by the heatmaps in **b**. Microregions were manually selected and represent single or small groups of intimately related acini or ductules from the

same lobule. **b**, The phylogenetic tree was inferred from P1 multiregion WGS: branches are scaled according to and annotated with the number of WGS mutations and driver mutation-containing genes. Branches and nodes are coloured to reflect the clones mapped in **a**. Only branches detected in P1-D1 and P1-D2 are coloured. WT, wild type. **c**, Cartoon of a lobe of the breast with normal anatomy (left) and DCIS (right), with lobules exhibiting monoclonal and polyclonal involvement. **d**, H&E images report representative subclone histological features in regions selected from **a**. Scale bars, 100 µm and 50 µm (vacuoles). **e**, Stacked bar plot summarizes histological features of microregions dominated by P1-green (*n* = 66) or P1-orange (*n* = 72). Nuclear pleomorphism is a measurement of the amount of variability in size and shape of the nuclei and is a major determinant of the histological grade.

ancestry. In P1-D3, a 3-mm length of a large duct exhibited both a genetic and a histological transition from normal ductal epithelium to DCIS along its length, confirming that, although neoplastic involvement was extensive in this lobe, it was incomplete (Extended Data Fig. 7a). On dividing the glandular tissue into lobules (white dashed contours; Fig. 4a), it was apparent that a handful of lobules contained a single clone, but often multiple clones co-occurred. Indeed, we were surprised to observe that the same clones repeatedly co-existed within lobules that spanned centimetres of tissue. These appearances seem at odds with the traditional model of clonal competition in which a fitter clone generates localized monoclonal sweeps (Fig. 4c).

However, at finer, sublobular resolution, complete or near-complete clonal sweeps are the dominant pattern, as exemplified by assaying 146 representative microscopic regions that represent individual or small clusters of intimately related acini and ducts (beige contours; Fig. 4a). The existence of frequent clonal sweeps as inferred by BaSISS (Fig. 4b) was corroborated by LCM–WGS of additional microregions (Extended Data Fig. 7b). In some instances, including P1-D1-88 (Extended Data Fig. 7c) and P1-D2-0 (Fig. 4a,b,d and Extended Data Fig. 7d–f), clonal interfaces are directly observed within a continuous anatomical space. However, more commonly, rapid clone field transitions (see interactive maps (https://www.cancerclonemaps.org/))

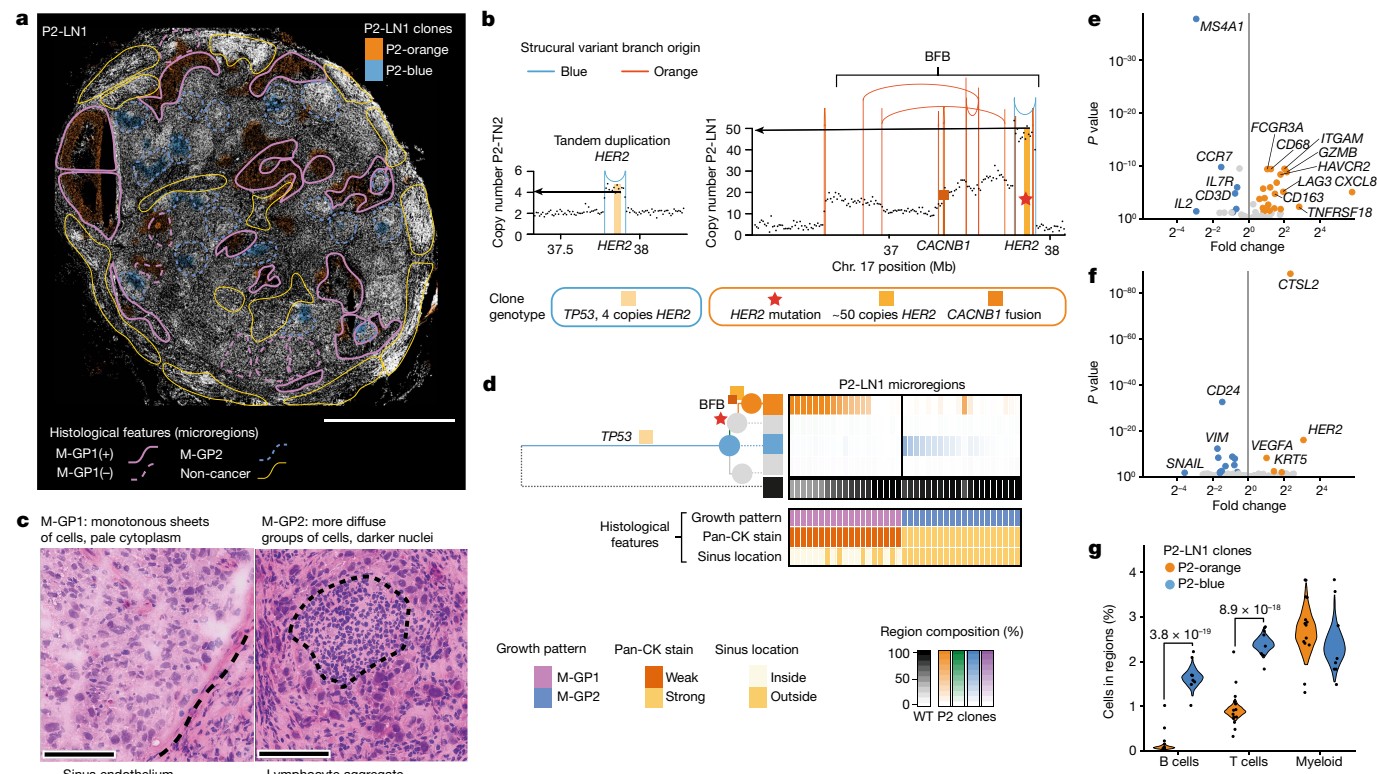

**Fig. 5 | Intrinsic and extrinsic features of metastatic subclones in a lymph node. a**, BaSISS map of P2-LN1, which relates to P2-TN1 (Fig. 3e) and P2-TN2 (Extended Data Fig. 6a,b). The most prevalent genetic clone colours are projected as coloured fields on the DAPI image (reported if the CCF is more than 25%; a threshold of 5% is used in regions of diffusely infiltrating blue to allow visualization in very high normal contamination regions). Scale bar, 2.5 mm. Coloured contours define microregions with distinct metastatic cancer growth patterns (M-GP1 and M-GP2); '+' indicates the surrounding sinus epithelium. **b**, Plots of the genomic structures in P2-blue and P2-orange clones in the vicinity of the *HER2* gene, derived from WGS data of P2-TN2 and P2-LN1. Vertical lines represent genomic rearrangement breakpoints coloured by the phylogenetic tree branch where the event occurred. Dots represent local (binned) copy number. *HER2* amplification, *CACNB1* fusion and *HER2* mutation are BaSISS targets used to track this complex event. BFB, breakage fusion bridge. **c**, Representative areas of the two main growth patterns stained with

H&E. Scale bar, 100 μm. **d**, Phylogenetic tree inferred from P2 multiregion WGS. Branch and node colours inform the clones mapped in **a**. The top heatmap reports the BaSISS clone contribution to 39 histologically annotated microregions from **a** (regions with 5% or more tumour cells are included); see https://www.cancerclonemaps.org/. The bottom heatmap reports microregion histological features. Pan-CK, pan-cytokeratin. **e**, Volcano plot of immune cell expression of the 62 genes in the ISS immune panel. **f**, Volcano plot of epithelial cell expression of the 91 genes in the ISS immune panel. Significantly (FDR > 0.1), differentially expressed (fold change of more than 1.5 both ways) genes are coloured. **g**, Violin plots depict clone-specific cell-type contribution posterior density of the generalized linear mixed model with region-specific random effect, and includes the 22 clone territories with a dominant clone fraction of more than 0.05 in P2-LN1. Significant comparisons were controlled for FDR using the Benjamini–Hochberg procedure.

coincided with the myoepithelial cell layer and/or basement membrane that define an acinus or ductule border. It thus transpires that the microanatomical structure of resident tissues can have, an as yet poorly understood, role in shaping observed subclonal architectures (Fig. 4a,c).

## DCIS clone-specific phenotypes

Integration of histological and spatial gene expression data from serial sections revealed that the DCIS clones, P1-green and P1-orange, exhibit many phenotypic differences that are consistent across large tissue areas (Fig. 4d,e and Extended Data Figs. 7e,f and 8a,b). Histogenetic associations were very strong, with regions dominated by P1-green being more likely to have an intermediate rather than a low nuclear grade ($P < 0.0001$; Fisher's exact test after Bonferroni correction), exhibit more nuclear pleomorphism ($P < 0.0001$), necrosis ($P < 0.0001$), vacuoles ($P < 0.0001$) and a non-solid architectural growth pattern ($P < 0.0001$) (Fig. 4d,e and Extended Data Fig. 7e,f).

Clone and cell type-resolved spatial gene expression analysis using targeted ISS further corroborated phenotype–genotype correlations. A total of 28 of 91 interrogated genes were differentially expressed by

the two main clones (FDR < 0.1, fold change > 1.5 both ways; Extended Data Fig. 8a,b). Consistent with a higher nuclear grade, P1-orange epithelial cells exhibited higher expression of the cell-cycle regulatory oncogenes *CCND1* and *CCNB1* and the oncogene *ZNF703*, which have been linked to adverse clinical outcome[40]. Overall, architectural and nuclear appearances and gene expression profiles were remarkably lineage-specific, and it was particularly notable that these different patterns could also be appreciated spatially, in regions with sublobular, microscopic clone intermixing, adding weight to the clone composition predictions by the model (Extended Data Fig. 7d).

## Metastatic clones in a lymph node

Lymph node metastasis is associated with higher rates of cancer mortality[41]. Whether it has an active role in facilitating cancer progression or simply reflects a more aggressive or distinct biology of certain clones is largely unknown. A substantial challenge is low cancer purity of diffusely infiltrated lymph nodes, which can make it difficult to separate cancer from immune cell-derived molecular signals. To demonstrate that BaSISS can facilitate the simultaneous study of cancer and immune compartments in such challenging cases, we analysed BaSISS,

histological annotation and ISS targeted gene expression datasets from sample P2-LN1 (Fig. 5 and Extended Data Fig. 9).

BaSISS in P2-LN1 targeted 13 trunk and branch alleles, including point mutations and an expressed novel internal fusion in the *CACNB1* gene that was co-amplified with the clinically targetable breast cancer oncogene *HER2* in a breakage fusion bridge event (Fig. 5b and Supplementary Data Table 1). The model detected two clones (P2-blue and P2-orange) that formed spatially segregated patterns in P2-LN1 (Fig. 5a,d). Only P2-blue was detected in primary breast tumours (P2-TN1 and P2-TN2) (Fig. 3e and Extended Data Fig. 6b).

Detailed histological annotation, blinded to the clone territories, was performed using a combination of H&E, CD45 and pan-cytokeratin IHC and identified multiple metastatic cancer growth patterns (coloured contours; Fig. 5a,c,d and Supplementary Table 2). Intersecting the clone maps and histological annotations revealed strong associations between the two detected clones and the two main histological growth patterns (*P* < 0.0001, Fisher's exact test) (Fig. 5d). The P2-orange clone formed monotonous sheets of cancer cells, exhibited weak immunoreactivity for pan-cytokeratin and often occupied sinusoidal structures. By contrast, P2-blue cells stained more strongly for pan-cytokeratin and, when clustered, surround densely packed lymphocyte cores (Fig. 5c,d and Extended Data Fig. 9a–d).

We sought to determine whether transcriptional differences support the spatial inference of clones. Consistent with the known *HER2* amplification, P2-orange expressed higher levels of *HER2* (Fig. 5f and Extended Data Fig. 9c). A total of 17 of 91 genes were differentially expressed and many of these are implicated in critical biological cancer pathways and/or have recognized prognostic value, including *CTSL2*, *VEGFA* (encoding vascular endothelial growth factor receptor A) and *CD24* (refs. [42,43]) (Fig. 5f). Spatially plotting these genes confirmed that clone-specific expression patterns are recapitulated within multiple, spatially distinct expansions across more than 1 cm$^2$ of tissue (Extended Data Fig. 9a–c).

Integration of spatial transcriptomics data also revealed that metastatic subclones occupied distinct immune microenvironments. Relative to P2-orange cells, P2-blue cells resided in neighbourhoods enriched for T cells and B cells (Fig. 5e,g). In fact, P2-blue cells frequently formed clusters around B cell-rich germinal-like centres, highlighting a potential clone-specific interaction with the adaptive immune system (Fig. 5c and Extended Data Fig. 9a,d). By contrast, P2-orange regions frequently resided inside the lymph node sinuses that were lined by endothelial cells expressing *CD34* and *PDGFRB* (Fig. 5c and Extended Data Fig. 9f). Most of the immune cells in P2-orange regions were myeloid cells with expression profiles consistent with the existence of both M1 and M2 macrophages (*CD163*, *CD68*, *HAVCR2* and *FCGR3A*), and the most highly enriched gene, *CXCL8*, is released by hypoxic macrophages[44] (Fig. 5e). Indeed, relative to P2-blue, it emerges that P2-orange experienced more hypoxic conditions manifesting as higher cancer cell expression of *VEGFA* and necrotic regions (Extended Data Fig. 9e,f). Hypoxia signatures are associated with adverse clinical outcomes, probably because they reflect the emergence of environments that can select for hypoxia-tolerant clones and/or cancer proliferation rates outstrip neoangiogenesis[45]. Together, these data demonstrate how BaSISS clone maps allow one to spatially relate such variation in microenvironments to individual clones.

## Discussion

Here we present BaSISS, a pipeline that combines a highly multiplexed fluorescence microscopy-based protocol and algorithms to map and phenotypically characterize the unique set of subclones of cancer. These maps served as the basis for further spatially and single-cell-resolved molecular and histological characterization of each clone. Applying BaSISS to a series of samples from the key stages of breast cancer progression—carcinoma in situ, invasive cancer and lymph node metastasis—it is notable that virtually every sample exhibited a spatial organization of clones, which warrants further investigation in larger cohorts. The fact that nearly all clones examined in this dataset displayed distinct clone-specific gene expression, stromal and immune microenvironments and microanatomical niches highlights the functional relevance of at least some subclonal diversification.

The ability to chart clonal growth patterns and clone-specific genetic underpinnings of the tumour microenvironment is likely to be instrumental in elucidating how different evolutionary processes operate and manifest across different cancer types—or even in histologically normal tissues[46]. Understanding the forces of malignant progression, especially invasion and metastasis, and how interactions with the tumour microenvironment shape clinical outcomes[10] appear of particular importance. Detailing the functional and microenvironmental characteristics of different clones is also relevant as some part of subclonal diversity in tumours may be due to selectively neutral drift, but the exact extent remains debated.

Particular advantages of the technology are that it is capable of interrogating very large tissue sections on the scale of squared centimetres, which enables studying entire cross-sections of smaller tumours. It is also comparably cheap, unlike solely relying on sequencing-based methods[47]. The three main limitations of the approach are relatively low sensitivity, which currently precludes single-cell genotyping, a reliance on RNA with the resulting variation in gene expression levels of targeted transcripts, and the fact that clone-defining mutations need to be detected first by separate sequencing-based assays. Greater sensitivity and spatial resolution may be achieved by including more targets per clone and by favouring mutations with higher predicted expression levels, for example, in higher copy number states. A switch to hybridization-based sequencing and direct RNA-binding probes may also improve base-specific detection by several fold[48,49]. Further discussion of the implications of our observations and limitations of the method is provided in a Supplementary Note.

It is often stated that "nothing in biology makes sense except in the light of evolution"[50], which is likely to be true for cancer biology. The ability to spatially locate and molecularly characterize different cancer subclones adds essential features to the spatial-omics toolkit. It provides a robust evolutionary framework that is necessary to interpret the biological relevance of many of the more plastic spatial characteristics of a cancer. Future widespread applications of spatial genomics approaches such as BaSISS will uncover how cancers grow in different tissues and allow us to track, trace and characterize the ill-fated clones that are responsible for adverse clinical outcomes.

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

## Reporting summary

Further information on research design is available in the Nature Portfolio Reporting Summary linked to this article.

## Data availability

Complete BaSISS and ISS datasets that are necessary to interpret, verify and extend the research in the article are available to download (ftp://ftp.sanger.ac.uk/pub/cancer/LomakinEtAl_BaSISS). Bulk tissue WGS data are deposited in the European Genome Phenome Archive and are available for download on request (https://ega-archive.org/datasets) with the following accessions: EGAD00001002696 (P2 samples, with IDs PD14780a, PD14780b, PD14780d and PD14780e) and EGAD00001000898 (P1 samples, with IDs PD9694a, PD9694b, PD9694c and PD9694d). Registered fluorescent microscopy images from ISS experiments have been deposited at BioImage Archive (https://www.ebi.ac.uk/bioimage-archive/) under accession number S-BIAD537. Public data used for single-cell RNA sequencing analysis were obtained from the NCBI's Gene Expression Omnibus (https://www.ncbi.nlm.nih.gov/geo/query/acc.cgi?acc=GSE176078). Source data are provided with this paper.

## Code availability

All scripts and custom code for data analysis, including step-by-step notebooks, are available at https://github.com/gerstung-lab/BaSISS and https://doi.org/10.5281/zenodo.703731. Code used to segment nuclei in images is available at https://github.com/yozhikoff/segmentation.

**Acknowledgements** This work was funded in part by the Wellcome Trust [Grant number 108413/A/15/D]. For the purpose of open access, the authors have applied a CC-BY public copyright licence to any Author Accepted Manuscript version arising from this submission.

L.R.Y. is funded by a Wellcome Trust Clinical Research Career Development Fellowship (ref.: 214584/Z/18/Z). This work was supported by a pump-priming award from the Cancer Research UK Cambridge Centre Early Detection Programme (CRUK grant ref.: A25117). M.N. is funded by the Swedish Research Council (project grant 2019-01238), Cancerfonden (project grant CAN 2021/1726) and the strategic research area U-CAN. C.S. is funded by a fellowship from the Swedish Cancer Society (210401FE) and a research grant from the Cancer Society in Stockholm/the King Gustaf V Jubilee Foundation (rafo 174292). S.S. is supported by NCI U54 CA225088 and the Ludwig Center at Harvard. H.R. is funded by the Regional Health Authorities South-East (project grant 2019057). ISS technical assistance was provided by the In Situ Sequencing unit, part of the spatial and single-cell biology platform, funded by Science for Life Laboratory, Stockholm, Sweden. Samples and clinical data used in this study were provided by Dana Farber Cancer Institute under support of DF/HCC Breast SPORE: Specialized Program of Research Excellence (SPORE), an NCI-funded programme, grant 1P50CA168504. The content of this publication is solely the responsibility of the authors and does not necessarily represent the official views of the NIH/NCI.

**Author contributions** J.S., P.J.C., M.N. and L.R.Y. designed the initial study. A.L., M.Gerstung and L.R.Y. analysed and interpreted the data and drafted the article and figures. J.S., M.N. and C.S. acquired the ISS data and contributed to data interpretation and manuscript preparation. A.L. and M. Gerstung developed the core mathematical models. A.S. and V.K. contributed to mathematical modelling. A.L. and G.R. developed the dedicated webtool interface. A.S. contributed to image segmentation and processing. A.L.R. and S.S. provided samples. S.S. and C.S. performed IHC. H.R., S.S., A.L.R. and S.P. contributed histopathological expertise. Y.S.J. contributed RNA sequencing expertise. S.D. performed WGS subclonality analysis. J.S.P., V.V., T.L., O.A.B. and M.Gataric contributed to the development of bespoke ISS analysis pipelines. All authors reviewed and commented on the manuscript.

**Competing interests** J.S. is now (but was not at the time of contribution to this manuscript) an employee of Spatial Transcriptomics, Part of 10X Genomics, Inc. Y.S.J. is co-founder of Genome Insight. C.S. is co-owner of HistoOne AB, and has research contracts with Prelude Dx. M.N. is an advisor to 10X Genomics. All other authors declare no competing interests.

**Additional information**

**Correspondence and requests for materials** should be addressed to Moritz Gerstung, Mats Nilsson or Lucy R. Yates.

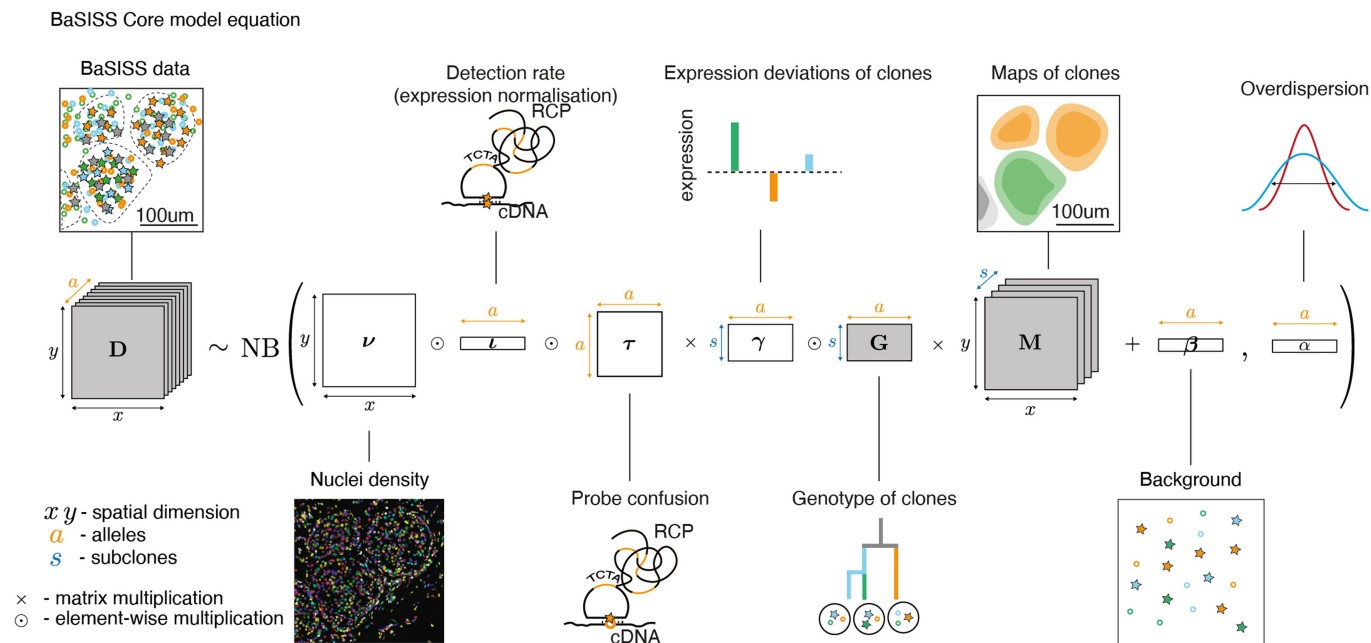

BaSISS Core model equation

BaSISS data

Detection rate (expression normalisation)

Expression deviations of clones

Maps of clones

Overdispersion

Nuclei density

Probe confusion

Genotype of clones

Background

$x\ y$ - spatial dimension
$a$ - alleles
$s$ - subclones

× - matrix multiplication
⊙ - element-wise multiplication

**Extended Data Fig. 1 | Mathematical modelling BaSISS data.** Mathematical model for generating quantitative clone maps. The essential idea is that the BaSISS signals count matrix D is decomposed into maps of clones M each with a distinct genotype G (grey shading), accounting for multiple sources of variability. For further details see Supplementary Methods.

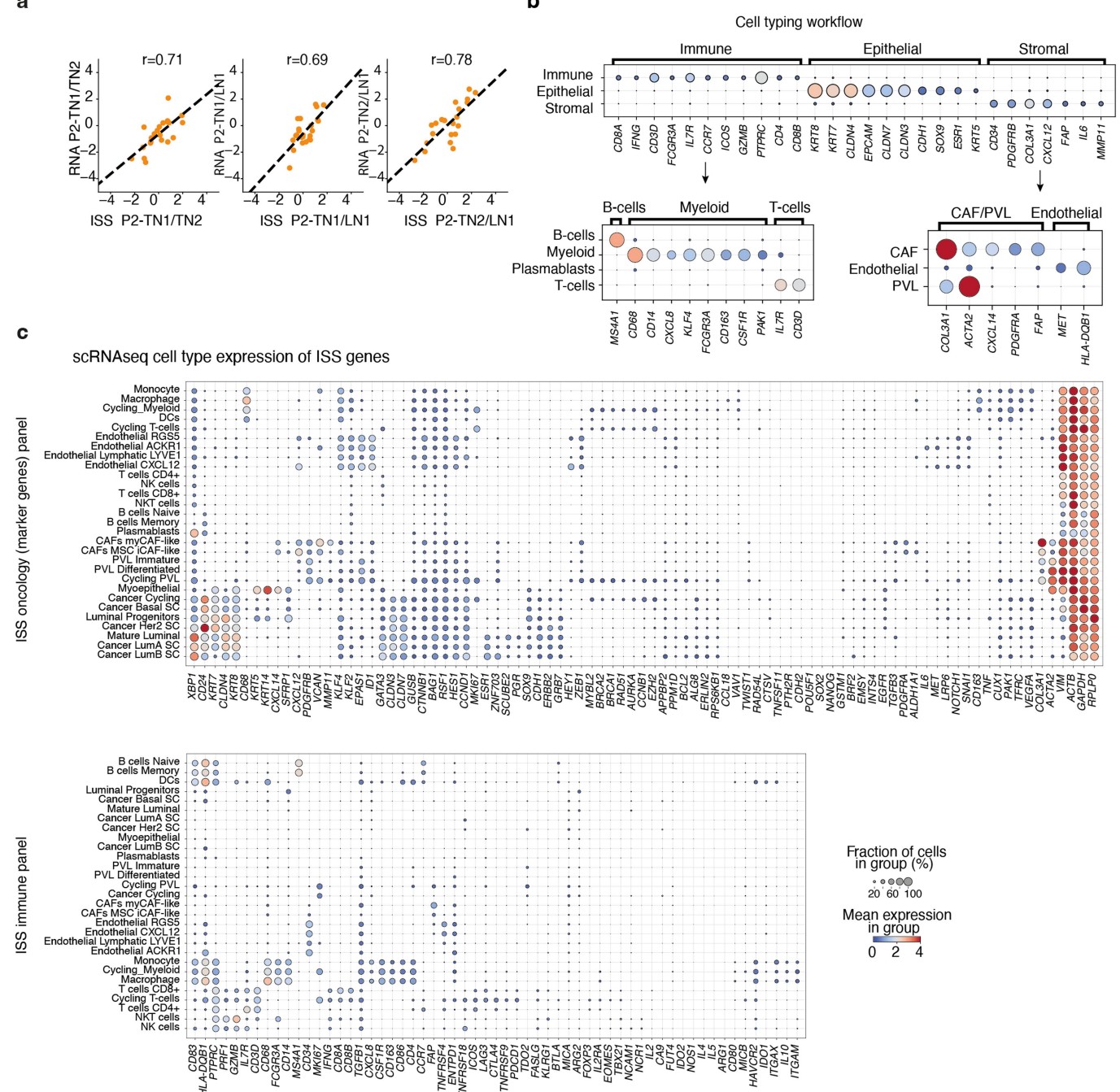

**Extended Data Fig. 2 | Hierarchical cellular typing workflow. a**, Scatterplots of between sample log$_2$-fold change of gene expression derived from RNAseq and combined ISS oncology and immune experiments. Correlations indicate that the probes are on-target. Included genes are those with transcripts per million (TPM) > 25 in RNAseq and 1000 detections per million cells in ISS whose deviation due to low counts would be negligible, R = Pearson's correlation coefficient. **b**, Marker genes for the cell typing were selected using hierarchical logistic regression. The input datasets are the targeted ISS oncology and immune panels. If nuclei have marker ISS signals within 5 µm from their centre, the corresponding cell types were assigned. At first iteration, nuclei were classified into 3 broad categories (Immune, Epithelial and Stromal). At the second iteration, nuclei with Immune and Stromal assignments were further subdivided into (B-cells, Myeloid and T-cells) and (CAF/PVL, Endothelial) groups. The identity of nuclei that did not have any marker genes in proximity or had a contradictory assignment was considered unknown. PVL = perivascular-like. **c**, Mean expression of the genes used in ISS immune and oncology panels was calculated from the breast cancer single cell RNA sequencing (scRNA) reference (derived from Wu et al. Nature Genetics, 2021) to aid interpretation of the observed ISS signal distribution.

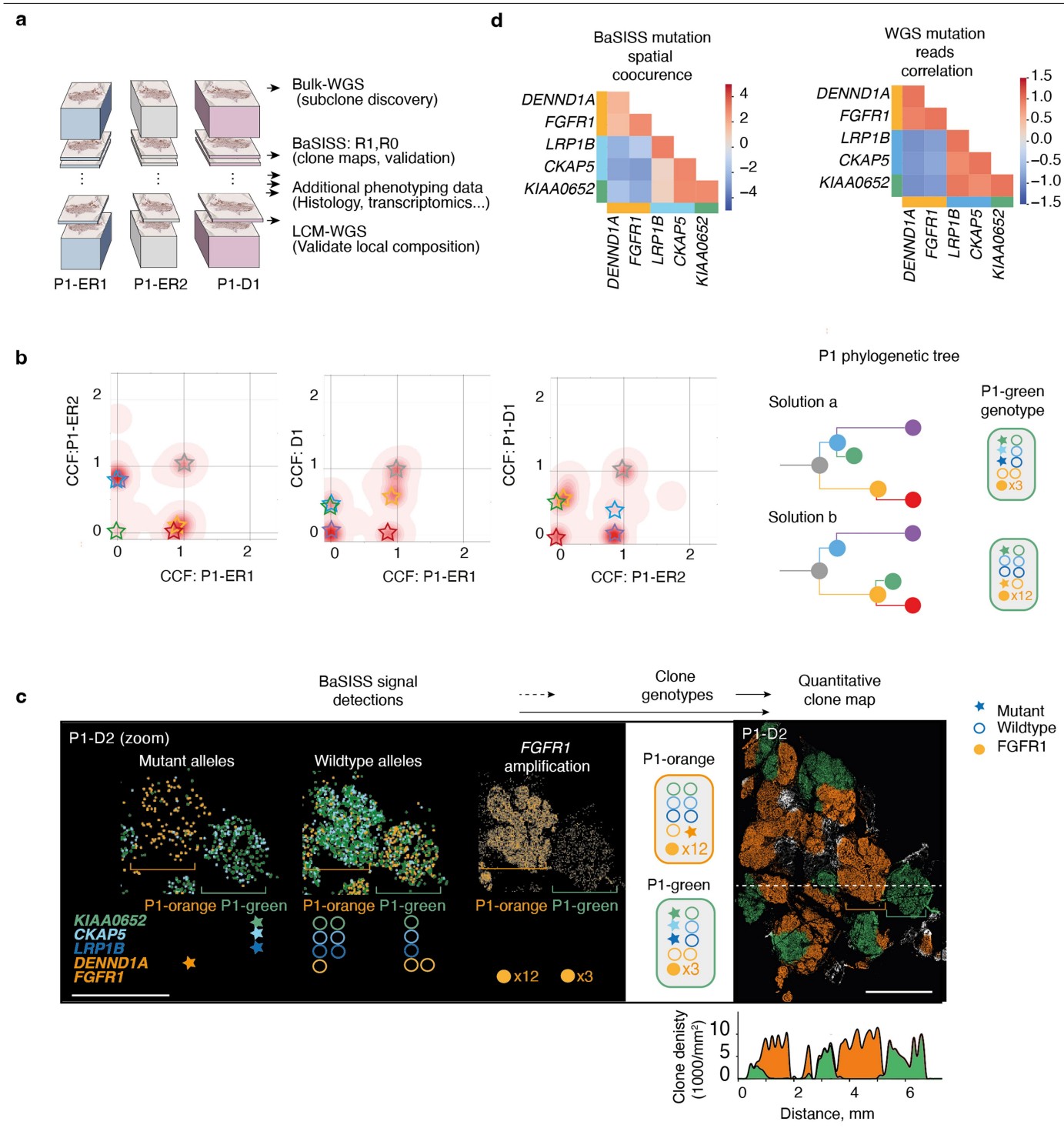

**Extended Data Fig. 3 | BaSISS resolves ambiguous WGS derived clonal architectures. a**, Cartoon illustrating tissue block handling for P1. **b**, Density plots of WGS derived point mutation, cancer cell fraction (CCF) estimates from pairs of samples (see Supplementary Methods for details). Mutation clusters are denoted by coloured stars. The two phylogenetic tree solutions most compatible with the mutation cluster CCFs are presented alongside their respective inferred P1-green genotypes. **c**, BaSISS signal detections in approximately 3 mm² region of D2 (repeated x3, left), exhibit co-occurrence

and segregation patterns of both wild-type and mutant alleles that support the phylogenetic tree solution 'a'. Sample D2 clone map (right) with frequency plot (below) of local, mean clone composition, corresponding to horizontal dashed line. The quantitative, continuous nature of these data can be examined more fully via the interactive web browser https://www.cancerclonemaps.org/. **d**, Spatial co-occurrence matrix of BaSISS mutant allele signals from P1-D1 (top) and LCM-WGS read correlations from 6 microdissected regions of P1-D1/P1-D2 reveal the same co-occurrence patterns that support tree solution 'a'.

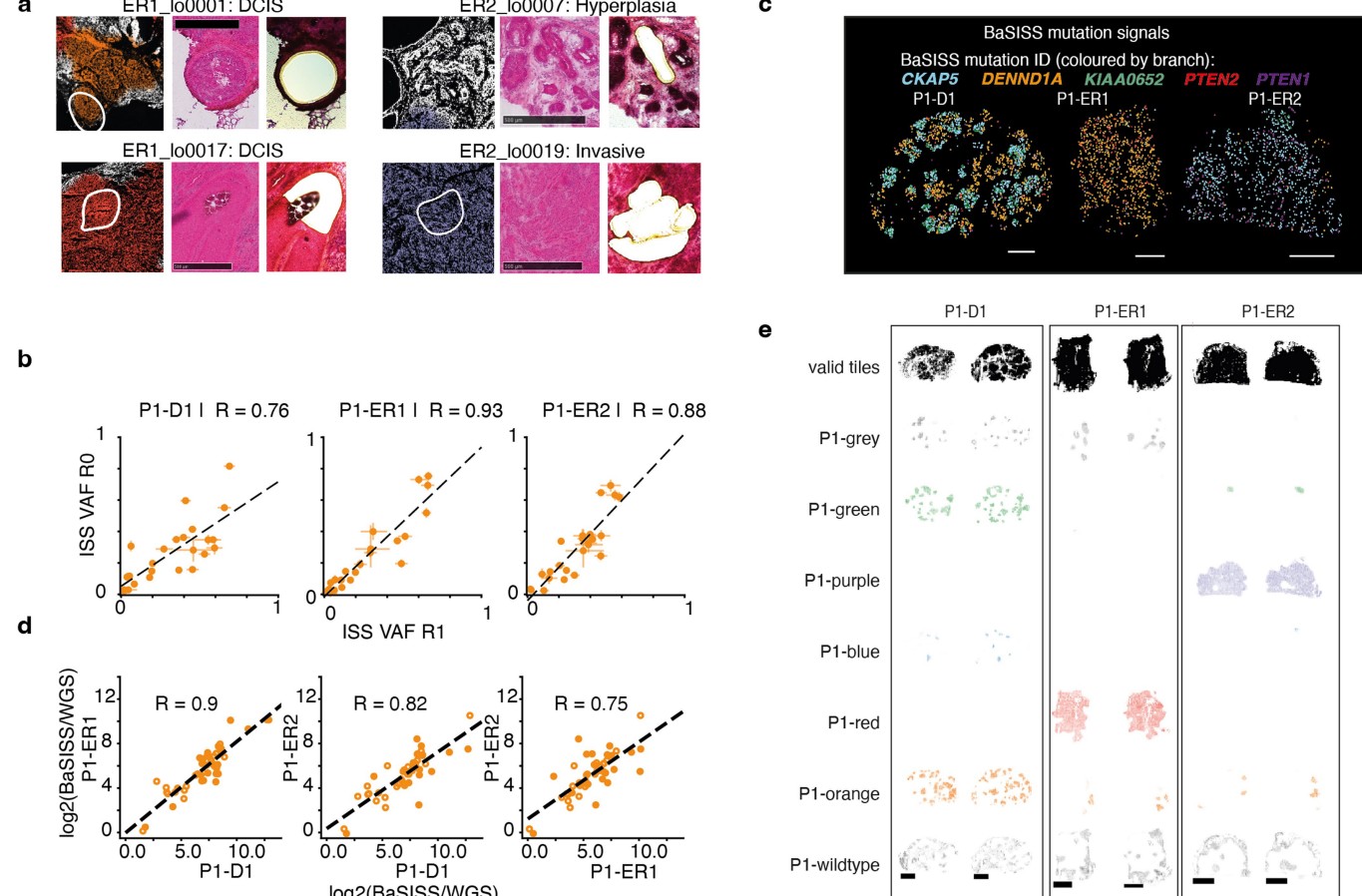

**Extended Data Fig. 4 | Validation of the BaSISS workflow. a**, Four examples of BaSISS clone map regions (left) (see Fig. 2e) selected for laser capture microdissection (LCM) and whole genome sequencing (WGS) validation. Corresponding regions in z-stack tissue sections stained with H&E before (middle) and after (right) LCM. Scale bar = 500 um. **b**, Scatterplots of BaSISS variant allele fractions (VAF) defined as the number of mutation specific signals divided by mutation plus wildtype signals (depth) for each mutation target between replicate BaSISS experiments. Data are presented as mean estimates and 95% HPDI. **c**, Replicate BaSISS experiments (relates to Fig. 2d). Signals for selected mutations are coloured according to branch of origin. Scale bar = 2.5 mm. **d**, Scatterplots of BaSISS VAFs (normalised to WGS VAFs) in related samples indicate that the BaSISS data provide a meaningful read out of genomic structure. R = Pearson's correlation coefficient. **e**, BaSISS clone fields derived from replicate sequencing data: Factor 2–7 are clones corresponding to the same coloured branch. Factor 1 is residual, 8 is normal. Scale bar = 2.5 mm.

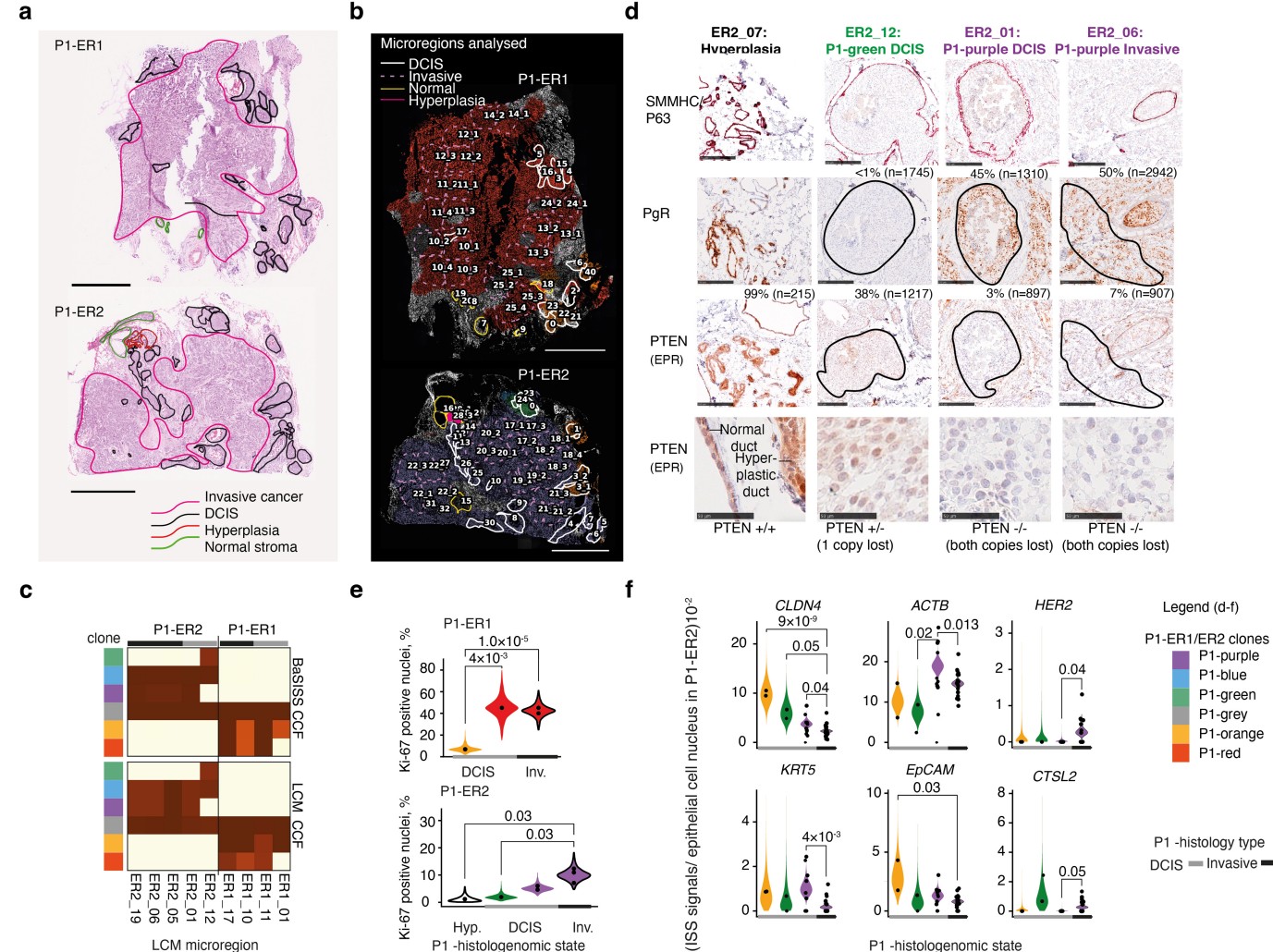

**Extended Data Fig. 5 | Phenotype characterisation of histo-genomic states in sample P1 PBCs. a**, Broadly annotated H&E tissue sections of the P1-ER1 and P1-ER2 primary breast cancers. **b**, Microregions selected for detailed analysis overlaid on BaSISS maps (regions relate to heatmaps in Fig. 3a; numbers relate to histological annotations in Supplementary Table 2). **c**, Comparison of the cancer cell fractions (CCF) of 9 regions of P1-ER1/P1-ER2 determined through both BaSISS (top) and laser capture microdissection (LCM) whole genome sequencing (WGS) (bottom). **d**, Snapshots of immunohistochemistry (IHC) staining in serial fresh frozen tissue cryosections from P1-ER2. Selected regions with confirmed clone compositions (by LCM WGS) are presented. SMMHC/P63 antibody stains myoepithelial cells red, PTEN protein and the progesterone receptor (PR) stain brown. % reports proportion of positive nuclei stained, n reports number of nuclei in region assessed by QuPath digital software. Row 1–3 scale bars = 250 um. Row 4 scale bar = 50 um. **e**, Violin plots depict clone specific Ki67 IHC staining rate posterior density of the generalised linear mixed model (glmm) with region specific random effect. Significant comparisons were controlled for FDR using the BH procedure. Analysis was limited to the 11 regions with confirmed clone compositions by WGS due to variation between IHC and BaSISS sections in z-stack morphology (relates to Fig. 2d). **f**, Violin plots depict clone specific gene expression contribution posterior density of the glmm with region specific random effect. A total of 36 regions of P1-ER2 with a dominant clone fraction > 0.7 were analysed. Significant comparisons were controlled for FDR using the BH procedure. DCIS = Ductal carcinoma in situ.

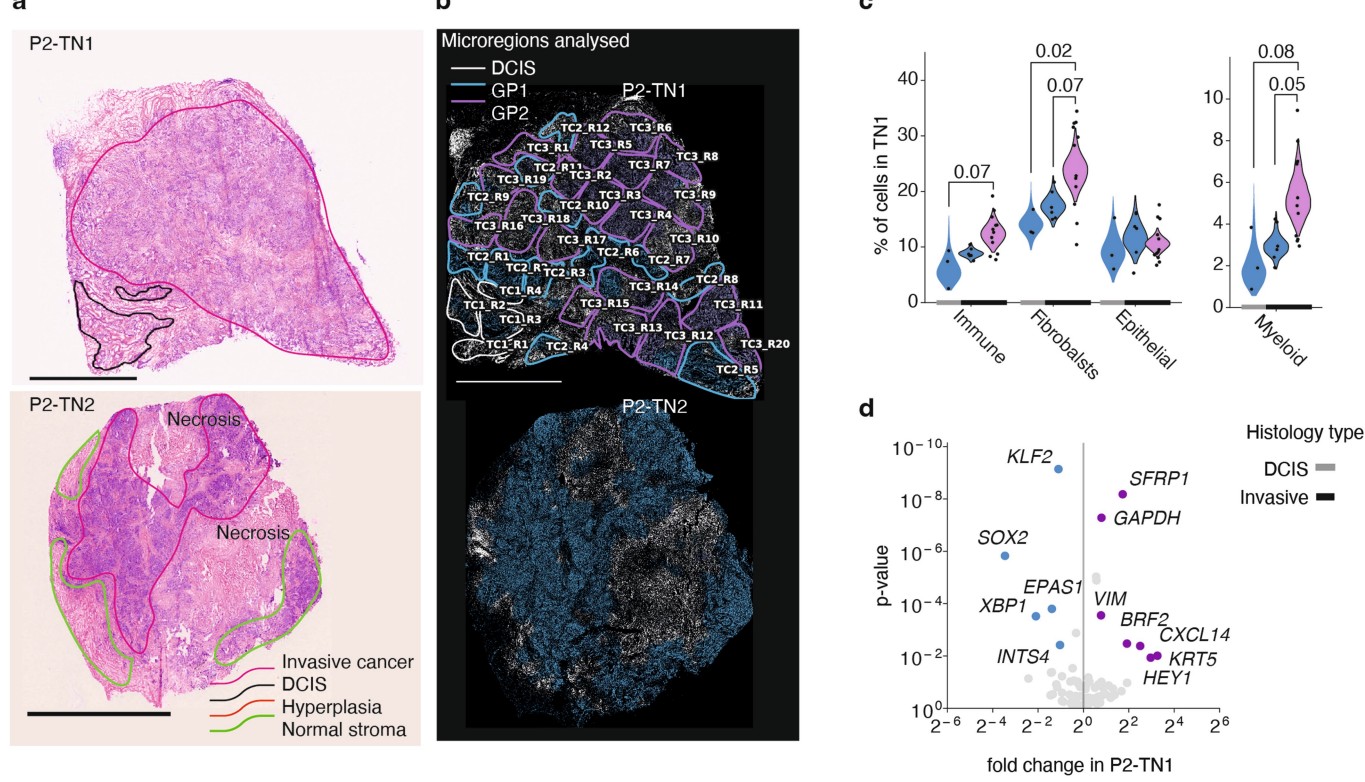

**Extended Data Fig. 6 | Ecosystem characterisation in P2-TN1.**
**a**, Haematoxylin and eosin (H&E) stained sections of the two primary breast cancers from case P2. **b**, Microregions selected for detailed analysis overlaid on BaSISS maps (regions relate to heatmaps in Fig. 3a; numbers relate to histological annotations in Supplementary Table 2). Microregions were not defined for P2-TN2 as a single clone was targeted and detected. **c**, Cell type contribution posterior density of the generalised linear mixed models (glmm) model with region specific random effect. Significant comparisons were

controlled for FDR using the BH procedure. 19 clone territories (with dominant clone fraction > 0.1) were analysed. Fibroblasts and perivascular-like cells (PVL) could not be differentiated within this experiment and are reported as 'fibroblasts'. **d**, Volcano plot of epithelial expression of the 91 oncology ISS panel genes in TN1 invasive regions. Significance was adjusted for multiple testing using BH procedure, only genes with FDR < 0.1 and fold change > 1.5 in both ways are coloured/labelled. DCIS = Ductal carcinoma in situ.

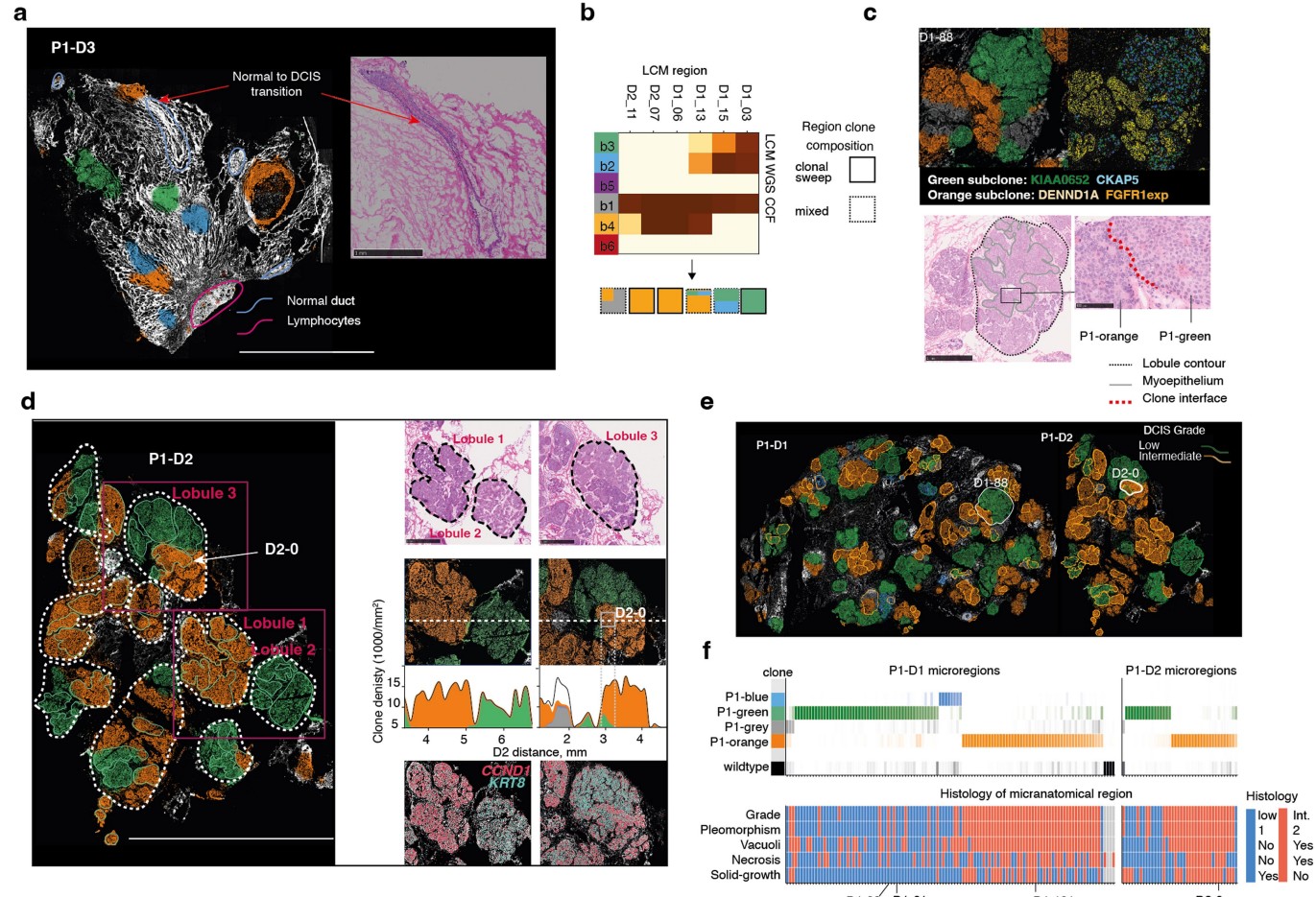

**Extended Data Fig. 7 | DCIS clone specific histologies. a**, BaSISS clone map of P1-D3, a sample that contains Ductal carcinoma in Situ (DCIS), stroma and normal glandular regions. The most prevalent genetic clone colour is projected as a coloured field on DAPI images (reported if cancer cell fraction > 25% and inferred local cell density > 300 cells/mm²). Scale bar = 5 mm. Inlaid, H&E stained image (from a serial tissue section) details the histological transition from normal to DCIS morphology, consistent with the clone field transition in the BASISS map (scale bar = 1 mm). **b**, Heatmap of cancer cell fractions (CCF) derived from LCM WGS of six regions of P1-D1/P1-D2 with cartoon of predicted clone composition indicating inference of monoclonal and polyclonal growth patterns. **c**, Example of a clone interface within a single sub-lobular space in P1-D1. Clone fields (top left); spatial BaSISS mutation signals (top right);

characteristic histological features on H&E (bottom left) with zoom image of clone interface (scale bar = 100 um) (bottom right). **d**, Histological, genetic and transcriptional features of three lobules (identified on the clone map of P1-D2; left, scale bar = 5 mm) are shown: H&E staining (top) scale bar = 1 mm; BaSISS clone fields projected on DAPI with frequency plots of the local, mean cancer (coloured areas) and non-cancer (white) corresponding to horizontal dashed line (middle); and ISS gene expression signals reporting *CCND1* and *KRT8* that exhibit clone specific spatial patterns. **e**, Clone maps of P1-D1/P1-D2 (as presented in Fig. 4a) but microregions are coloured according to histological grade. **f**, Histopathological annotations for each microregion presented alongside the same clone composition heatmap as shown in Fig. 4b.

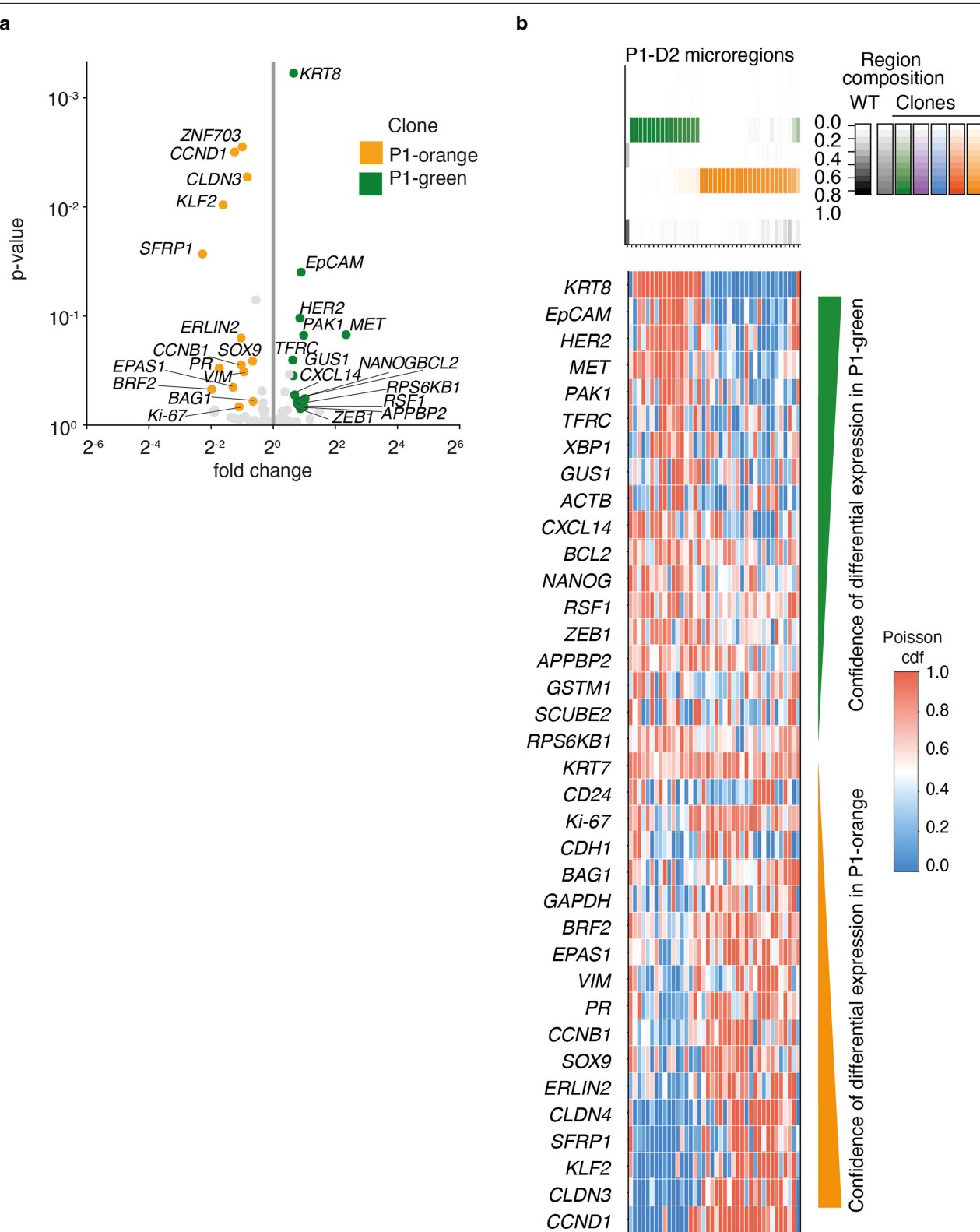

**Extended Data Fig. 8 | Distinct transcriptional profiles of two DCIS clones.**
**a**, Volcano plot of epithelial expression of the 91 oncology ISS panel genes in
P1-D2. Significance was adjusted for multiple testing using BH procedure, only
genes with FDR < 0.1 and fold change > 1.5 in both ways are coloured/labelled.
The coloured genes are included in the by region plot in **b**. **b**, Heatmap of gene
expression data within each of the 41 sampled regions in P1-D2, presented
alongside the relevant clone composition regions (top) as per Fig. 4b. ISS
counts in each regions are transformed by applying Poisson cdf with λ = mean
(P1-green expression, P1-orange expression) × nuclei count in each region, thus
divergence from 0.5 reflects deviation from the global mean expression. Only
genes with FDR < 0.1 are presented and ordered by the confidence of
differential.

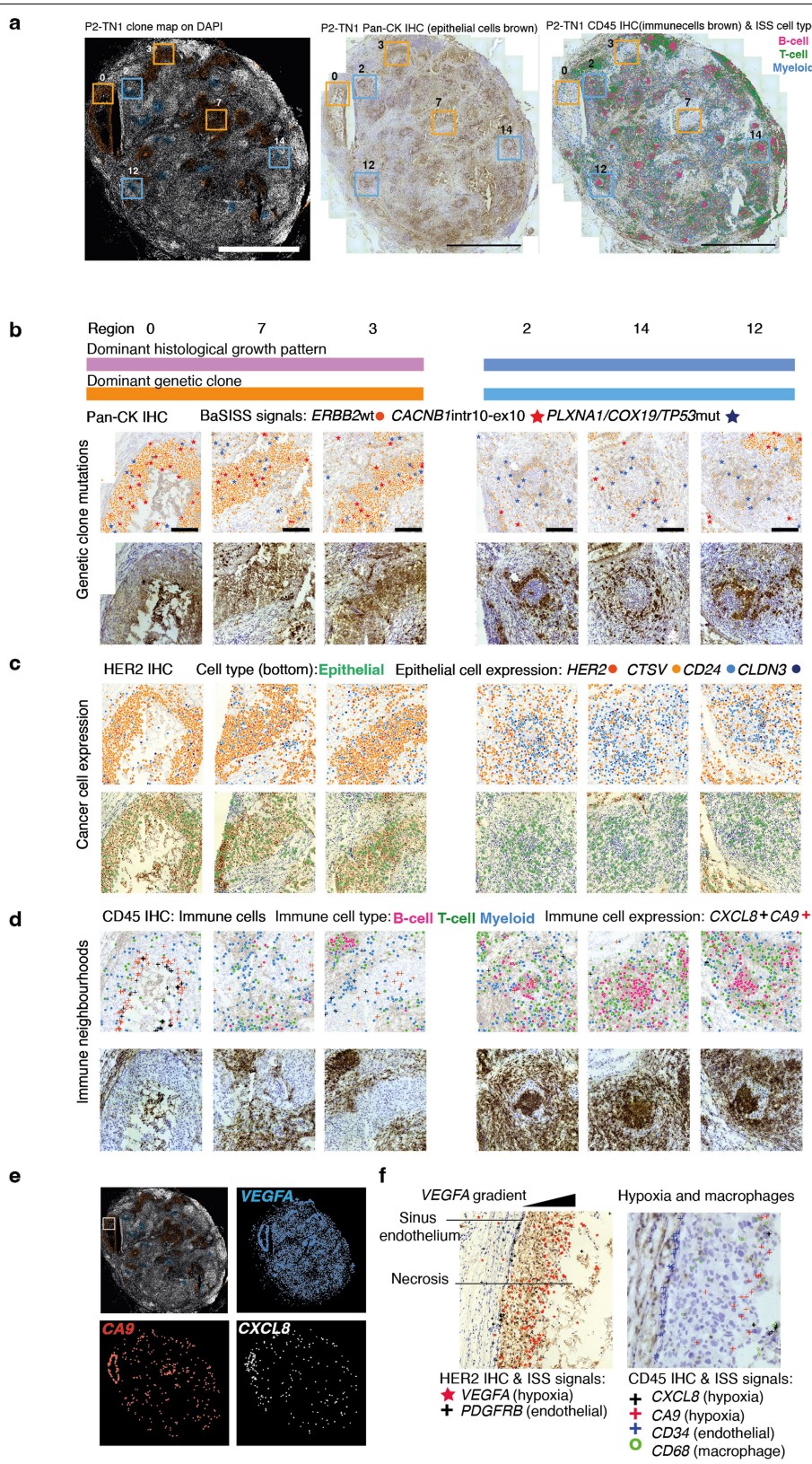

**Extended Data Fig. 9** | See next page for caption.

**Extended Data Fig. 9 | Highly recurrent clone specific ecosystems in a metastatic lymph node. a**, P2-LN1 sample (left) DAPI image with BaSISS subclone fields (as shown in Fig. 5a) and coloured squares mark regions depicted in **b,c,d**; (middle) pan-cytokeratin immunohistochemistry stained (IHC) (epithelial cells appear brown); (right) CD45 antibody (immune cells appear brown) with ISS immune panel derived cell types projected as coloured dots. **b**–**d**, Snapshots of example regions dominated by P2-blue or P2-orange clones, as indicated in **a**. In each case signals (dots) from selected targets in BaSISS **b**, ISS oncology **c** or ISS immune panels **d** are presented overlaid on sections stained by IHC following the BaSISS/ISS experiment. In the bottom row of **c** and top row of **d** inferred epithelial and immune cell types are presented. In top rows of **c** and **d**, 80% transparency is applied to the underlying IHC image to aid visualisation of overlaid dots. **e**, Spatial patterns of three hypoxia related genes are projected on the entire P2-LN1 tissue section. **f**, Spatial patterns of *PDGFRB*, *CD34*, *CD68* and hypoxia related ISS signals overlaid on HER2 (left) and CD45 IHC stained sections(right) correspond to region of white square on top left clone field image in **e**.

# Reporting Summary

## Statistics

For all statistical analyses, confirm that the following items are present in the figure legend, table legend, main text, or Methods section.

| n/a | Confirmed | |
|---|---|---|
| ☐ | ☒ | The exact sample size (*n*) for each experimental group/condition, given as a discrete number and unit of measurement |
| ☐ | ☒ | A statement on whether measurements were taken from distinct samples or whether the same sample was measured repeatedly |
| ☐ | ☒ | The statistical test(s) used AND whether they are one- or two-sided<br>*Only common tests should be described solely by name; describe more complex techniques in the Methods section.* |
| ☐ | ☒ | A description of all covariates tested |
| ☐ | ☒ | A description of any assumptions or corrections, such as tests of normality and adjustment for multiple comparisons |
| ☐ | ☒ | A full description of the statistical parameters including central tendency (e.g. means) or other basic estimates (e.g. regression coefficient) AND variation (e.g. standard deviation) or associated estimates of uncertainty (e.g. confidence intervals) |
| ☐ | ☒ | For null hypothesis testing, the test statistic (e.g. $F$, $t$, $r$) with confidence intervals, effect sizes, degrees of freedom and $P$ value noted<br>*Give P values as exact values whenever suitable.* |
| ☐ | ☒ | For Bayesian analysis, information on the choice of priors and Markov chain Monte Carlo settings |
| ☒ | ☐ | For hierarchical and complex designs, identification of the appropriate level for tests and full reporting of outcomes |
| ☐ | ☒ | Estimates of effect sizes (e.g. Cohen's *d*, Pearson's *r*), indicating how they were calculated |

*Our web collection on statistics for biologists contains articles on many of the points above.*

## Software and code

Policy information about availability of computer code

| Data collection | Subclonal clusters were acquired as reported in previous publications using DPClust https://github.com/Wedge-Oxford/dpclust Version 2.2.8 QuPath software (Version 0.3.0) https://qupath.github.io/<br><br>Large scale probe design was facilitated using an in-house Python (3) software package as described previously (Ref PMID 31740815) which utilizes ClustalW and BLAST+ to ensure probe specificity.<br><br>From the total of 51 image sets, 43 were stitched with Carl-Zeiss ZEN software (version 3.1), and the other 8 failed image sets were stitched using BigStitcher (version 0.9). The registration across imaging cycles was performed in two steps: affine registration on DAPI channel and subsequently local warping on anchor channel. For both steps we used algorithms provided in libraries OpenCV-contrib (version 4.3.0) and scikit-image (version 0.17).<br><br>Analysis of sample P2-LN required IHC signal projection performed on a consecutive slide back to BaSISS slide. To achieve this, we performed a spline-based elastic registration implemented in ImageJ package UnwarpJ(Arganda-Carreras et al. 2008).<br><br>Nuclei segmentation code is available at yozhikoff/segmentation |
|---|---|
| Data analysis | The manuscript used publicly available, open source R and Python libraries/packages for data analysis as described in the methods section. All scripts and custom code for data analysis, including step-by-step notebooks are available at GitHub repository (https://github.com/gerstung-lab/BaSISS) and under the DOI: 10.5281/zenodo.703731<br><br>Software and package version used during the analysis: |

```
Python (3.8.12) with packages:
- numpy (1.22.2)
- pandas (1.1.5)
- scipy (1.7.3)
- opencv (4.5.5)
- matplotlib (3.5.1)
- pymc (4.0.1)
- numpyro (0.10.0)
- scikit-image (0.19.1)
- scikit-learn (1.0.2)
- scanpy (1.8.1)
- shapely (1.8.0)
R (4.1.3) with packages:
- dbmss (2.7-10)
```

For manuscripts utilizing custom algorithms or software that are central to the research but not yet described in published literature, software must be made available to editors and reviewers. We strongly encourage code deposition in a community repository (e.g. GitHub). See the Nature Portfolio guidelines for submitting code & software for further information.

## Data

Policy information about availability of data

All manuscripts must include a data availability statement. This statement should provide the following information, where applicable:

- Accession codes, unique identifiers, or web links for publicly available datasets
- A description of any restrictions on data availability
- For clinical datasets or third party data, please ensure that the statement adheres to our policy

All figures are derived from data that is available for download via ftp://ftp.sanger.ac.uk/pub/cancer/LomakinEtAl_BaSISS
Bulk tissue whole genome sequencing data are deposited in the European Genome Phenome Archive and are available for download on request (EGA, https://ega-archive.org/datasets) with the following accessions: EGAD00001002696 (P2 samples: with IDs PD14780a,b,d,e) and EGAD00001000898 (P1 samples: with IDs PD9694a,b,c,d).

Registered fluorescent microscopy images from in-situ sequencing experiments are deposited at BioImage Archive under the accession number S-BIAD537.

Public data used for single cell RNA-seq analysis were obtained from the NCBI's Gene Expression Omnibus (https://www.ncbi.nlm.nih.gov/geo/query/acc.cgi?acc=GSE176078)

## Human research participants

Policy information about studies involving human research participants and Sex and Gender in Research.

| | |
|---|---|
| Reporting on sex and gender | Both participants reported female sex, gender is not relevant |
| Population characteristics | Participants are female, age 37 and 66 years old at time of diagnosis and surgery for multifocal breast cancers of no special type. No prior cancer diagnosis or treatment was administered. No genotyping was performed on patients. |
| Recruitment | At the time of tissue collection all patients with a diagnosis of primary breast cancer, attending Dana-Farber Cancer Institute were invited to participate in project SHARE. Participants provided written consent for inclusion in the study that entails the donation of tissue from planned clinical procedures (that exceeds pathological diagnostic requirements and would normally be discarded) and clinical data for research purposes. The specific pathological specimens in this analysis were identified by the local pathologist based on the presence of additional histological stages of disease (an involved lymph node or extensive pre-cancerous lesion) and the availability of sufficient tissue blocks to perform the planned experiments. It is feasible that the amount of intra-tumour heterogeneity observed in this study exceeds what would be observed in some smaller primary breast cancers without multiple histological features. |
| Ethics oversight | Samples and data were obtained and managed in line with the declaration of Helsinki under "project SHARE" #93-085, approved by the Dana-Farber Harvard Cancer Center Institutional Review Board. Sample and data handling at the Wellcome Sanger Institute, Cambridgeshire, UK was performed under the wider framework and approval for the Breast Cancer Genome Analyses for the International Cancer Genome Consortium Working Group under REC reference: 09/H0306/36 (Cambridgeshire 3 Research Ethics Committee). The study was later transferred to a protocol REC: 20/PR/0905 (London-Harrow Research Ethics Committee). |

Note that full information on the approval of the study protocol must also be provided in the manuscript.

# Field-specific reporting

Please select the one below that is the best fit for your research. If you are not sure, read the appropriate sections before making your selection.

☒ Life sciences   ☐ Behavioural & social sciences   ☐ Ecological, evolutionary & environmental sciences

For a reference copy of the document with all sections, see nature.com/documents/nr-reporting-summary-flat.pdf

# Life sciences study design

All studies must disclose on these points even when the disclosure is negative.

| | |
|---|---|
| Sample size | This was a biological study, not a clinical trial so we did not undertake a power calculation for the number of patients. A total of 8 tissue samples from 2 donors was considered sufficient to develop and demonstrate the spatial profiling techniques. |
| Data exclusions | All patients and samples analysed were included in the data presented. No samples were excluded. |
| Replication | For samples P1-ER1, P1-ER2, P1-D1 , P2-TN2 a replicate BaSISS experiment was conducted with a slightly altered ISS protocol. Phi29 buffer (Thermo Fisher 10X reaction buffer: 330 mM Tris-acetate (pH 7.9 at 37°C), 100 mM Mg-acetate, 660 mM K-acetate, 1% Tween 20 and 10 mM DTT) and no Exonuclease 1 in the rolling circle amplification step. Results of clone mapping are shown in Extended Data Fig. 5 -  clone field distributions are largely replicated. <br><br>For P1-ER1 and P1-ER2 whole genome sequencing  of laser capture microdissected regions that were selected based upon an ability to identify them in both LCM and BaSISS z-stack sections. All regions that fulfilled this criteria validated the BaSISS model and are shown in Fig.2. For some regions including those in P1-D1 and P1-D2 the tissue structure was such that we could not link regions in the z-plane. Nonetheless, although regional clone compositions could not be validated, the general patterns of mutation/clone co-occurrence and segregation could be confirmed from these data.  Tissue availability precluded LCM in patient P2. |
| Randomization | This was a biological study and not a clinical trial and therefore we did not randomise subjects. |
| Blinding | This was a biological study and not a clinical trial and therefore we did not blind. Histopathological annotation was performed by qualified clinical pathologists without prior knowledge of the spatial genomic data. |

# Reporting for specific materials, systems and methods

We require information from authors about some types of materials, experimental systems and methods used in many studies. Here, indicate whether each material, system or method listed is relevant to your study. If you are not sure if a list item applies to your research, read the appropriate section before selecting a response.

## Materials & experimental systems

| n/a | Involved in the study |
|---|---|
| ☐ | ☒ Antibodies |
| ☒ | ☐ Eukaryotic cell lines |
| ☒ | ☐ Palaeontology and archaeology |
| ☒ | ☐ Animals and other organisms |
| ☒ | ☐ Clinical data |
| ☒ | ☐ Dual use research of concern |

## Methods

| n/a | Involved in the study |
|---|---|
| ☒ | ☐ ChIP-seq |
| ☒ | ☐ Flow cytometry |
| ☒ | ☐ MRI-based neuroimaging |

## Antibodies

| | |
|---|---|
| Antibodies used | panCK, Dako(Agilent), 1:100, Catalogue number:M3515, clone AE1/AE3, LOT: L11139890 <br> CD45, 1:100, Dako(Agilent), Catalogue number:M0701, clone 2B11 + PD7126, LOT:20026786 <br> Her2/ErbB2, 1:50, CellSignaling, Catalogue number:4290T, clone:D8Fl2,  LOT:2 <br> SM-MHC, 1:100, BioCare Medical LLC, Catalogue number:CM 420B, mouse monoclonal, clone: SMMS-1, LOT number unknown <br> P63, 1:150, BioCare Medical LLC,  Catalogue number:CM 163C, mouse monoclonal, clone: BC4A4, LOT number unknown <br> PR, DakoCytomation, 1:75,  Catalogue number:M3569, mouse monoclonal, clone:PgR636, LOT number unknown <br> Ki67, DakoCytomation, 1:400, Catalogue number:M7240, mouse monoclonal, clone: MIB-1, LOT number unknown <br> PTEN, Abcam Anti-PTEN antibody, 1:500, Catalogue number:ab267787, clone:EPR22636-122, LOT number unknown <br> PTEN, Santa Cruz Biotechnologies, 1:300,  Catalogue number:sc-7974, clone:A2B1, LOT number unknown <br> PTEN Cell Signalling, 1:400, Catalogue number:9559, clone:138G6, LOT number unknown <br> PTEN Cell Signalling, 1:500, Catalogue number:9188, clone:D4.3, LOT number unknown <br> ImmPRESS HRP Anti-mouse IgG, VectorLaboratories, Catalogue number:MP-7402-50 ready-to-use, clone not applicable, LOT number unknown <br> ImmPRESS HRP Anti-rabbit IgG, VectorLaboratories, Catalogue number:MP-7401-50, ready-to-use, clone not applicable, LOT number unknown <br> Labelled polymer-HRP anti-mouse (polymer-M): DakoCytomation, Catalogue number: K4007 <br> Poly-AP anti-mouse IgG (poly-AP-M): Leica, Catalogue number: PV6110 |
| Validation | panCK - Approved for in vitro diagnostics by IHC (CE-IVD) according to datasheet. No specific IHC validation reported by manufacturer. Antibody AEl immunoreacts with an antigenic determinant present on most of the subfamily A cytokeratins, including cytokeratins 10, 13, 14, 15 16 and 19. Antibody AE3 reacts with an antigenic determinant shared by the subfamily B cytokeratins including 1, 2, 3, 4, 5, 6, 7 and 8. Please see suppliers datasheet for corresponding references. |

CD45 - Approved for in vitro diagnistics by IHC (CE-IVD) according to datasheet.
Her2/ErbB2 - Recommended for IHC by the supplier; according to datasheet this antibody may cross-react slightly with other overexpressed RTKs. No specific IHC validation reported by manufacturer.

For Her2: no IHC validation reported by the manufacturer; the manufacturer demonstrates that the antibody recognizes a protein band at 185kDa in Western Blot analysis of lysates from Her2 positive human cells (MCF7 and SKBR3).

PTEN IHC was validated using several anti-PTEN antibodies (Cell Signaling Technologies: anti-PTEN (D4.3) Antibody (dilution 1:50) and anti-PTEN (138G6) Antibody; Santa Cruz Biotechnologies anti-PTEN Antibody (A2B1) (dilution 1:300); and Abcam anti-PTEN Antibody [EPR22636-122], dilution 1:500). The Abcam anti-PTEN Antibody [EPR22636-122] displayed the best signal on frozen tissue sections and was used in the study.

SMMHC, P63, Ki-67 and PR antibodies are used by the clinical service at Brigham and Women's Hospital and undergo validation as part of clinical use in Clinical Laboratory Improvement Amendments (CLIA) certified pathology laboratory using control samples (e.g., breast carcinoma, tonsil, etc) as recommended by the suppliers.

