## [Peer Review File · Nature]

Manuscript Title: Spatial genomics maps the structure, nature and evolution of cancer clones

Reviewer Comments & Author Rebuttals

Reviewer Reports on the Initial Version:

Referees' comments:

Referee #1 (Remarks to the Author):

Dear authors,

I have reviewed the work presented by Artem Lomakin and colleagues, where they present a workflow to infer in-situ spatial genomic heterogeneity and apply it to breast cancer samples to identify novel and interesting findings in tumor biopsies, including mapping of the subclonal structure and evolutionary patterns of (pre-)malignant clones, and combine this with spatial gene expression to identify transcriptional and immunological changes over tumor evolution and explain aggressiveness in distant metastasis.

The work is an incremental evolution of existing methodologies to investigate tumor heterogeneity via WGS to model clonal dynamics and ISS to profile spatial gene expression and point mutations, gelled together by an elegant computational method. I believe that the workflow is sufficiently novel and the results presented are also interesting, together defining a new cutting edge for exploiting spatial omics for understanding diseases. The manuscript is well written and mostly easy to understand.

With major revisions, I believe this work would be suitable for publication in a top tier journal. I hope to explain the shortfalls of the submitted work in the following points.

MAJOR

1) What 'exactly' is BaSISS? While this is not major, it is very central to the concept of the paper.
1a) A question I kept asking myself was "how is exactly BaSISS different from ISS?" Base-specific ISS makes it sound like a novel development of the ISS wetlab method. However, prior works by Mats Nilsson (<https://doi.org/10.1038/nmeth.2563>) already demonstrate the use of ISS for detecting "point mutations" (quoted from the abstract) suggesting that ISS already encompassed the concept of BaSISS. Also, the methods section of the submitted work combines BaSISS and ISS into a single sub-section, with no further details of what separates them. The group has also demonstrated its application to investigate tumor heterogeneity (<https://www.oncotarget.com/article/1527/text/>). Given these prior works and no clear difference in the experimental procedure, I feel that the depiction of BaSISS being a new experimental method (figure 1A) is misleading. Are the wetlab methods used to profile point mutations as previously indeed similar/the same as previous works by the group of Mats Nilsson? The authors should explicitly outline the differences between ISS and BaSISS, with the overarching aim of making it clear what BaSISS actually is, also specifically

addressing whether it is a wetlab method for profiling point mutations in transcripts, or is it the entire spatial genomics workflow?

1b) A core part of the workflow is determining which tissues are profiled using WGS to identify which mutations are clonally heterogeneous. The authors depict that this is done through WGS of adjacent slices. However, in the methods section the authors seem to describe this in the context of multi-sample analysis. While this is reasonable, it is conflicting with the idea that the workflow can be meaningfully applied to a single sample as depicted in figure 1A. I feel that it would be extremely difficult to identify meaningful probes from a single patient sample. I am not sure to what extent the current workflow can identify subclonal mutations to target, but if not, then I would suggest fastclone (<https://doi.org/10.1038/s41467-020-18169-2>, https://github.com/GuanLab/FastClone_GuanLab). Unless the authors can demonstrate that demonstrate for the majority of their samples that the workflow would indeed be meaningful having WGS data from only a single sample, then the authors should amend the description of the their workflow, and figure 1A, to explicitly state that the workflow requires multiple samples.

2) Validation of single cell point mutations. The temporal accumulation of point mutations should be validated using targeted single cell DNAseq (e.g. Tapestry) for at least 2 different samples.

3) Validation ISS panels and comparative gene expression using scRNAseq.

3a) Comparisons of mutational clones to gene expression ISS raises interesting hypothesis, but without scRNAseq results the authors cannot claim that e.g. all gene in their “immune” panel originate from immune cells. The authors should perform scRNAseq with at least 2 samples to delineate whether the genes on the immune/cancer panel are indeed unique to immune/cancer cells, or if they are also present in non-immune cells in the tissue being analysed. The results can be presented as a confusion matrix of cell types identified, vs ISS marker genes. Based on the findings, the descriptive text of any ambiguous markers should be revised accordingly.

3b) Single cell RNAseq data could be used to validate the gene expression normalisation of ISS experiments. It is not clear to me how the gene expression obtain from ISS was normalised between different experiments. This should be fairly complex as each individual sample would be a batch, and perhaps even each probe. This makes me a little sceptical about the origins of global gene expression changes, such as those shown in figure 3E - I cannot tell how authors can distinguish between these genes being more expressed in more evolved clones, or if batch effects leads to global differences in gene expression detection sensitivity.

3c) Have the immune and cancer panel been described previously? If not, the authors would need to validate that the ISS probe design are indeed accurately reporting gene expression. This could be done by comparing the ISS results to scRNAseq of the same tissue.

4) Modelling sample fluctuations. The authors model sampling fluctuations using a negative binomial (NB) model, however they do not empirically show that that the data indeed follows a NB distribution. The Authors should indeed show that the ISS/BaSISS data follows a NB distribution or e.g. Marginalised Zero Inflated Poisson, or cite a paper where this has already been empirically shown.

5) Spatial associations between cancer clones and normal tissue. I do not see usefulness of the “frequency plots” in figures 2A, 3B and 4E – they only show a single 1D plane and are not easily

generisable. The authors should attempt to model this quantitatively e.g. as done in figure 7 C,D,E in <https://doi.org/10.1053/j.gastro.2020.11.010>. The functionality has been implemented into SquidPy (<https://doi.org/10.1101/2021.02.19.431994>, https://squidpy.readthedocs.io/en/latest/auto_examples/graph/compute_co_occurrence.html)

6) Does the modelling over prioritise the WGS data? I came to this suspicion because of inconsistencies between figures 1D, in particular for the green clone for the PD9694c clone is clearly visible at the top of Fig 1D third panel, but not present in the bar chart in Figure 1D fourth panel, and also modelled as a earlier blue clone if the cancer clone maps shown in figure 1E. The authors should evaluate whether their computational model has a strong bias to preferentially model the WGS data over the spatial point mutations.

INTERMEDIATE

7) A fundamental novelty of BaSISS is the downstream computational framework. While the authors have made a good effort in providing notebooks and a conda YAML file, however, the code is very unstructured (it seems to be a single undocumented python script called `pile_of_code.py` and a bunch of empty `__init__.py` files). The authors should improve the structure of their code, and improve their documentation to at least match the quality of e.g. novosparc (<https://github.com/rajewsky-lab/novosparc/>, <https://novosparc.readthedocs.io/>), or MutationTimeR.

8) Age of tumors. The authors estimate the age of the tumor clones. In the case of patient PD9694 (age 37), the predicted age of the tumor seems to be nearly 40 and the initial events already being present at the age of 10 (figure 2B). I am not qualified to say whether this is plausible, but this seems to be very early. The authors should look into whether the literature support the idea that tumors can initiate from such an early age. If the authors cannot find support that this could be plausible, it would be prudent not to mislead readers with details of the tumor age in precise years.

9) Figure 2D employs the use of Mann-Whitney U, however the size of the groups is very large and it would be highly appropriate to also report the effect size, e.g., using Cohen's D.

10) The interpretation of cell type markers in the ISS gene paper should be reviewed. The genes used in the panels are fairly well established, however some are a little ambiguous, e.g. TNFRSF18 is a marker for T-regs, but are more highly expressed in NK-cells (<https://www.proteinatlas.org/ENSG00000186891-TNFRSF18/blood>). This would affect the interpretation of the results, such as line 308-314 where the authors describe localisation of TNFRSF18 signal to indicate the presence of T-reg cells.

MINOR

11) Should BaSISS be described as "spatial genomics"? I understand that that mutations being probed stem from the genome, but the spatial analyte is mRNA. I feel that is someone interrogated bulk RNAseq for e.g. recurrent KRAS point mutations, this would still be referred to as transcriptomics. The authors should evaluate between themselves what they believe is the most

accurate description of the work they have done, in relation to how it would have been described in the standards defined by work on bulk sequencing data.

12) Readability of samples IDs and references to clones. The sample IDs do not make the paper easier to follow. Perhaps they can be renamed to something simpler, e.g. PD01-d01, PD1-c01, PD1-m01 (where d, c and m stand for DCIS, cancer and metastasis). Also, I do not like the references to e.g. “the purple clone” in the text, however I cannot suggest anything better off the top of my head.

13) While the conda YAML file is excellent, the authors should be specific about versions of tools in manuscript and/or reporting summary. For example, the version number (e.g. the hash commit or tag) of dpclust is not reported. Similarly, versions of libraries used within e.g. R and python should be reported.

14) Some in text references are to PMIDs.

15) Line 54-55 “they do not perform the critical function of isolating genetic subclones in tissue context because gene expression profiles are highly plastic”. Can the authors support this with evidence that isolating genetic subclones using transcriptional readouts is not possible? Otherwise the sentence can end after “context”, and the link of genetic subclones to transcriptional subclones can proceed this, or be moved elsewhere.

16) Line 88 “that are related by the underlying phylogenetic tree”. Technically speaking, relationships are determined by descent, and the phylogenetic tree is a representation of these relationships. Perhaps it would be more accurate to say “whose evolutionary relationship can be represented as a phylogenetic tree”.

17) Line 88-89. While cell biology is not my speciality, surely someone has observed “spatial patterns created by co-existing cancer clones” using staining (perhaps e.g. <https://www.nature.com/articles/labinvest201291>). Do the authors indeed stand by this statement, or should it be refined by adding another adjective to make the statement more precise (e.g. “genetically resolved subclones”)?

18) Line 90. I am questioning the use of “standard” to describe the WGS. Based on the general description of the workflow, this would need to be multi-sample WGS, which isn’t really standard (see my point 1a). Based on the authors response to my point 1a, this may have to change.

19) Line 108. What exactly is meant by “on target”? The authors should describe this in their methods.

20) Line 108-109. While the reporting of the number of detected signals is accurate, it lacks context. Perhaps this would be more informative if this was reported along with the number of mutant reads detected per cancer cell, and the number of wild type reads per wild type cell?

21) Lines 111, 114, 147, maybe others. There are well established descriptive terms for correlation

(e.g. moderate positive, or high positive correlation). The authors should replace the terms “largely consistent” and “concordant” with these terms.

22) Line 120. What is meant by “large scale” in the subheading? Can this be changed to something that is more informative, or just removed?

23) Line 149 – “Spatial Genomics Workflow”. This could be abbreviated to “SGW” (which is the reverse of “WGS”). Authors are free to choose.

24) Line 192 – “more likely”. Do the authors really believe this is “more likely” or is it more reasonable to postulate that it is “possible”.

25) Line 204, and others. The authors literally stand by their inference of clone age. I believe it would be more reasonable to not associate clone age with year (see my point 6). The authors may choose to revise this given their response to my point 6.

26) Line 232. Could these increased CD34 and FAP indicate cancer associated fibroblasts? Could this be reasonable associated with the FGFR1 amplification affected fibroblasts in the TME?

27) Line 233. I cannot see why Fig 2F is referenced here. Is that a mistake?

28) Line 295. The reference should be to figure 3 D,E (not figure 2).

29) Line 317-319 – I found this hard to follow: "For around half of the genes, at least 50% of the increase in expression level seen in the invasive cancer is evident within the late DCIS". Can the authors reword this to be clearer?

30) Line 320 – Ki67 is not a gene. Authors should correct this to MKI67

31) Line 368 – is the WGS data presented in Fig S5A subclonally resolved? If not, “subclonal” on line 368 should be removed.

32) Line 706, table of ISS immune panel. Typo – “Immunevasion”

FIGURES

33) Figure 1A. There isn't a arrow connecting the WGS analysis to the “Mathematically modelling box”. I think there should be one.

34) Figure 1 E, bottom left subpanel, PD9694d inset has one part not annotated to one of the clones (brown blob of cells below the green circled blob at the middle top)

35) 3B top panel, the “z” is missing in the labelling of “x, y, z”

36) 3E – label for GSTM1 is shifted up

37) Figure S4b states that this is the fold change of DCIS:invasive. These numbers are above 1, suggesting higher expression in DCIS, which is in contradiction with other parts of the text. Do the authors mean it to be the other way around?

Referee #2 (Remarks to the Author):

In their manuscript entitled “Spatial genomics maps the structure, character and evolution of cancer clones”, Lomakin et al. apply BaSISS, a new in situ sequencing and data analysis method to eight samples from two breast cancer patients. They visualize the distribution of subclones (which were defined by a preceding analysis of whole genome sequencing data) across tissue sections and overlay these data with gene expression measurements, with the goal of gaining new insights into subclonal evolution.

The tissue images in this paper are terrific, and it certainly is very interesting and rewarding to see the distribution of multiple mutations visualized in such a beautiful manner. There is no doubt that such images will be inspiring to others in the field. However, beyond the impressive pictures, I could not help but feel that the paper did not deliver any robust insights into subclonal evolution. Part of the issue may be the experimental design. The authors focused their analysis on a very small number of samples and chose to analyze these samples in very large depth (including overlay of RNA expression). Therefore, observations about the behavior of subclones remain anecdotal. It is largely unclear to what degree the patterns reflect the idiosyncrasies of individual evolutionary trajectories, and to what degree they may reflect the “laws” of breast cancer progression. The gene expression component does not seem to generate significant insight and is the weakest part of the paper, in my personal opinion. I cannot quite evaluate just how large the methodological contribution is that BaSISS represents (see first point of the major comments). If the technological advance were the main selling point and the biology were merely supposed to be an “example” of what the technology can do, then the manuscript would probably attain its goal. However, if novel biology were supposed to be the main contribution, then I think the authors would need a clearer hypothesis and more generalizable data that can really generate new insights in subclonal evolution. E.g. a quantitatively rigorous analysis of subclone mixing in 5 DCIS vs. 5 invasive cancer specimens would likely generate more generalizable, higher-level insights than extensive interpretation of gene expression patterns in two clones from one patient.

Major comments:

1. In my opinion, the BaSISS methodology is insufficiently explained, which makes it difficult to identify what (if anything) the technological advance is, at least to somebody who is not a deep expert in in situ sequencing. As the authors point out, previous papers have visualized the distribution of specific subclonal mutations in tissue sections (e.g. Baker et al. Nature Communications 2017). Can the authors articulate more clearly what exactly their technological improvement is? Further, the computational model specifically designed for the ISS data seems impressive, but a reading of the methods makes it clear that the raw imaging data are complex and

not easy to summarize. These difficulties are not addressed in the manuscript, and there appears to be no discussion of the assumptions that go into the Bayesian model (e.g. limited mixing of clones, p.33) or a critical evaluation of its performance. How is the reader to understand the relationship between the input data and the model output in detail, how is he/she to understand pitfalls and potential problems? To me personally, for example, it came as a complete surprise that I read only at the very end of the paper, in the discussion, that single cell genotyping is not possible with the method - the images in Figure 1D e.g. certainly seemed to suggest single cell genotypes? Perhaps the authors can consider possibilities for making their approach more transparent and more suitable for a broad audience of readers in the cancer evolution field?

2. The subclones that the authors chose to evaluate are very difficult to interpret biologically, and I think that a discussion of this is very much missing in the manuscript. In figure 1C and 2B, we are shown a subclone tree that raises many questions in my mind. The patient is exceptionally young (37 years, as found in Supplementary Table 1) and the tree spans her entire life. What the authors call “DCIS clones” are two mutations that - according to their timing estimates which are not further explained - arose during puberty or early adolescence. Is it fair to call the cells that carry these mutations “DCIS clones”? Formally this is probably accurate, but it may be misleading in that it suggests that these are two clones that emerged after their common ancestor became a DCIS – is this realistic given that at the time, the patient would have been ~12 years old? Is it not possible that these are just two mutations that occurred during the pubertal development of the breast and can presumably be found in other, healthy parts as well? The authors do not show any staining of normal ducts, so this is very hard to evaluate, but it is important because the authors spend a lot of time comparing these clones and possibly overinterpreting the differences between them. Certainly it seems questionable to me to call the orange clone “ill-fated” and imply in the follow-up analysis that the other, green clone must be more harmless – the phylogenetic tree suggests that both clones derive from clades that gave rise to a cancer eventually; the green clone shares a trunk with the purple (cancer) clone and presumably the only difference between the green and the orange clone is that the authors found a few mutations that are green-specific but none that are orange-specific? In PD9694c, the light blue clone (an ancestor of the green clone) and the purple cancer clone are found in similar proportions, are the authors saying that the light blue cells are not really “cancer”? I find this very confusing and would appreciate if the authors could clarify their arguments. Unfortunately, most of the motivation for the follow-up analysis rests on the conviction that the green clone somehow is less “dangerous” than the orange clone, which I cannot really conclude from the data as they are presented here.

3. It seems to me that the overlay of the subclone localization and gene expression data did not really generate many significant insights aside from the fact that some genes reached statistical significance in their differential expression, which in itself is not necessarily surprising. However, the biological implications are not so clear (see also point above). Most of the text associated with these analyses provides anecdotal results. The same applies to the lymph node metastasis analysis, where a few subclones are visualized and their exact growth patterns described in a lot of detail, with unclear significance. That lymph node metastases tend to be polyclonal has been reported by multiple groups. How representative or general the growth pattern analysis is, or what its significance is for disease progression, remains unclear.

Minor comments:

- It might be helpful if the authors incorporated uncertainties that still abound in the field of subclonal evolution into their text. For example, the beginning of the introduction says “Cancer growth is the result of mutation and selection of ever more proliferative clones analogous to Darwinian evolutionary theory. A consequence of this relentless process is that cancers are patchworks of genetically related but distinct groups of cells termed subclones.” - Statements like this ignore the fact that the field is still actively debating whether subclones are typically subject to significant selection or not, and I don’t think that everybody would agree that “Cancer growth is the result of selection of ever more proliferative clones”. (See work by Sottoriva, Curtis and others). The authors should probably avoid such sweeping statements and acknowledge controversies in the field a bit more explicitly. The text contains multiple other such oversimplifications.
- Related to the point above, the paper would benefit from more careful and formal wording throughout. E.g. I see no evidence that the orange and green clones are “competing” (p. 12)

Referee #3 (Remarks to the Author):

The authors report an implementation of Base-Specific In-Situ Sequencing (BaSISS), that allows highly multiplexed identification of mutations and gene expression in breast cancer tissue. This is a clever extension of an existing technology (ISS). The result is a complex and sophisticated method that the authors have developed and implemented to add a spatial dimension to studies of clonal evolution.

The ability to operate across large tissue areas at a resolution of ~ 100um is an impressive technical achievement, as is the ability to measure expression of >150 genes simultaneously. The data presented has been analysed carefully and with associated statistical analysis.

This technology addresses a major knowledge gap, namely the integration of genetic, spatial and phenotypic features of cancer cells. The manuscript is clearly written and the figures are high quality.

The manuscript has some deficiencies, the result being that the data presented is not always able to support the conclusions:

- 1) The method is relatively insensitive, thus clonal patterns and gene expression features are inferred from limited datapoints.
- 2) The method cannot map gene expression to individual cells. The result is that gene expression changes observed with reflect changes in cellular composition, rather than solely cell-autonomous changes in expression, as the authors claim.
- 3) Most results are based on analysis of a single tissue section taken from a small number of cancer cases and therefore do not support the generalised conclusions reached by the authors. The authors must analyse additional cases to support these biological conclusions.

It is not clear whether the method is a substantial new development or an incremental advance on previous methods. Several prior papers (doi.org/10.1007/s00418-017-1557-5; <https://doi.org/10.1038/s41467-017-02295-5>; doi.org/10.1038/nprot.2014.191), including some

from the authors of this manuscript (doi.org/10.1093/nar/gkv772; doi: 10.18632/oncotarget.1527; doi.org/10.1038/nmeth.1448), report on multiplexed transcript or mutant allele detection in tissue, by a similar strategy. Details of the methodological advances of this study compared to the field are very briefly provided in the introduction and not at all in the discussion. Therefore the novelty of this work is hard to evaluate.

The method has several limitations that warrant more substantial consideration. For instance, it relies on expression to detect mutations, meaning that sensitivity is a function of expression level of alleles. It also requires prior WGS and RNA-Seq data for the design of padlock probes which adds to the complexity and expense of the method.

The integration of mutation data with expression data is a strength of this technique, and the authors use it to describe phenotypic differences between clones, as shown in Figures 3 & 4. However it is also limited by the low resolution of this data. Many of the genes included in the 'immune' panel used in Figure 3 can be expressed by other lineages, such as stromal cells, thus the conclusions regarding immune dynamics may also be the result of other changes, such as stromal remodelling. This is also an issue in figure 4, where the different neoplastic cellularity of clones would confound the ability to assign gene expression features to specific cancer clones.

Breast cancer genomes are dominated by structural changes, such as copy number gain and loss (DOI: 10.1038/s41586-021-03357-x), with relatively few point mutations compared to other cancers (especially ER+ disease as shown in figure 1). As such, point mutations may be a poor surrogate for the clonal structure of these tumours under study. To address this, the authors should analyse WGS data for both point mutations and structural changes, to develop a higher resolution model of clonal diversity in these tumours, and their relationship to the clones detected using BaSISS.

The authors entirely rely on this one method for their findings. The validity of the key spatial findings would be greatly strengthened by validation using orthogonal methods applied to serial sections. This could include IHC, ISH or microdissection followed by sequencing for mutational patterns; or more sophisticated methods such as spatial transcriptomics for validation of expression profiles.

The authors use case PD9694 as a case study for 'pure DCIS' but this case also had invasive cancer in other regions. Therefore it is conceivable that regions of IDC contaminate the DCIS analysis and may lead to artefacts.

The authors make several speculative claims that are incompletely explained and appear to go beyond the evidence provided. For instance, the first of the "three key messages" at the end of the introduction states "Patterns of spatial genetic heterogeneity are profoundly influenced by resident tissue structures". This statement is presumably based on the varying location and cellular constituents of different clones. However, this is merely a correlation and the data does not provide information on directionality sufficient to claim that genetic heterogeneity varies as a function of tissue context, as suggested. Furthermore, this is based on a small number of cancer cases and clones so is susceptible to random variability and sampling error.

There are several other instances of broad interpretations of the data, for instance on page 9: "These appearances might arise due to mutual tolerance or equal fitness, but in this case, as

discussed below, the ability to spatially characterise co-existent clones provides evidence that we are more likely to be observing an incomplete clonal sweep by a fitter clone”.

Other statements can be considered obvious and predictable, for instance key message 2 “Coexistent genetic clones can have distinct transcriptional, histological and immunological characteristics”

Similarly, key message 3: “In preinvasive, invasive and locally metastatic breast tumours, the emergence of aggressive disease features can be temporally ordered and localised in genetic and histological contexts providing insights into the biology underlying cancer progression”. It is well accepted that clonal progression is associated with increasingly aggressive disease features, and previous data shows that clones are often spatially segregated.

Other comments:

There were a few areas in which the authors introduce and set parameters (e.g., “ $\tau=0.5$ as an example of our weak prior beliefs ...”, on page 35) but don’t really explore or explain what the impact would be to the results if these values are changed. I think at least some explanatory notes around how these are set would be useful.

At several times the authors refer to figures on masse (Eg “Fig. 2-4”), out of order, or refer to other figures within an earlier figure (Eg Fig 2A) which are confusing.

In figure 3, the authors show broad gene expression differences between DCIS and IDC for most of the genes comprising the expression panel. To be certain that this reflects genuine differences in the expression of these genes, how is expression normalised to account for differences in total transcriptional activity, RNA content or integrity between regions?

Referee #4 (Remarks to the Author):

This is an interesting study, well performed, albeit with limited number of cases, wherein subclonal patterns of evolution are matched with histological topographies and spatial transcriptomic data. The methodology used is informative for the scientific community, and this Reviewer would suggest the following items to clarify more the messages of the manuscript:

Major:

1. General comment:

a. The number of samples is very limited, and firm conclusions on the importance of clonal evolution in relation to morphology in breast cancer are hard to draw from this limited number of samples. However, this study is certainly of interest, and hints towards their method to be used in more extensive studies in order to confirm their findings. This Reviewer would therefore suggest that in the discussion, which so far is very limited, more emphasis is laid on the importance of matching clonal topography with what is seen by the pathologists using an HE, with some concrete examples as described in this manuscript (see also item 2). The examples of the regions of the cancer the authors analyze are very instructive and demonstrate the power of the method that was developed, yet much more samples and a more thorough analysis across much more samples are needed before

firm conclusions can be drawn on the causality and co-localization patterns between (sub)clonal evolution and morphology.

2. Pathology-related questions:

a. The authors mention in the abstract (line 29) and across the manuscript that fixed tissues were used, suggesting to the reader that it concerns FFPE-tissues. However, the methods section (line 490) details that it is merely frozen sections. Can the authors detail what exactly the matrix was, frozen vs FFPE, and what exactly the notion of “fixed tissues” is?

b. In line-item n°1, if as this Reviewer assumes that the work was done on frozen tissues, do the authors have any evidence on the applicability of the BaSISS method to FFPE-tissues, or whether fixation artefacts may influence their findings?

c. As this manuscript is of interest to the Pathology community, relating expression profiles with clonal patterns and localization on the HE-slide, this Reviewer would suggest adding much more detail on those variables of interest to pathologists. The morphology of the lesions investigated and the localization of the (sub)clonal patterns and expression data need to match much more with what the morphology and immunohistochemistry dictates (localization in the breast, exact subtype, exact histology, histological grade, TILs, location of those TILs, tumor size, presence of LVI, amount of necrosis, MAI, more details on the ER/PR/HER-score -which score was used -Allred-score, as a continuous variable, HER2 confirmed by FISH or not, absolute HER2/CEP17-copy numbers, etc....

Four non-limitative examples to illustrate what this Reviewer means:

i. The ER-, PR-status of the different subclones in the lobules with DCIS, is this recapitulated within the invasive cancer and LN-metastasis of this same case? Idem for the MAI for example.

ii. For the TILs we know that TILs surrounding DCIS may not be the same phenotype and importance as TILs in invasive cancer, and sometimes the invasive cancer does not contain immune cells, while the DCIS-component has plenty, suggesting that a linear evolution in immune-activity may not necessarily be that linear at all, with emergence of immune-evasive subclones.

iii. What about the clonal patterns and immune infiltration in normal lobules near DCIS and invasive cancer?

iv. How heterogeneous is histological grade, and how does the DCIS-grade compare with the grade of the invasive cancer and the lymph node metastasis, and with the subclonal patterns?

d. Also, what analysis was exactly done on what patient sample, is not 100% clear for this reviewer. There were 2 cases with 8 samples in total, of multifocal breast cancer, with each patient having 2 tumors. Was each tumor analyzed and how many regions/tumor/patient were analyzed for example? It may be clearer if it is described which analysis was done for which patients/ This Reviewer thinks he has reconstructed each analysis performed in relation to what sample from which patient, but it was puzzling. A figure of a breast showing a summary of their findings may be illustrative.

e. Line 419, on the data on VEGFA and CA9, which looking at figure 4G, the higher expression is merely based on the fact this image is right next to a necrotic zone, so it is normal that hypoxia-features are found. What are the expression patterns of VEGFA and CA9 in regions far from necrotic zones? This example suggest once more that accurate histopathological evaluation is crucial to understand any type of spatial profiling at the clonal and expression-level.

f. It seems that the LN-metastasis contains stroma, while not all LN-metastasis elicit a desmoplastic reaction. Some LN with desmoplasia have TIL-infiltration, while others do not, and some have mixed patterns, with cancer cells embedded within desmoplastic stroma, while in the same LN there is a purely parenchymatous localization of the cancer cells, without stromal reaction. The interpretation

of the signals in the involved LN in this study therefore puzzling. Are these the pre-existing immune cells that are analyzed, that have a different expression pattern than infiltrating immune cells (TILs) in the stroma of the involved LN?

g. The immune part of this paper is very limited, is mostly restricted to the DCIS, and the lymph node metastasis, while the most important relevance of immunity these days, is that observed within the invasive cancer, so in between the DCIS and the metastasis. Can the authors inform on one example of patterns of subclonal evolution within a single breast with multifocal cancer and relationship with immune expression at the genomic level, including the morphological observations of the patterns of the TILs, all this in order to complement their paper. This Reviewer understands that doing this analysis in all cases would actually be a new manuscript, which is not my intention. Rather, this suggestion is to complement the current findings in between the DCIS-stage and the analysis in the lymph node metastasis. This may be informative for any future research project using the technology, as suggested by the authors.

3. Methodology-related questions:

a. So far, as the authors are well aware, the number of reads depend on the cellular content and/or technical variation in RNA-capture, with variation of total read counts related to the morphology and the number of cells of different types, stromal, mostly immune cells and fibroblasts, versus epithelial, acknowledging that some cell types produce more RNA than others. In general, more reads mostly reflect more cell density. While with normalized read counts more contrasts between individual genes would be encountered, so are informative on its relative proportions, rather than absolute read counts which are biologically informative also, but do not reflect relative proportions. Can the authors comment whether comparing raw read counts with raw read counts normalized for total counts and with expression estimates and cell density in their analysis affect their interpretation of the data? This Reviewer would expect that with normalization some immune genes expressed in both immune as well as epithelial cells vs those expressed in immune cells alone would give different results, depending on whether a normalization was performed (cfr. Figure S4).

b. Related to 3a, it is well known that when the cancer cell fraction of a subclone decreases, the power to detect unique SNV decreases also, leading to an overestimation of the subclone's CCF. To what extent does the CCF in this study affect the fraction of SNV that could be missed, more specifically, what is the detection limit to reliably detect SNV's? The author used 25% CCF and 300 cells/mm². How did the authors come to these numbers?

Minor:

c. Page 15, line 351, the statement "Lymph Node metastasis predicts distant metastasis and death" is too strong as this depends on the subtype and much other characteristics. Maybe remove the words "and death"?

d. Digital Pathology is mentioned as a tool that was used? What exactly was measured and with what tool?

e. Abstract, line 33: just out of curiosity, why is mentioning "4.9 cm²", which is the amount of combined surface analyzed from the 2 patients relevant?

Author Rebuttals to Initial Comments:

Point by point replies

Referee #1:

Dear authors,

I have reviewed the work presented by Artem Lomakin and colleagues, where they present a workflow to infer in-situ spatial genomic heterogeneity and apply it to breast cancer samples to identify novel and interesting findings in tumor biopsies, including mapping of the subclonal structure and evolutionary patterns of (pre-)malignant clones, and combine this with spatial gene expression to identify transcriptional and immunological changes over tumor evolution and explain aggressiveness in distant metastasis.

The work is an incremental evolution of existing methodologies to investigate tumor heterogeneity via WGS to model clonal dynamics and ISS to profile spatial gene expression and point mutations, gelled together by an elegant computational method. I believe that the workflow is sufficiently novel and the results presented are also interesting, together defining a new cutting edge for exploiting spatial omics for understanding diseases. The manuscript is well written and mostly easy to understand.

With major revisions, I believe this work would be suitable for publication in a top tier journal. I hope to explain the shortfalls of the submitted work in the following points.

MAJOR

1) What 'exactly' is BaSISS? While this is not major, it is very central to the concept of the paper.

1a) A question I kept asking myself was "how is exactly BaSISS different from ISS?" Base-specific ISS makes it sound like a novel development of the ISS wetlab method. However, prior works by Mats Nilsson (<https://doi.org/10.1038/nmeth.2563> [doi.org]) already demonstrate the use of ISS for detecting "point mutations" (quoted from the abstract) suggesting that ISS already encompassed the concept of BaSISS. Also, the methods section of the submitted work combines BaSISS and ISS into a single sub-section, with no further details of what separates them. The group has also demonstrated its application to investigate tumor heterogeneity (<https://www.oncotarget.com/article/1527/text/> [oncotarget.com]). Given these prior works and no clear difference in the experimental procedure, I feel that the depiction of BaSISS being a new experimental method (figure 1A) is misleading. Are the wetlab methods used to profile point mutations as previously indeed similar/the same as previous works by the group of Mats

Nilsson? The authors should explicitly outline the differences between ISS and BaSISS, with the overarching aim of making it clear what BaSISS actually is, also specifically addressing whether it is a wetlab method for profiling point mutations in transcripts, or is it the entire spatial genomics workflow?

Thank you for raising this point that was also raised by other reviewers: BaSISS is an entire workflow. Given the fundamental importance of this, we have rewritten the manuscript (including introduction and abstract and results section entitled “Base Specific In Situ Sequencing Workflow”) and updated the overview figure Figure 1A, pasted above.

Specifically you ask how it differs from previously published work by Mats Nilsson (<https://doi.org/10.1038/nmeth.2563> [doi.org]). In the existing work ISS was used to detect a single mutation in cell line spike-in experiments. We have specified this in the introduction and more clearly specified the differences we employ – namely multiplexing and modelling to allow multiple clones (not just a single mutation) to be localised in a quantitative manner in the more challenging setting of clinical human tissue samples.

1b) A core part of the workflow is determining which tissues are profiled using WGS to identify which mutations are clonally heterogeneous. The authors depict that this is done through WGS of adjacent slices. However, in the methods section the authors seem to describe this in the context of multi-sample analysis.

We apologise for causing confusion here. The WGS approach used in these 2 cases was indeed multiregion, whereby we sequenced tissue from three different tissue blocks. However, to generate these DNA libraries we typically slice multiple thick sections, pool and digest, then extract DNA for library production. Other serial sections are meant to illustrate that different experiments are done on replicate sections (BaSISS, H&Es, ISS gene expression panels etc.). To improve clarity we have changed Figure 1A (see above) and added Figure S2B.

While this is reasonable, it is conflicting with the idea that the workflow can be meaningfully applied to a single sample as depicted in figure 1A. I feel that it would be extremely difficult to identify meaningful probes from a single patient sample. I am not sure to what extent the current workflow can identify subclonal mutations to target, but if not, then I would suggest fastclone (<https://doi.org/10.1038/s41467-020-18169-2> [doi.org], https://github.com/GuanLab/FastClone_GuanLab [github.com]). Unless the authors can demonstrate that demonstrate for the majority of their samples that the workflow would indeed be meaningful having WGS data from only a single sample, then the authors should amend the description of the their workflow, and figure 1A, to explicitly state that the workflow requires multiple samples.

Thank you for raising this point. Indeed, the concept of BaSISS is that it could be applied to any cancer with detectable subclones. Pan-cancer analyses have recently reported that the vast majority of individual cancer samples sequenced to moderate (40-60X) depth have detectable subclone structure (Dentro et al. 2017; Andor et al. 2016). We agree that it is important to

consider what could be achieved with a single cancer (as these are now so frequently sequenced in clinical contexts) and in the new manuscript version we endeavoured to show that BaSISS can detect meaningful subclones in individual samples. In response to your comments, in the new manuscript version, we therefore focus a section of the manuscript on precisely this idea (“Disentangling genetic and histological progression in primary breast cancers”). In this new section, by examining three primary breast cancers that contained both invasive cancer and preinvasive (DCIS) regions we demonstrate that BaSISS provides the opportunity to relate genetic clones to the histological elements across the whole tissue section. This seemed particularly topical as a recent study of 494,801 breast cancer patients in the NCDB database reported that two thirds of cancers have intermixed DCIS lesions (Kole et al. 2019). The findings of the analysis are summarised in Figure 3 (pasted above).

2) Validation of single cell point mutations. The temporal accumulation of point mutations should be validated using targeted single cell DNAseq (e.g. Tapestry) for at least 2 different samples.

We have validated the temporal ordering of mutations by performing laser capture microdissection (LCM) and low input library whole genome sequencing (WGS) of 19 micro biopsies from 4 of the samples in case 1. This is described in “Supp. Methods: LCM WGS Validation”, Figure 2C,G (pasted above). Furthermore, we have reported our spatial co-occurrence signal analyses (see “Supp. Methods: Allele co-occurrence analysis”). We demonstrate how one can predict which signals arise from the same cells based on spatial co-occurrence and validate this with correlation of WGS VAFs in microdissections in Figure 2E-F. We prefer the LCM WGS approach here over single cell analyses because genetic mutation detection in single cells remains imperfect and we do not feel that this would further strengthen the analyses.

In general, we have discussed our methods and validation approaches more explicitly in the text within an added section “Accurate and reproducible clonal maps of breast cancers” and Figure 2 (pasted above).

3) Validation ISS panels and comparative gene expression using scRNAseq.

3a) Comparisons of mutational clones to gene expression ISS raises interesting hypothesis, but without scRNAseq results the authors cannot claim that e.g. all gene in their "immune" panel originate from immune cells. The authors should perform scRNAseq with at least 2 samples to delineate whether the genes on the immune/cancer panel are indeed unique to immune/cancer cells, or if they are also present in non-immune cells in the tissue being analysed. The results can be presented as a confusion matrix of cell types identified, vs ISS marker genes. Based on the findings, the descriptive text of any ambiguous markers should be revised accordingly.

Thank you for this helpful suggestion. We have now used published scRNAseq breast cancer atlas (Wu et al, 2021) data to make a broad cell type classification. This allows us to 1) Look at the cell type compositional differences and 2) compare expression differences among clones within a specific cell type (e.g. only changes in Epithelial or Immune cells). The expression of

genes from the ISS panels and the marker genes used for cell-type annotation as well as the classification approach is described in Supp. Methods: Single cell typing, and in Figure S1C-D.

3b) Single cell RNAseq data could be used to validate the gene expression normalisation of ISS experiments. It is not clear to me how the gene expression obtained from ISS was normalised between different experiments. This should be fairly complex as each individual sample would be a batch, and perhaps even each probe. This makes me a little sceptical about the origins of global gene expression changes, such as those shown in figure 3E - I cannot tell how authors can distinguish between these genes being more expressed in more evolved clones, or if batch effects lead to global differences in gene expression detection sensitivity.

As with most RNA based experiments, batch effects do exist between samples and this would not necessarily be resolved by single cell analyses, that would also be subject to batch effects. Of note in this (and in the previous version of the manuscript), recognising the persistent impact of batch effects in any RNAseq experiment, we are particularly cautious not to draw comparisons between different samples (batches) but rather focus on comparisons within samples. However, we accept that the ISS gene expression analyses in the previous manuscript version could be improved and as discussed above we now use a cell-type based analysis of clone-specific expression and this is explained in "Supp. Methods: Single cell typing".

3c) Have the immune and cancer panel been described previously? If not, the authors would need to validate that the ISS probe design are indeed accurately reporting gene expression. This could be done by comparing the ISS results to scRNAseq of the same tissue.

The oncology expression panel (but not immune panel) is published and this citation has been added to the text (Svedlund et al. 2019). To ascertain if the ISS probes are accurately reporting gene expression differences between samples we used bulk RNAseq data from TN1, TN2 and LN1. We examined the variation in gene expression detected between pairs of related samples measured in RNAseq and ISS data. Confirming that the probes are detecting consistent variations in expression between samples, we observe high positive correlation ($R=0.63, 0.74, 0.75$) (scatterplots; Figure S2D) ($R=0.75$). Furthermore, cell typing approaches (Supp. Methods: Single cell typing) provide additional spatial confirmation that signals are related to the expected cell type. For example in Figure 5B, immune cell types (coloured dots) co-localise with CD45 IHC stained regions in the lymph node (immune cells = brown, cancer = lilac), and predicted B cells localise to germinal-like centre structures. Similarly, predicted epithelial cells co-localise with pan-cytokeratin stained regions.

4) Modelling sample fluctuations. The authors model sampling fluctuations using a negative binomial (NB) model, however they do not empirically show that that the data indeed follows a NB distribution. The Authors should indeed show that the ISS/BaSISS data follows a NB distribution or e.g. Marginalised Zero Inflated Poisson, or cite a paper where this has already been empirically shown.

Thank you for raising this issue which we haven't discussed in the previous version of the manuscript. To verify the appropriateness of the negative binomial model we calculated the tail probabilities of each BaSISS signal in each tile under the negative binomial model. If the model correctly accounts for the observed fluctuations across tiles, the probability that the observed BaSISS counts exceed their expected (modelled) value should be uniformly distributed, $r_i = \Pr(X_i > m_i) \sim \text{Unif}(0,1)$. Since the most concerning outliers manifest at the extreme of the unit interval it is instructive to assess the inverse normal transformed values of r_i , which should be $N(0,1)$ distributed, similar to a Pearson residual. The following diagnostic plot shows that this is largely the case, perhaps with the exception of *FGFR1* and *ZNF1mut* (which could indicate further biological variation at these loci, which are not modelled). We therefore conclude that a negative binomial model (in conjunction with the different noise terms shown in Extended Data Figure 2) sufficiently captures the technical variation of the data.

5) Spatial associations between cancer clones and normal tissue. I do not see usefulness of the "frequency plots" in figures 2A, 3B and 4E – they only show a single 1D plane and are not easily generalisable. The authors should attempt to model this quantitatively e.g. as done in figure 7 C,D,E in <https://doi.org/10.1053/j.gastro.2020.11.010>. [doi.org] The functionality has been implemented into SquidPy (<https://doi.org/10.1101/2021.02.19.431994> [doi.org], https://squidpy.readthedocs.io/en/latest/auto_examples/graph/compute_co_occurrence.html [squidpy.readthedocs.io])

Thank you for the suggestion to attempt to model spatial associations quantitatively. We have adopted a new method to model clone neighbourhoods as described in “Supp Methods: Clone neighbourhood analysis”.

The point of the frequency plots is to demonstrate that the underlying genetic clone data is quantitative and multidimensional and to show examples of rapid fluctuations in relation to presence or absence of histological structures (eg. anatomical border of a duct). To better convey this element of the data and allow the reader to explore the quantitative nature of the data we have developed an interactive browser to facilitate this (http://88.198.180.220:8080/test_map.html (“BaSISS Maps”).

6) Does the modelling over prioritise the WGS data? I came to this suspicion because of inconsistencies between figures 1D, in particular for the green clone for the PD9694c clone is clearly visible at the top of Fig 1D third panel, but not present in the bar chart in Figure 1D fourth panel, and also modelled as a earlier blue clone if the cancer clone maps shown in figure 1E. The authors should evaluate whether their computational model has a strong bias to preferentially model the WGS data over the spatial point mutations.

Thank you for raising this point. In response to your comments we did revisit our assumptions and priors and we added a section “Supp. Methods: Model limitations and assumptions”. For example, we were using a very conservative assumption that every area would be a mixture of clones to avoid overfitting. Indeed, relaxing this resulted in the green clone that you mention, reappearing in the replicate. In using relaxed priors, including reducing the WGS prior the model appears to perform well, as validated by our subsequent LCM validation experiments. However, running the model with no WGS priors does reduce performance with purple and blue clones merging and red and orange clones lacking differentiation. In the latter case this is likely due to the sparsity of red clone specific signals and may be avoided by denser allele tiling in future experiments as mentioned in our discussion. It is noted that given the first step in the BaSISS workflow is to perform subclone discovery by WGS, one can expect to have reasonable prior knowledge of clone composition in most cases, even taking into account the possibility of sampling bias.

INTERMEDIATE

7) A fundamental novelty of BaSISS is the downstream computational framework. While the authors have made a good effort in providing notebooks and a conda YAML file, however, the code is very unstructured (it seems to be a single undocumented python script called `pile_of_code.py` and a bunch of empty `__init__.py` files). The authors should improve the structure of their code, and improve their documentation to at least match the quality of e.g. novosparc (<https://github.com/rajewsky-lab/novosparc/> [github.com], <https://novosparc.readthedocs.io/> [novosparc.readthedocs.io]), or MutationTimeR.

The suggested changes have been implemented.

8) Age of tumors. The authors estimate the age of the tumor clones. In the case of patient PD9694 (age 37), the predicted age of the tumor seems to be nearly 40 and the initial events already being present at the age of 10 (figure 2B). I am not qualified to say whether this is plausible, but this seems to be very early. The authors should look into whether the literature support the idea that tumors can initiate from such an early age. If the authors cannot find support that this could be plausible, it would be prudent not to mislead readers with details of the tumor age in precise years.

Thank you for this comment. We do not wish to mislead readers with details of tumour age per se in years but rather the emergence of clonal lineages in relative time, we have therefore refined our language in the main text and removed explicit references to age of clones:

“ The predicted long natural history of the observed DCIS clones is reinforced by evolutionary timing estimates that, based on mutation burden, predict divergence of the two main lineages at less than 50% through evolutionary time which could equate to puberty or early adulthood given the age at surgery (100% evolutionary time) was 37 years (Figure 4B; Supp Methods). “

The age of the tree is intended to span the time from fertilised egg, through to midway through the patient's 37th year of life (i.e. surgical sampling) and we have attempted to make this more explicit in Figure 2B when we first present the tree (pasted above).

We have also added “Supp. Methods: Mutation timing estimates”. Evolutionary timing estimates are based on an assumption of a constant mutation rate. We accept that this assumption could be violated by the activity of different mutational signatures over time, and that a change in proliferation rate for example, in cancerous cells compared to normal cells does compromise our ability to pinpoint an exact chronological age of events. To address the first point we examined the mutational signatures within each branch of the cancer and reassuringly found that they are all clock-like (reflecting time or mitoses) (added : Table S2).

Regarding this being a surprising finding – sequencing of normal tissues is indeed a rapidly emerging field and the messages from studies in a range of tissues are similar: normal cells

acquire mutations at a relatively constant rate (approx. 10-30/ year, according to tissue) and driver mutations are remarkably common (Lee-Six et al. 2019; Moore et al. 2020; Yoshida et al. 2020; Abascal et al. 2021). In a study of normal endometrial tissues for example, several hundred somatic mutations had accumulated and a driver mutation was detected in virtually every sampled gland by ~70 years of age (Moore et al. 2020). In the context of cancer it has also been reported that somatic driver mutations can appear very early, even in embryogenesis (Coorens et al. 2019). Given the developmental pattern of breast duct morphogenesis that stops in infancy and then accelerates in response to hormonal surges during puberty the emergence of clonal expansions at this time seems consistent. Notably the longevity of the DCIS clones in this case is further supported by a radiological study that estimated DCIS growth rates from microcalcifications (Thomson et al. 2001). Based on such studies the distance spanned by the sampled clones would be consistent with some DCIS having been present for at least 10 years. This discussion has been added to the manuscript and Supp. Methods: Mutation timing estimates.

9) Figure 2D employs the use of Mann-Whitney U, however the size of the groups is very large and it would be highly appropriate to also report the effect size, e.g., using Cohen's D.

We found the digital analysis of nuclear features to be unnecessary in light of the extensive and detailed histopathological examination of the samples in the new manuscript version so have removed them.

10) The interpretation of cell type markers in the ISS gene paper should be reviewed. The genes used in the panels are fairly well established, however some are a little ambiguous, e.g. TNFRSF18 is a marker for T-regs, but are more highly expressed in NK-cells (<https://www.proteinatlas.org/ENSG00000186891-TNFRSF18/blood> [proteinatlas.org]). This would affect the interpretation of the results, such as line 308-314 where the authors describe localisation of TNFRSF18 signal to indicate the presence of T-reg cells.

Thank you for this very useful suggestion. We have replaced the expression analysis in the paper with a spatial single cell typing approach as described in Supp. Methods: Cell typing. Below you will find a snapshot of the new Figure S1C that shows single cell expression of the panel genes in a recently published breast single cell atlas (Wu et al. 2021). As you rightly pointed out this gene is expressed by T and NK cells, and to our surprise also by cancer cells. In light of this throughout the text we have been much more cautious in attributing any particular gene's expression to a certain cell type, including this particular example.

MINOR

11) Should BaSISS be described as "spatial genomics"? I understand that that mutations being probed stem from the genome, but the spatial analyte is mRNA. I feel that is someone interrogated bulk RNAseq for e.g. recurrent KRAS point mutations, this would still be referred to as transcriptomics. The authors should evaluate between themselves what they believe is the most accurate description of the work they have done, in relation to how it would have been described in the standards defined by work on bulk sequencing data.

Thank you for your suggestion, we take your point on board here. We have considered various alternatives but on reflection, given the focus is on mapping genetic clones we believe that this is probably the most accurate description of what the workflow does.

12) Readability of samples IDs and references to clones. The sample IDs do not make the paper easier to follow. Perhaps the can be renamed to something simpler, e.g. PD01-d01, PD1-c01, PD1-m01 (where d, c and m stand for for DCIS, cancer and metastasis). Also, I do not like the references to e.g. "the purple clone" in the text, however I cannot suggest anything better off the top of my head.

Thank you for this helpful suggestion. We have changed the clone/ sample IDs a) case 1: sample ER1,ER2, D1,D2,D3 whereby D indicates DCIS and ER indicates an ER positive primary tumour. b) case 2: TN1, TN2 (triple negative primary tumours) and LN1 (lymph node). ER+ and TN being the most common 'types' of breast cancer.

Regarding clone names we felt that retaining a link to the colour of the clone was necessary and helpful to the reader given the core model outputs are coloured maps. To relate the clones to relevant cases we have named them c1-colour for case 1 and c2-colour for case2.

13) While the conda YAML file is excellent, the authors should be specific about versions of tools in manuscript and/or reporting summary. For example, the version number (e.g. the hash commit or tag) of dpclust is not reported. Similarly, versions of libraries used within e.g. R and python should be reported.

Thank you, we have added these.

Comments 14-18 were no longer considered relevant (i.e. corrected or removed) in the revised manuscript.

14) Some in text references are to PMIDs.

15) Line 54-55 "they do not perform the critical function of isolating genetic subclones in tissue context because gene expression profiles are highly plastic". Can the authors support this with evidence that isolating genetic subclones using transcriptional readouts is not possible? Otherwise the sentence can end after "context", and the link of genetic subclones to transcriptional subclones can proceed this, or be moved elsewhere.

16) Line 88 "that are related by the underlying phylogenetic tree". Technically speaking, relationships are determined by descent, and the phylogenetic tree is a representation of these relationships. Perhaps it would more accurate to say "whose evolutionary relationship can be represented as a phylogenetic tree".

17) Line 88-89. While cell biology is not my speciality, surely someone has observed "spatial patterns created by co-existing cancer clones" using staining (perhaps e.g. <https://www.nature.com/articles/labinvest201291> [nature.com]). Do the authors indeed stand by this statement, or should it be refined by adding another adjective to make the statement more precise (e.g. "genetically resolved subclones")?

18) Line 90. I am questioning the use of "standard" to describe the WGS. Based on the general description of the workflow, this would need to be multi-sample WGS, which isn't really standard (see my point 1a). Based on the authors response to my point 1a, this may have to change.

19) Line 108. What exactly is meant by "on target"? The authors should describe this in their methods.

We have amended this statement: "On average, 97% of detected BaSISS spot signals were converted into feasible barcodes using GMM bar code deconvolution (Gataric et al., n.d.) (Table S4)."

20) Line 108-109. While the reporting of the number of detected signals is accurate, it lacks context. Perhaps this would be more informative if this was reported along with the number of mutant reads detected per cancer cell, and the number of wild type reads per wild type cell?

The number of detections per cell is now included in the text. We cannot distinguish between an individual cancer and normal to perform the latter in the context of BaSISS.

21) Lines 111, 114, 147, maybe others. There are well established descriptive terms for correlation (e.g. moderate positive, or high positive correlation). The authors should replace the terms "largely consistent" and "concordant" with these terms.

Noted and addressed

Comments 22-37 were no longer considered relevant (i.e. corrected or removed) in the revised manuscript.

22) Line 120. What is meant by "large scale" in the subheading? Can this be changed to something that is more informative, or just removed?

23) Line 149 – "Spatial Genomics Workflow". This could be abbreviated to "SGW" (which is the reverse of "WGS"). Authors are free to choose.

24) Line 192 – "more likely". Do the authors really believe this is "more likely" or is it more reasonable to postulate that it is "possible".

25) Line 204, and others. The authors literally stand by their inference of clone age. I believe it would be more reasonable to not associate clone age with year (see my point 6). The authors may choose to revise this given their response to my point 6.

26) Line 232. Could these increased CD34 and FAP indicate cancer associated fibroblasts? Could this be reasonable associated with the FGFR1 amplification affected fibroblasts in the TME?

27) Line 233. I cannot see why Fig 2F is referenced here. Is that a mistake?

28) Line 295. The reference should be to figure 3 D,E (not figure 2).

29) Line 317-319 – I found this hard to follow: "For around half of the genes, at least 50% of the increase in expression level seen in the invasive cancer is evident within the late DCIS". Can the authors reword this to be clearer?

30) Line 320 – Ki67 is not a gene. Authors should correct this to MKI67

31) Line 368 – is the WGS data presented in Fig S5A subclonally resolved? If not, "subclonal" on line 368 should be removed.

32) Line 706, table of ISS immune panel. Typo – "Immunevasion"

FIGURES

33) Figure 1A. There isn't a arrow connecting the WGS analysis to the "Mathematically modelling box". I think there should be one.

34) Figure 1 E, bottom left subpanel, PD9694d inset has one part not annotated to one of the clones (brown blob of cells below the green circled blob at the middle top)

35) 3B top panel, the "z" is missing in the labelling of "x, y, z"

36) 3E – label for GSTM1 is shifted up

37) Figure S4b states that this is the fold change of DCIS:invasive. These numbers are above 1, suggesting higher expression in DCIS, which is in contradiction with other parts of the text. Do the authors mean it to be the other way around?

Referee #2:

In their manuscript entitled "Spatial genomics maps the structure, character and evolution of cancer clones", Lomakin et al. apply BaSISS, a new in situ sequencing and data analysis method to eight samples from two breast cancer patients. They visualize the distribution of subclones (which were defined by a preceding analysis of whole genome sequencing data) across tissue sections and overlay these data with gene expression measurements, with the goal of gaining new insights into subclonal evolution.

The tissue images in this paper are terrific, and it certainly is very interesting and rewarding to see the distribution of multiple mutations visualized in such a beautiful manner. There is no doubt that such images will be inspiring to others in the field. However, beyond the impressive pictures, I could not help but feel that the paper did not deliver any robust insights into subclonal evolution. Part of the issue may be the experimental design. The authors focused their analysis on a very small number of samples and chose to analyze these samples in very large depth (including overlay of RNA expression). Therefore, observations about the behavior of subclones remain anecdotal. It is largely unclear to what degree the patterns reflect the idiosyncrasies of individual evolutionary trajectories, and to what degree they may reflect the "laws" of breast cancer progression. The gene expression component does not seem to generate significant insight and is the weakest part of the paper, in my personal opinion. I cannot quite evaluate just how large the methodological contribution is that BaSISS represents (see first point of the major comments). If the technological advance were the main selling point and the biology were merely supposed to be an "example" of what the technology can do, then the manuscript would probably attain its goal. However, if novel biology were supposed to be the main contribution, then I think the authors would need a clearer hypothesis and more generalizable data that can really generate new insights in subclonal evolution. E.g. a quantitatively rigorous analysis of subclone mixing in 5 DCIS vs. 5 invasive cancer specimens would likely generate more generalizable, higher-level insights than extensive interpretation of gene expression patterns in two clones from one patient.

Major comments:

1. In my opinion, the BaSISS methodology is insufficiently explained, which makes it difficult to identify what (if anything) the technological advance is, at least to somebody who is not a deep expert in in situ sequencing. As the authors point out, previous papers have visualized the distribution of specific subclonal mutations in tissue sections (e.g. Baker et al. Nature Communications 2017). Can the authors articulate more clearly what exactly their technological improvement is?

Thank you for raising this point that was also raised by other reviewers. Please see response to reviewer 1(Point 1a) and Figure 1A pasted above.

Further, the computational model specifically designed for the ISS data seems impressive, but a reading of the methods makes it clear that the raw imaging data are complex and not easy to summarize. These difficulties are not addressed in the manuscript, and there appears to be no discussion of the assumptions that go into the Bayesian model (e.g. limited mixing of clones, p.33) or a critical evaluation of its performance. How is the reader to understand the relationship between the input data and the model output in detail, how is he/she to understand pitfalls and potential problems?

Thank you for pointing this out, we agree it is an important missing element of the manuscript. The explanation of the model has been revised and we have added a section in “Supp. Methods: Model limitations and assumptions”, that specifically discusses the assumptions used in the model. Figure 1a is updated.

To me personally, for example, it came as a complete surprise that I read only at the very end of the paper, in the discussion, that single cell genotyping is not possible with the method - the images in Figure 1D e.g. certainly seemed to suggest single cell genotypes? Perhaps the authors can consider possibilities for making their approach more transparent and more suitable for a broad audience of readers in the cancer evolution field?

Thank you for highlighting this. We think that the misunderstanding arose from the histological images in Figure 1A that depict 3 round structures. These are DCIS filled ducts but in retrospect we can see that they may be interpreted as individual cells because they are obscured by the signals/ clone fields? We have adapted Figure 1A (pasted above) hopefully making the whole BaSISS workflow and concepts clearer.

2. The subclones that the authors chose to evaluate are very difficult to interpret biologically, and I think that a discussion of this is very much missing in the manuscript. In figure 1C and 2B, we are shown a subclone tree that raises many questions in my mind. The patient is exceptionally young (37 years, as found in Supplementary Table 1) and the tree spans her

entirely life. What the authors call "DCIS clones" are two mutations that - according to their timing estimates which are not further explained - arose during puberty or early adolescence. Is it fair to call the cells that carry these mutations "DCIS clones"? Formally this is probably accurate, but it may be misleading in that it suggests that these are two clones that emerged after their common ancestor became a DCIS – is this realistic given that at the time, the patient would have been ~12 years old? Is it not possible that these are just two mutations that occurred during the pubertal development of the breast and can presumably be found in other, healthy parts as well?

The authors do not show any staining of normal ducts, so this is very hard to evaluate, but it is important because the authors spend a lot of time comparing these clones and possibly overinterpreting the differences between them. Certainly it seems questionable to me to call the orange clone "ill-fated" and imply in the follow-up analysis that the other, green clone must be more harmless – the phylogenetic tree suggests that both clones derive from clades that gave rise to a cancer eventually; the green clone shares a trunk with the purple (cancer) clone and presumably the only difference between the green and the orange clone is that the authors found a few mutations that are green-specific but none that are orange-specific? In PD9694c, the light blue clone (an ancestor of the green clone) and the purple cancer clone are found in similar proportions, are the authors saying that the light blue cells are not really "cancer"? I find this very confusing and would appreciate if the authors could clarify their arguments. Unfortunately, most of the motivation for the follow-up analysis rests on the conviction that the green clone somehow is less "dangerous" than the orange clone, which I cannot really conclude from the data as they are presented here.

We apologise that our delivery seems to have caused some misunderstandings here. We have made several changes to the text and figures to make things clearer. We respond to each of the points you raise above in order.

Firstly, you are correct that the phylogenetic tree does span an individual's entire life - depicting the somatic mutations acquired from the fertilised egg until the last detectable clonal expansion. A number of normal tissue sequencing studies confirm that mutation accumulation is continuous(Lee-Six et al. 2019; Moore et al. 2020; Yoshida et al. 2020). We have added this information to the first presented tree in Figure 2B for clarity.

Secondly, you mention that "*DCIS clones are two mutations*". We think that the two mutations that you refer to are the 2 driver mutations in the cancer genes (*CREBBP* and *SF3B1*) in the trunk of the tree. To make things clearer we have reported the number of WGS mutations in each branch of the tree Figure 2B and all subsequent trees (example form Figure 4B). You will see that the total number of WGS mutations in the trunk is actually 447. Therefore the most recent common ancestor of the DCIS clones contains several hundred mutations and 2 driver mutations. We targeted 11 mutations (using 22 probes) from the trunk of the tree in our ISS experiment (Figure 1C).

Thirdly, you mention that “The authors do not show any staining of normal ducts” and as a consequence we do not know if we are just tracing mutations in the normal breast. We thank you for this valuable suggestion. There are several normal and even hyperplastic ducts in the tissues that we subjected to BaSISS. We have now highlighted these on the relevant images and discussed this in the text. You will see in Figure 3A for example that there is no staining of these regions, this indeed indicates that these regions are wildtype for the clones we trace.

Please find pasted below Figure 3A with normal ducts circled in yellow, hyperplasia in pink. Intriguingly, LCM WGS of two hyperplastic ducts from the pink region confirmed that these ducts are not genetically related (no somatic mutations shared with the clones we trace, however, several hundred were detected in 2 sampled adjacent hyperplastic ducts) to the neoplastic clones we are tracing (this is evident from heatmaps in Figure 2G; Table S3). 3C shows the subclone composition of multiple microregions across the tissue and confirms absence of clones from these areas.

We also show similar findings in Figure 4C: Normal ducts in blue. Histopathological examination of associated H&E confirmed transition from normal ductal epithelium to DCIS along the length of the indicated duct.

The following is added to the main text “ *Immune clusters and occasional normal or hyperplastic ducts appear white – consistent with a different genetic ancestry (Figure 4C). In D3 a 3mm length of (probable subsegmental) duct exhibits both a genetic and histological transition from normal ductal epithelium to DCIS along its length, confirming that although neoplastic involvement is extensive in this lobe, it is subtotal (Figure 4C).*”

Finally, it seems that the arguments about clone fate were potentially confusing or at least distracting and somewhat nuanced and specific to a single cancer. While we stand by our claims about the phylogenetic relationships between the histological states (DCIS and invasive histology), we have reworked the manuscript considerably to make these concepts clearer and more broadly applicable.

3. It seems to me that the overlay of the subclone localization and gene expression data did not really generate many significant insights aside from the fact that some genes reached statistical significance in their differential expression, which in itself is not necessarily surprising. However, the biological implications are not so clear (see also point above). Most of the text associated with these analyses provides anecdotal results. The same applies to the lymph node metastasis analysis, where a few subclones are visualized and their exact growth patterns described in a lot of detail, with unclear significance. That lymph node metastases tend to be polyclonal has been reported by multiple groups. How representative or general the growth pattern analysis is, or what its significance is for disease progression, remains unclear.

Thank you for this comment. We accept that we probably got a bit carried away and focused too much on specific biological insights that might be derived from the gene expression part of the data. In the new version of the manuscript we have tightened the focus on the main novel development, which is the clone maps. We place more focus on the idea that the spatial gene expression data is rather consistent across clones and supports the models predictions of clone

fields in DCIS (Figure 4H; Figure S4F-G) and in the different lymph node clones (Figure 5F-H; Figure S5C) scattered across large areas. We also use the gene expression data to perform cell typing and provide insights into clone specific cellular neighbourhoods (see Figure 3B,I and 5G,I and Supp. Methods: Cell typing; Supp. Methods: Clone neighbourhood analysis).

Minor comments:

- It might be helpful if the authors incorporated uncertainties that still abound in the field of subclonal evolution into their text. For example, the beginning of the introduction says "Cancer growth is the result of mutation and selection of ever more proliferative clones analogous to Darwinian evolutionary theory. A consequence of this relentless process is that cancers are patchworks of genetically related but distinct groups of cells termed subclones." - Statements like this ignore the fact that the field is still actively debating whether subclones are typically subject to significant selection or not, and I don't think that everybody would agree that "Cancer growth is the result of selection of ever more proliferative clones". (See work by Sottoriva, Curtis and others). The authors should probably avoid such sweeping statements and acknowledge controversies in the field a bit more explicitly. The text contains multiple other such oversimplifications.
- Related to the point above, the paper would benefit from more careful and formal wording throughout. E.g. I see no evidence that the orange and green clones are "competing" (p. 12)

Thank you for pointing this out. We have reworded the text to avoid such statements.

Referee #3:

The authors report an implementation of Base-Specific In-Situ Sequencing (BaSISS), that allows highly multiplexed identification of mutations and gene expression in breast cancer tissue. This is a clever extension of an existing technology (ISS). The result is a complex and sophisticated method that the authors have developed and implemented to add a spatial dimension to studies of clonal evolution.

The ability to operate across large tissue areas at a resolution of ~ 100um is an impressive technical achievement, as is the ability to measure expression of >150 genes simultaneously. The data presented has been analysed carefully and with associated statistical analysis. This technology addresses a major knowledge gap, namely the integration of genetic, spatial and phenotypic features of cancer cells. The manuscript is clearly written and the figures are high quality.

The manuscript has some deficiencies, the result being that the data presented is not always able to support the conclusions:

- 1) The method is relatively insensitive, thus clonal patterns and gene expression features are inferred from limited datapoints.

You are correct, this is unfortunately a major challenge inherent in these data. This is now highlighted in the discussion. Using laser capture microdissection WGS on multiple regions z-stacked tissues sections has allowed us to validate the model performance in dealing with these sparse and noisy data. A dedicated panel on the methods/ validation has been included to show that the clones that the model detects are nonetheless robust (Figure 2G).

2) The method cannot map gene expression to individual cells. The result is that gene expression changes observed with reflect changes in cellular composition, rather than solely cell-autonomous changes in expression, as the authors claim.

We have changed our gene expression analysis approach (Supp. Methods: Cell typing; Supp. Methods: Clone neighbourhood analysis; Figure S1C-D) so that now we not only report spatial gene expression per nucleus but also according to main cell types.

3) Most results are based on analysis of a single tissue section taken from a small number of cancer cases and therefore do not support the generalised conclusions reached by the authors. The authors must analyse additional cases to support these biological conclusions.

We acknowledge and accept that sequencing more samples using BaSIS will inevitably lead to many biological conclusions but it is clear that with the insights available from these approaches in itself would lead to many individual papers. The main aim of this paper however is not to make conclusive biological claims about a single cancer subtype/ entity, but rather to describe and demonstrate what is possible with what we believe is an extremely informative and novel way of studying cancers. We have rewritten the manuscript to emphasise this.

It is not clear whether the method is a substantial new development or an incremental advance on previous methods. Several prior papers (doi.org/10.1007/s00418-017-1557-5; <https://doi.org/10.1038/s41467-017-02295-5>; doi.org/10.1038/nprot.2014.191), including some from the authors of this manuscript (doi.org/10.1093/nar/gkv772; [doi: 10.18632/oncotarget.1527](https://doi.org/10.18632/oncotarget.1527); doi.org/10.1038/nmeth.1448), report on multiplexed transcript or mutant allele detection in tissue, by a similar strategy. Details of the methodological advances of this study compared to the field are very briefly provided in the introduction and not at all in the discussion. Therefore the novelty of this work is hard to evaluate.

We have rewritten the manuscript to provide closer focus on this point.

The method has several limitations that warrant more substantial consideration. For instance, it relies on expression to detect mutations, meaning that sensitivity is a function of expression level of alleles. It also requires prior WGS and RNA-Seq data for the design of padlock probes which adds to the complexity and expense of the method.

Limitations have been more clearly discussed in discussion:

“Limitations of the approach are relatively low sensitivity, which currently precludes single cell genotyping and a reliance on RNA and the resulting variation in gene expression levels of targeted transcripts. For these reasons, we focus on fresh frozen tissues in this study, as opposed to formalin fixed paraffin embedded tissues (FFPE) tissues that have been used for ISS but are generally technically less reliable²⁹. For standard ISS, additional sensitivity can be achieved by tiling transcripts with more probes; unfortunately, this is not feasible for point mutations at a defined genomic location. A practical solution for BaSISS would be to design for more targets per clone and to favour mutations with higher predicted expression levels, for example in higher copy number states. A switch to hybridisation based sequencing and direct RNA binding probes, which eliminate the requirement for reverse transcription are currently limited to gene expression, but with further development should also improve base specific detection several fold⁵⁷.

Particular advantages of the technology are that it is capable of interrogating very large tissue sections and it is comparably cheap, unlike solely relying on sequencing based methods⁵⁸. However, of course the deNovo mutation detection step must also be considered in terms of the complexity and cost of the entire workflow. The granularity of clones mapped by BaSISS depends on this critical phase. Here we used multi-region WGS (~40 fold coverage per sample) to identify and map relatively broad subclone populations but the spatial genomics approach could equally be applied to single genomes, high depth targeted capture approaches or more detailed phylogenies obtained through LCM or even single cell approaches. In theory, the approach also holds the potential to create three dimensional genomic tomographs by aligning consecutive tissue sections and this will be particularly important for examining branched anatomical structures such as breast, prostate and lung.”

The integration of mutation data with expression data is a strength of this technique, and the authors use it to describe phenotypic differences between clones, as shown in Figures 3 & 4. However it is also limited by the low resolution of this data. Many of the genes included in the 'immune' panel used in Figure 3 can be expressed by other lineages, such as stromal cells, thus the conclusions regarding immune dynamics may also be the result of other changes, such as stromal remodelling. This is also an issue in figure 4, where the different neoplastic cellularity of clones would confound the ability to assign gene expression features to specific cancer clones.

Thank you for this insightful point. We have fundamentally adjusted our expression analyses to address this. Please see Supp. Methods: Cell typing and Supp. Methods: Clone neighbourhood analysis.

Breast cancer genomes are dominated by structural changes, such as copy number gain and loss (DOI: 10.1038/s41586-021-03357-x), with relatively few point mutations compared to other cancers (especially ER+ disease as shown in figure 1). As such, point mutations may a poor surrogate for the clonal structure of these tumours under study. To address this, the authors should analyse WGS data for both point mutations and structural changes, to developed a

higher resolution model of clonal diversity in these tumours, and their relationship to the clones detected using BaSISS.

We agree that copy number is important to consider in cancer evolution studies. Virtually all breast cancers contain thousands of point mutations and it is point mutations, unlike CNVs, that serve as molecular clocks in phylodynamic analysis. The trees included in this paper were constructed as previously published using mutations that are clustered, whilst taking into copy number aberrations between samples such that for example, a copy number segment loss in one sample will not lead to the appearance of mutation gain in another (Yates et al. 2017; Yates et al. 2015). The method allows the incorporation of copy number changes in addition to rearrangement breakpoints (as long as they are expressed). We mention this in the text now in the workflow description and specifically in relation to case 2 (sample LN1) analysis.

The authors entirely rely on this one method for their findings. The validity of the key spatial findings would be greatly strengthened by validation using orthogonal methods applied to serial sections. This could include IHC, ISH or microdissection followed by sequencing for mutational patterns; or more sophisticated methods such as spatial transcriptomics for validation of expression profiles.

Thank you for this suggestion, we have included IHC and laser microdissection WGS validation (Figure 2F-G, Figure S2G; Figure S3C, E, Figure S4E) .

The authors use case PD9694 as a case study for 'pure DCIS' but this case also had invasive cancer in other regions. Therefore it is conceivable that regions of IDC contaminate the DCIS analysis and may lead to artefacts.

This is a good point. We cannot see any evidence of this from the spatial patterns of clones that we observe. The invasive clones c-purple and c-red are not detectable in DCIS samples. You will see from our new figures that we have now provided detailed views of subclone composition in tens to hundreds of micro-regions per sample, to give a better quantitative view of the detected clones (the main maps are in max projection). In Figure 4B below, for example, you can see that c1-purple clone (top) and c1-red clone (bottom) are completely absent in all assayed regions.

The authors make several speculative claims that are incompletely explained and appear to go beyond the evidence provided. For instance, the first of the "three key messages" at the end of the introduction states "Patterns of spatial genetic heterogeneity are profoundly influenced by resident tissue structures". This statement is presumably based on the varying location and

cellular constituents of different clones. However, this is merely a correlation and the data does not provide information on directionality sufficient to claim that genetic heterogeneity varies as a function of tissue context, as suggested. Furthermore, this is based on a small number of cancer cases and clones so is susceptible to random variability and sampling error.

There are several other instances of broad interpretations of the data, for instance on page 9: "These appearances might arise due to mutual tolerance or equal fitness, but in this case, as discussed below, the ability to spatially characterise co-existent clones provides evidence that we are more likely to be observing an incomplete clonal sweep by a fitter clone".

Thank you for pointing this out, we have removed these claims.

Other statements can be considered obvious and predictable, for instance key message 2 "Coexistent genetic clones can have distinct transcriptional, histological and immunological characteristics"

We do not find this "obvious" nor "predictable". As reviewer 2 points out, debate abounds in the field about the presence of selection vs drift in cancer evolution and so it is quite possible that some/ many/all of the genetically distinct clones that one can detect in the tissues might have very similar phenotypes if they are not under selection. Furthermore, the fact that we see the same clones sitting literally next to each other, and spanning complex duct systems, apparently together, with these very different phenotypes is most unexpected to us and usually generates intense debate whenever we present these data to a new audience.

Similarly, key message 3: "In preinvasive, invasive and locally metastatic breast tumours, the emergence of aggressive disease features can be temporally ordered and localised in genetic and histological contexts providing insights into the biology underlying cancer progression". It is well accepted that clonal progression is associated with increasingly aggressive disease features, and previous data shows that clones are often spatially segregated.

Again we cannot agree that this is necessarily the case. We have focused on the link between histological progression and genomic clonal progression in the new manuscript as these ideas might be well accepted within the framework of both of these models yet the relationship between the 2 models is not well established. What we find is that there are actually lags and disconnects between these states, revealing that situations are often more nuanced than one might assume.

Other comments:

There were a few areas in which the authors introduce and set parameters (e.g., " $\tau=0.5$ as an example of our weak prior beliefs ...", on page 35) but don't really explore or explain what the impact would be to the results if these values are changed. I think at least some explanatory notes around how these are set would be useful.

Good point, we have expanded the methods in accordance with this suggestion.

At several times the authors refer to figures on masse (Eg "Fig. 2-4"), out of order, or refer to other figures within an earlier figure (Eg Fig 2A) which are confusing.

In figure 3, the authors show broad gene expression differences between DCIS and IDC for most of the genes comprising the expression panel. To be certain that this reflects genuine differences in the expression of these genes, how is expression normalised to account for differences in total transcriptional activity, RNA content or integrity between regions?

Thank you, the new analysis is restricted to major cell types and clones based on single cell partitioning.

Referee #4:

This is an interesting study, well performed, albeit with limited number of cases, wherein subclonal patterns of evolution are matched with histological topographies and spatial transcriptomic data. The methodology used is informative for the scientific community, and this Reviewer would suggest the following items to clarify more the messages of the manuscript:

Major:

1. General comment:

a. The number of samples is very limited, and firm conclusions on the importance of clonal evolution in relation to morphology in breast cancer are hard to draw from this limited number of samples. However, this study is certainly of interest, and hints towards their method to be used in more extensive studies in order to confirm their findings. This Reviewer would therefore suggest that in the discussion, which so far is very limited, more emphasis is laid on the importance of matching clonal topography with what is seen by the pathologists using an HE, with some concrete examples as described in this manuscript (see also item 2). The examples of the regions of the cancer the authors analyze are very instructive and demonstrate the power of the method that was developed, yet much more samples and a more thorough analysis across much more samples are needed before firm conclusions can be drawn on the causality and co-localization patterns between (sub)clonal evolution and Morphology.

Thank you for these comments. We too are very excited about the opportunities that this approach has for delving into the relationship between genetics/clonal evolution and cancer morphology. In response to your suggestions we obtained highly detailed annotations of microanatomical areas (by enlisting the help of an experienced breast cancer pathologist who was blinded to the outcomes of the model) and have now focused the paper much more strongly in this direction. We were fascinated to observe extremely strong associations between morphology and the clone fields in DCIS, invasive and lymph node metastasis settings.

2. Pathology-related questions:

a. The authors mention in the abstract (line 29) and across the manuscript that fixed tissues were used, suggesting to the reader that it concerns FFPE-tissues. However, the methods section (line 490) details that it is merely frozen sections. Can the authors detail what exactly the matrix was, frozen vs FFPE, and what exactly the notion of "fixed tissues" is?

Fixation is performed with freshly prepared 3% (w/v) paraformaldehyde in DEPC-PBS for 5 min at RT. We recognise that the term "fixed tissues" will tend to be interpreted as FFPE so we have removed this from the text and made it clear these begin the protocol as frozen sections.

b. In line-item n°1, if as this Reviewer assumes that the work was done on frozen tissues, do the authors have any evidence on the applicability of the BaSISS method to FFPE-tissues, or whether fixation artefacts may influence their findings?

We have now mentioned this in discussion. ISS has been performed successfully in FFPE but yields are typically lower and less reliable, probably reflecting different states of degradation. Newer iterations of the ISS technology (HybRISS) that do not require reverse transcription offer better performance in FFPE tissues, but as we mention in the discussion, this is not currently feasible for point mutations due to the sequence specificity of this approach (Lee et al. 2022)

c. As this manuscript is of interest to the Pathology community, relating expression profiles with clonal patterns and localization on the HE-slide, this Reviewer would suggest adding much more detail on those variables of interest to pathologists. The morphology of the lesions investigated and the localization of the (sub)clonal patterns and expression data need to match much more with what the morphology and immunohistochemistry dictates (localization in the breast, exact subtype, exact histology, histological grade, TILs, location of those TILs, tumor size, presence of LVI, amount of necrosis, MAI, more details on the ER/PR/HER-score -which score was used -Allred-score, as a continuous variable, HER2 confirmed by FISH or not, absolute HER2/CEP17-copy numbers, etc.... Four non-limitative examples to illustrate what this

Thank you as stated above we have significantly enriched the pathological emphasis and details in the manuscript (also see Table S2). Note these are fresh frozen sections.

Reviewer means:

i. The ER-, PR-status of the different subclones in the lobules with DCIS, is this recapitulated within the invasive cancer and LN-metastasis of this same case? Idem for the MAI for example.

The lymph node and the DCIS samples you refer to are a separate case. Please see our response to d. In case 1 all DCIS and invasive cancer is ER+, no heterogeneity of ER staining was observed. You will see that PR status is variable amongst DCIS clones but PR+ in all invasive clones.

ii. For the TILs we know that TILs surrounding DCIS may not be the same phenotype and importance as TILs in invasive cancer, and sometimes the invasive cancer does not contain immune cells, while the DCIS-component has plenty, suggesting that a linear evolution in immune-activity may not necessarily be that linear at all, with emergence of immune-evasive subclones.

We agree, and to avoid over generalising we have removed this element of the results section and instead (as per your comments below) we have chosen to place more focus on the differences between invasive clones in TN1.

iii. What about the clonal patterns and immune infiltration in normal lobules near DCIS and invasive cancer?

We found a few normal lobules and they lacked the target clones (wildtype). While they did contain some immune cells, we do not have power to detect anything statistically significant in these analyses and will need larger cohorts to make meaningful statements.

iv. How heterogenous is histological grade, and how does the DCIS-grade compare with the grade of the invasive cancer and the lymph node metastasis, and with the subclonal patterns?

This is a great point. It transpired that the grade was very clone specific in the DCIS case. Please see Figure 4 and Figure S4. Essentially: DCIS clone c1-orange is intermediate grade, DCIS c1-green is usually low grade, both invasive cancers were grade 2.

d. Also, what analysis was exactly done on what patient sample, is not 100% clear for this reviewer. There were 2 cases with 8 samples in total, of multifocal breast cancer, with each patient having 2 tumors. Was each tumor analyzed and how many regions/tumor/patient were analyzed for example? It may be clearer if it is described which analysis was done for which patients/ This Reviewer thinks he has reconstructed each analysis performed in relation to what sample from which patient, but it was puzzling. A figure of a breast showing a summary of their findings may be illustrative.

Thank you, this is a very helpful suggestion. We have addressed this confusion by making relationships between cases/ samples clearer (please see newly added figure 1B):

e. Line 419, on the data on VEGFA and CA9, which looking at figure 4G, the higher expression is merely based on the fact this image is right next to a necrotic zone, so it is normal that hypoxia-features are found. What are the expression patterns of VEGFA and CA9 in regions far from necrotic zones? This example suggest once more that accurate histopathological

evaluation is crucial to understand any type of spatial profiling at the clonal and expression-level.

Indeed, it is in necrotic zones. The point we wished to make though is that these necrotic zones are clone specific. We believe that this is clearer in the revised text and figures.

f. It seems that the LN-metastasis contains stroma, while not all LN-metastasis elicit a desmoplastic reaction. Some LN with desmoplasia have TIL-infiltration, while others do not, and some have mixed patterns, with cancer cells embedded within desmoplastic stroma, while in the same LN there is a purely parenchymatous localization of the cancer cells, without stromal reaction. The interpretation of the signals in the involved LN in this study therefore puzzling. Are these the pre-existing immune cells that are analyzed, that have a different expression pattern than infiltrating immune cells (TILs) in the stroma of the involved LN?

We are not sure how we can differentiate between cells that pre-existed vs are more recent. We have adapted our analysis and presentation here to focus on the key messages: we detect 2 genetically distinct clones scattered through the lymph node, and we find that they have distinct morphologies and transcriptional phenotypes and occupy distinct 'niches'. We have addressed our gene expression analysis to allow us to report immune cell specific and cancer specific expression patterns as described in: Supp. Methods: Cell typing and Supp. Methods: Clone neighbourhood analysis.

g. The immune part of this paper is very limited, is mostly restricted to the DCIS, and the lymph node metastasis, while the most important relevance of immunity these days, is that observed within the invasive cancer, so in between the DCIS and the metastasis. Can the authors inform on one example of patterns of subclonal evolution within a single breast with multifocal cancer and relationship with immune expression at the genomic level, including the morphological observations of the patterns of the TILs, all this in order to complement their paper. This Reviewer understands that doing this analysis in all cases would actually be a new manuscript, which is not my intention. Rather, this suggestion is to complement the current findings in between the DCIS-stage and the analysis in the lymph node metastasis. This may be informative for any future research project using the technology, as suggested by the authors.

Thank you for this valuable suggestion. Please see Figure 3, sample TN1. By adding this case to the discussion of clonal evolution in relation to DCIS to invasive transition we demonstrate that the detected clonal progression actually localises to the invasive compartment. We find that the switch in genotype is echoed closely by a change in nuclear features and cellular architecture, cancer expression and a change in the microenvironment (more fibroblasts and immune, mainly myeloid cells).

3. Methodology-related questions:

a. So far, as the authors are well aware, the number of reads depend on the cellular content and/or technical variation in RNA-capture, with variation of total read counts related to the morphology and the number of cells of different types, stromal, mostly immune cells and fibroblasts, versus epithelial, acknowledging that some cell types produce more RNA than others. In general, more reads mostly reflect more cell density. While with normalized read

counts more contrasts between individual genes would be encountered, so are informative on its relative proportions, rather than absolute read counts which are biologically informative also, but do not reflect relative proportions. Can the authors comment whether comparing raw read counts with raw read counts normalized for total counts and with expression estimates and cell density in their analysis affect their interpretation of the data? This Reviewer would expect that with normalization some immune genes expressed in both immune as well as epithelial cells vs those expressed in immune cells alone would give different results, depending on whether a normalization was performed (cfr. Figure S4).

Thank you for this comment. In the previous manuscript we indeed did not know the origin of the gene expression but we have now developed the methods such that ISS signals are now detected by cell and also by cell type. These are described in: Supp. Methods: Cell typing and Supp. Methods: Clone neighbourhood analysis.

b. Related to 3a, it is well known that when the cancer cell fraction of a subclone decreases, the power to detect unique SNV decreases also, leading to an overestimation of the subclone's CCF. To what extent does the CCF in this study affect the fraction of SNV that could be missed, more specifically, what is the detection limit to reliably detect SNV's? The author used 25% CCF and 300 cells/mm². How did the authors come to these numbers?

This is essentially a pragmatic selection of level to show a sense of clone locations. We experimented by attempting to show isolines and gradient shading but this was difficult to interpret. The interactive web browser permits a more complete appreciation of the quantitative nature of the data.

Minor:

c. Page 15, line 351, the statement "Lymph Node metastasis predicts distant metastasis and death" is too strong as this depends on the subtype and much other characteristics. Maybe remove the words "and death"?

Removed.

d. Digital Pathology is mentioned as a tool that was used? What exactly was measured and with what tool?

A neural networks approach to detect and measure nuclear features (supp methods).

In the new manuscript we use freely available quPath software to detect stained cells/ nuclei in the new IHC validation work (PGR, PTEN and Ki67). We restricted the analysis to regions where we also validated the clones with LCM WGS. This was necessary as z-stacking was tens of microns through the block and we could not be certain of the relationship to the BaSISS sections.

e. Abstract, line 33: just out of curiosity, why is mentioning "4.9 cm²", which is the amount of combined surface analyzed from the 2 patients relevant?

It is merely to highlight the scale of the technology. Some genomics approaches operate across tiny areas. It is not present in the current manuscript.

Reviewer Reports on the First Revision:

Referees' comments:

Referee #1 (Remarks to the Author):

Dear authors,

I have reviewed the resubmitted work and response letter by Artem Lomakin, Jessica Svedlund and colleagues. Their resubmitted work addresses the concerns that I previously raised. I stand by my previous statements that the work is suitable for publication in a top-tier journal.

While I have raised some minor issues and remarks, I would like to congratulate the authors for an excellent piece of work.

Minor

1. The concept of microregions (e.g. Fig 3) is not immediately clear. I only understood this after looking at the supplementary figures where one example labels all the regions in the tissue and in the heatmap. Inconsistencies in how the authors referred to these microregions added to the confusion. The authors should:

1a. Use either “microregion” (Fig 3) or “microanatomical regions” (Fig 4) or “genetic clone composition in microanatomical regions” (Fig 5) as consistent titles

1b. Use the same/similar layout for the microregion heatmaps between e.g. Fig 3 and 4

1c. I assume that Fig 5D also related to microregions, but this isn't explicit

1d. Perhaps refer to the supplemental figures that demark the microregions in the main figure captions

2. Fig 2D right panel – images have black background, but the panel is dark grey

3. Fig 2D – add co-occurrence plot of imputed?

4. Fig 2G has a stray “LCM WGS sample” y-axis label

5. Figure 3D bottom – what are “growth patterns”?

6. 3I, 5E – y axis label should be “p-value” or the units should be changed for log10. Tick and label marking is awkward – please make this easier to interpret by e.g. marking 0, marking landmark p-values (e.g. 10^{-10} , 10^{-20}) rather than seemingly random ones (e.g. Fig 5E shows 10^{-7} followed by 10^{-33})

7. The sentence “An interactive web browser, <https://www.cancerclonemaps.org/>, allows the reader to explore the quantitative nature of these data” seems to be directly addressing the reader – in my opinion, it would be less awkward to simply state that the results can be interactively viewed on the website without referring to the reader.

Remarks

- I am happy to see that the authors have also cited recent works by Zhao et al (slide-DNA-seq).
- I am still not a fan of the 1D clone density plots (e.g. bottom panel of Fig 2E), especially since the analysis is invariant to rotation and it provides no obvious benefit over the heatmaps of the microanatomy (in my opinion). The current state-of-the-art is inferring 2D/3D organisation of cells from non-spatial data (e.g. <https://doi.org/10.1038/s41467-020-15968-5>, <https://doi.org/10.15252/msb.20209438>, <https://doi.org/10.1038/s41422-020-0353-2>, <https://doi.org/10.1038/s41586-019-1773-3>), so I am frustrated by the analysis of a random 1D plane of 2D data in this study. The authors postulate that BaSISS might also be applicable for 3D analysis – would the authors still feel that these 1D clone density plots are really the best way of presenting the data? However, I admit that I had fun investigating this on your website ;p

Referee #2 (Remarks to the Author):

In this appeal, Lomakin et al. have substantially reworked their original submission to address a variety of reviewer suggestions and concerns.

Upon re-reading the paper, my updated impression is:

- The text has now been cleared of overreaching statements and interpretations. A large concern with the original submission was that the authors (sometimes substantially) overstated the biological conclusions that could be drawn from limited/anecdotal observations. This problem has been resolved; interpretations now appear largely appropriate throughout.
- The authors have clarified that they consider the technological advance of BaSISS to be the main achievement of this work. The original version left me with some confusion as to whether the authors thought that biological insights into subclonal evolution or technology development were the main selling point of the manuscript. The revised version focuses on the technological aspect more clearly.
- I cannot judge whether the technical advance presented here merits publication in a top tier journal, as I am not a spatial genomics/transcriptomics expert. Somewhat worryingly, multiple reviewers mentioned during the first round of review that they could not tell whether BaSISS represented an incremental improvement over existing methods (essentially a type of “scaling up” of existing single-mutation methodologies to visualize multiple expressed mutations through multiplexing) or whether there is something fundamentally new here. I continue to be unable to judge this and will leave other reviewers to comment on this aspect.
- Similarly, the computational innovation or quality of the BaSISS workflow is hard for me to judge, given that the data that is generated by BaSISS is highly specific to the method and the challenges/pitfalls associated with such data analysis cannot be easily evaluated from the outside.
- The part that I feel qualified to evaluate – the authors’ insights into the laws of tumor evolution – has not changed substantially in this revised submission. As in the original manuscript, largely anecdotal observations are reported (albeit much more cautiously than previously, which is an

improvement) and it remains unclear what level of generality or importance can or should be assigned to them. In their rebuttal, the authors argue that it is typical of “landmark studies” in cancer genomics to feature individual cases and not necessarily provide any generalizable insights. I will leave it to the expert reviewers in spatial genomics/transcriptomics to judge whether the authors are correct in placing their study in that realm of technical advance.

Referee #3 (Remarks to the Author):

The paper has undergone marked revision and development. Congratulations to the authors for a far more clear and impactful manuscript. They have addressed my major concerns

I have some minor corrections to suggest:

Some terminology is still confusing. Eg in Fig 3A how is there a ‘case c1’ from two different tumours? Case typically refers to a clinical case. And c1 is alternatively used to label clone 1

The Figure legend refers to “case 1 samples” , which is confusing. A clone labelling system that includes the patient, sample and clone would be very helpful. Eg P1-ER1-c1
Clones are also referred to variously as “c1-purple in ER2” and “c1-purple (ER2)”

So please make nomenclature consistent and provide more explanation of nomenclature. A supplementary figure showing the relationships of all patients, samples, cases, clones would be helpful

Please provide some more details on the sensitivity and resolution of the gene expression work for example in Figure 3G. How many of these gene expression changes are directly measured at cellular resolution?

This paragraph starting with “There are various possible explanations for the observed DCIS growth patterns” Would probably best belong in the discussion

Referee #4 (Remarks to the Author):

The authors have adequately replied to the comments and suggestions raised, and have developed a much better and comprehensive analysis, better illustrated and validated than in the first version reviewed. Much congrats from this reviewer.

Reviewer Reports on the First Revision:

27th August 2022

We thank all of the reviewers for their continued positive feedback and comments. In response to their comments, we have addressed the highlighted inconsistencies in nomenclature. To comply with editorial requests, including to reduce the manuscript in size we have tightened the text, in particular removing a substantial part of the discussion to the supplementary notes. Figures 2-5 have also been reduced in size and complexity by moving more items to Extended Data.

Referees' comments:

Referee #1 (Remarks to the Author):

Dear authors,

I have reviewed the resubmitted work and response letter by Artem Lomakin, Jessica Svedlund and colleagues. Their resubmitted work addresses the concerns that I previously raised. I stand by my previous statements that the work is suitable for publication in a top-tier journal.

While I have raised some minor issues and remarks, I would like to congratulate the authors for an excellent piece of work.

Minor

1. The concept of microregions (e.g. Fig 3) is not immediately clear. I only understood this after looking at the supplementary figures where one example labels all the regions in the tissue and in the heatmap. Inconsistencies in how the authors referred to these microregions added to the confusion. The authors should:

1a. Use either "microregion" (Fig 3) or "microanatomical regions" (Fig 4) or "genetic clone composition in microanatomical regions" (Fig 5) as consistent titles

1b. Use the same/similar layout for the microregion heatmaps between e.g. Fig 3 and 4

1c. I assume that Fig 5D also related to microregions, but this isn't explicit

1d. Perhaps refer to the supplemental figures that demark the microregions in the main figure captions

Thank you for highlighting this. We have used clearer and more consistent language in text, figures and legends ("microregions") and as suggested referred the reader to Extended Data Figures and/or the web viewer to enable a better understanding of what these are.

2. Fig 2D right panel – images have black background, but the panel is dark grey

Corrected

3. Fig 2D – add co-occurrence plot of imputed?

We did not feel that this adds anything beyond what is conveyed by the existing heatmaps.

Furthermore, the co-occurrence plots also had to be moved to Extended Data in line with journal formatting requirements.

4. Fig 2G has a stray “LCM WGS sample” y-axis label

This is meant to be the y axis title but now we see it might be confusing so have moved it and used a clearer title “Region ID”.

5. Figure 3D bottom – what are “growth patterns”?

We have clarified this in the legend: “different growth patterns, defined by distinct nuclear and architectural features,”. And pointed the reader to the methods for more details

6. 3I, 5E – y axis label should be “p-value” or the units should be changed for log10. Tick and label marking is awkward – please make this easier to interpret by e.g. marking 0, marking landmark p-values (e.g. 10^{-10} , 10^{-20}) rather than seemingly random ones (e.g. Fig 5E shows 10^{-7} followed by 10^{-33})

We have amended this.

7. The sentence “An interactive web browser, <https://www.cancerclonemaps.org/>

[cancerclonemaps.org](https://www.cancerclonemaps.org/)], allows the reader to explore the quantitative nature of these data” seems to be directly addressing the reader – in my opinion, it would be less awkward to simply state that the results can be interactively viewed on the website without referring to the reader.

Fair, amended.

Remarks

- I am happy to see that the authors have also cited recent works by Zhao et al (slide-DNA-seq).
- I am still not a fan of the 1D clone density plots (e.g. bottom panel of Fig 2E), especially since the analysis is invariant to rotation and it provides no obvious benefit over the heatmaps of the microanatomy (in my opinion). The current state-of-the-art is inferring 2D/3D organisation of cells from non-spatial data (e.g. <https://doi.org/10.1038/s41467-020-15968-5> [doi.org], <https://doi.org/10.15252/msb.20209438> [doi.org], <https://doi.org/10.1038/s41422-020-0353-2> [doi.org], <https://doi.org/10.1038/s41586-019-1773-3> [doi.org]), so I am frustrated by the analysis of a random 1D plane of 2D data in this study. The authors postulate that BaSISS might also be applicable for 3D analysis – would the authors still feel that these 1D clone density plots are really the best way of presenting the data? However, I admit that I had fun investigating this on your website ;p

We share the reviewer's frustration with the difficulties of spatial multidimensional data representation in 2D space. 1D cross-section plots in the paper serve mainly as a reminder to a reader that in some cases it is an uneven mixture of clones that hides behind a max projected coloured clone map. We did experiment with the clone specific heatmap representation but found that 1) it takes up more space, 2) the dynamic range of clonal density is often lost when represented

only with colours. In addition, the choice of 1D plots for the online viewer was based on the compactness and simplicity of implementation. However, we do agree that interactive toggleable multilayer gaussian field visualisation is more useful for investigation of spatial patterns and we will continue to work on its implementation further ^-^

Referee #2 (Remarks to the Author):

In this appeal, Lomakin et al. have substantially reworked their original submission to address a variety of reviewer suggestions and concerns.

Upon re-reading the paper, my updated impression is:

- The text has now been cleared of overreaching statements and interpretations. A large concern with the original submission was that the authors (sometimes substantially) overstated the biological conclusions that could be drawn from limited/anecdotal observations. This problem has been resolved; interpretations now appear largely appropriate throughout.
- The authors have clarified that they consider the technological advance of BaSISS to be the main achievement of this work. The original version left me with some confusion as to whether the authors thought that biological insights into subclonal evolution or technology development were the main selling point of the manuscript. The revised version focuses on the technological aspect more clearly.
- I cannot judge whether the technical advance presented here merits publication in a top tier journal, as I am not a spatial genomics/transcriptomics expert. Somewhat worryingly, multiple reviewers mentioned during the first round of review that they could not tell whether BaSISS represented an incremental improvement over existing methods (essentially a type of “scaling up” of existing single-mutation methodologies to visualize multiple expressed mutations through multiplexing) or whether there is something fundamentally new here. I continue to be unable to judge this and will leave other reviewers to comment on this aspect.
- Similarly, the computational innovation or quality of the BaSISS workflow is hard for me to judge, given that the data that is generated by BaSISS is highly specific to the method and the challenges/pitfalls associated with such data analysis cannot be easily evaluated from the outside.
- The part that I feel qualified to evaluate – the authors’ insights into the laws of tumor evolution – has not changed substantially in this revised submission. As in the original manuscript, largely anecdotal observations are reported (albeit much more cautiously than previously, which is an improvement) and it remains unclear what level of generality or importance can or should be assigned to them. In their rebuttal, the authors argue that it is typical of “landmark studies” in cancer genomics to feature individual cases and not necessarily provide any generalizable insights. I will leave it to the expert reviewers in spatial genomics/transcriptomics to judge whether the authors are correct in placing their study in that realm of technical advance.

Referee #3 (Remarks to the Author):

The paper has undergone marked revision and development. Congratulations to the authors for a far more clear and impactful manuscript. They have addressed my major concerns

I have some minor corrections to suggest:

Some terminology is still confusing. Eg in Fig 3A how is there a 'case c1' from two different tumours? Case typically refers to a clinical case. And c1 is alternatively used to label clone 1

The Figure legend refers to "case 1 samples", which is confusing. A clone labelling system that includes the patient, sample and clone would be very helpful. Eg P1-ER1-c1
Clones are also referred to variously as "c1-purple in ER2" and "c1-purple (ER2)"

So please make nomenclature consistent and provide more explanation of nomenclature. A supplementary figure showing the relationships of all patients, samples, cases, clones would be helpful

We have updated the nomenclature and consistency.

Please provide some more details on the sensitivity and resolution of the gene expression work for example in Figure 3G. How many of these gene expression changes are directly measured at cellular resolution?

This has been added to the Single Cell Typing section of the Supplementary Methods section "A conservative distance cut-off of 5µm was used to decrease the chance of misannotation which led to the ~30% of signal loss." All the reported expression changes are based on the ISS signals, assigned to single cells (within 5µm radius from the nucleus centre).

This paragraph starting with "There are various possible explanations for the observed DCIS growth patterns" Would probably best belong in the discussion

This has been removed from the main text through formatting to achieve a more succinct format as requested by the editor.

Referee #4 (Remarks to the Author):

The authors have adequately replied to the comments and suggestions raised, and have developed a much better and comprehensive analysis, better illustrated and validated than in the first version reviewed. Much congrats from this reviewer.